# Quantitative imaging of loop extruders rebuilding interphase genome architecture after mitosis

Andreas Brunner[1,2], Natalia Rosalía Morero[1], Wanlu Zhang[1], M. Julius Hossain[3], Marko Lampe[4], Hannah Pflaumer[1], Aliaksandr Halavatyi[4], Jan-Michael Peters[5], Kai S. Beckwith[1,6], and Jan Ellenberg[1,7]

How cells establish the interphase genome organization after mitosis is incompletely understood. Using quantitative and super-resolution microscopy, we show that the transition from a Condensin to a Cohesin-based genome organization occurs dynamically over 2 h. While a significant fraction of Condensins remains chromatin-bound until early G1, Cohesin-STAG1 and its boundary factor CTCF are rapidly imported into daughter nuclei in telophase, immediately bind chromosomes as individual complexes, and are sufficient to build the first interphase TAD structures. By contrast, the more abundant Cohesin-STAG2 accumulates on chromosomes only gradually later in G1, is responsible for compaction inside TAD structures, and forms paired complexes upon completed nuclear import. Our quantitative time-resolved mapping of mitotic and interphase loop extruders in single cells reveals that the nested loop architecture formed by the sequential action of two Condensins in mitosis is seamlessly replaced by a less compact but conceptually similar hierarchically nested loop architecture driven by the sequential action of two Cohesins.

## Introduction

DNA loop extrusion by structural maintenance of chromosomes (SMC) complexes has emerged as a key principle in the spatial organization of chromosomes during interphase and mitosis (Yatskevich et al., 2019; Davidson and Peters, 2021). In mitosis, the two pentameric ring-like Condensin complexes I and II, consisting of two shared coiled-coil subunits (SMC2 and SMC4) and three isoform-specific subunits (the kleisin CAP-H or CAP-H2 and two HAWK proteins CAP-D2/3 and CAP-G/2, Hirano and Mitchison, 1994; Hirano et al., 1997), have been shown to be capable of processive DNA loop extrusion (Ganji et al., 2018). Both Condensin I, activated through mitotic phosphorylation and KIF4A (Kimura et al., 1998; Bazile et al., 2010; Tane et al., 2022; Cutts et al., 2024, Preprint), and Condensin II, deactivated during interphase by MCPH1 (Houlard et al., 2021) and associated with chromosomes through M18BP1 in mitosis (Borsellini et al., 2024, Preprint), localize to the longitudinal axis of mitotic chromosomes (Ono et al., 2003; Hirota et al., 2004). Condensin I and II impact the shape of mitotic chromosomes distinctly, with Condensin II compacting chromosomes axially from prophase onward, and Condensin I compacting chromosomes laterally once it gains access to DNA during prometaphase (Ono et al., 2003, 2004; Hirota et al., 2004; Shintomi and Hirano, 2011; Green et al., 2012). Through their sequential action, the Condensins shape mitotic chromosomes into rod-shaped entities and provide mechanical rigidity (Houlard et al., 2015) to ensure the faithful segregation of sister chromatids by spindle forces. Based on quantitative and super-resolution imaging, as well as HiC and polymer modeling, it has recently been proposed that Condensins organize mitotic chromosomes into nested loops, with the less abundant and stably binding Condensin II extruding big DNA loops (~450 kb) already during prophase that are subsequently nested into smaller sub-loops (~90 kb) by the more abundant and more dynamically associated Condensin I complex after nuclear envelope breakdown (Walther et al., 2018; Gibcus et al., 2018). These Condensin-driven loops are randomly generated across the linear chromosomal DNA molecules, thereby erasing sequence-specific interphase structures (Naumova et al., 2013).

In interphase, the two closely related Cohesin complexes Cohesin-STAG1 and Cohesin-STAG2 govern the loop extruder-based genome organization (Wutz et al., 2017, 2020). Like the Condensins, the Cohesins are ring-like protein complexes

[1]Cell Biology and Biophysics Unit, European Molecular Biology Laboratory (EMBL), Heidelberg, Germany; [2]Collaboration for Joint PhD Degree Between EMBL and Heidelberg University, Faculty of Biosciences, Heidelberg, Germany; [3]Centre for Cancer Immunology, University of Southampton, Southampton, UK; [4]Advanced Light Microscopy Facility, European Molecular Biology Laboratory (EMBL), Heidelberg, Germany; [5]Research Institute of Molecular Pathology, Vienna BioCenter, Vienna, Austria; [6]Department of Biomedical Laboratory Science, Norwegian University of Science and Technology (NTNU), Trondheim, Norway; [7]Science for Life Laboratory (SciLifeLab), Solna, Sweden.

Correspondence to Jan Ellenberg: jan.ellenberg@embl.de.

consisting of two shared coiled-coil subunits (SMC1 and SMC3), a shared kleisin subunit (RAD21, also called SCC1), and one isoform-specific HEAT-repeat subunit (STAG1 or STAG2, Losada et al., 1998, 2000; Sumara et al., 2000). In the presence of the accessory HEAT repeat protein NIPBL, Cohesin complexes can extrude DNA loops (Kim et al., 2019; Davidson et al., 2019) until they are stalled by the protein CTCF binding to the conserved essential surface of STAG1/2 (Li et al., 2020). CTCF is a zinc-finger–containing protein that is enriched at its asymmetric cognate binding sites in the genome (de Wit et al., 2015; Guo et al., 2015), yielding the most efficient stalling of loop extruding Cohesin when arranged in a convergent orientation (Rao et al., 2014). The protein WAPL functions as an unloader of Cohesin on chromatin, restricting its maximal residence time on chromatin and thereby achieving a constant turnover of DNA loops (Kueng et al., 2006; Wutz et al., 2017). The combined action of these proteins leads to the continuous and dynamic generation of sequence-specifically positioned DNA loops in the genome (Rao et al., 2014; Sanborn et al., 2015; Fudenberg et al., 2016; Gabriele et al., 2022; Mach et al., 2022; Beckwith et al., 2023, Preprint), thereby creating more compact domains in the genome termed topologically associated domains (TADs, Nora et al., 2012; Dixon et al., 2012). While their functional role is still an active area of research, TADs have been implicated in the regulation of gene expression through the active regulation of enhancer–promoter contact frequency (Lupiáñez et al., 2015; Flavahan et al., 2019).

Similar to the two Condensin isoforms, the Cohesin isoforms STAG1/2 display different expression levels and chromatin residence times in HeLa cells, with Cohesin-STAG1 being the less abundant subunit with a long residence time, and Cohesin-STAG2 making up 75% of the total Cohesin pool and being more dynamically bound to chromatin (Losada et al., 2000; Holzmann et al., 2019; Wutz et al., 2020). While the two isoforms share a large portion of common binding sites in the genome and display a certain functional redundancy in the generation of DNA loops, Cohesin-STAG1/2 also have unique binding sites, with Cohesin-STAG1 being preferentially enriched at CTCF binding sites and TAD boundaries, and Cohesin-STAG2 being enriched at non-CTCF sites (Kojic et al., 2018).

While the bona fide interphase organization and the formation of mitotic chromosomes have been subject to thorough investigation, much less is known about how the interphase organization is rebuilt after mitosis. Previously, the genome-wide reorganization of chromatin has been studied using a combination of HiC and ChIP-seq in cell populations fixed after pharmacological synchronization in a long mitotic arrest. This revealed a slow and gradual transition of the mitotic to the interphase fold over the course of several hours, via an apparently unstructured folding intermediate during telophase that is devoid of Condensin and Cohesin loop extruders, as well as a gradual build-up of TAD structures over the course of several hours during G1 (Abramo et al., 2019; Zhang et al., 2019).

Here, we set out to systematically quantify and map the actions of the Condensin and Cohesin loop extrusion machinery during mitotic exit in single living cells, aiming to characterize the dynamic molecular processes underlying the reformation of the loop-extrusion governed interphase genome organization

after mitosis. We find that the switch from mitotic to interphase organization takes about 2 h in unsynchronized cells, passing a transition state during telophase during which a minimal set of three Condensins and Cohesins each are simultaneously bound per megabase of genomic DNA. We found that Cohesin-STAG1 is rapidly imported into the newly formed daughter nuclei alongside CTCF, capable of the formation of large TAD-scale loops early after mitosis as a monomer. We found that Cohesin-STAG2 likely also extrudes DNA loops as a monomer, but that it undergoes a concentration-dependent dimerization on chromatin upon its full import into the nucleus. Based on our quantitative imaging data, we can infer that this phenomenon is a result of the high occupancy of eight chromatin-bound Cohesin-STAG2 per megabase in late G1, leading to frequent encounters of neighboring complexes that lead to a nested/stacked arrangement of extruded loops. Surprisingly, we also found that CTCF is increasingly stabilized on chromatin throughout G1 due to its increasing interaction with the two Cohesin complexes. Based on these data, we propose a double-hierarchical loop model to generate interphase genome architecture after mitosis, in which the two interphase Cohesin loop extruders sequentially build a nested arrangement of large and then small DNA loops, conceptually similar to how Condensins have been suggested to drive mitotic genomic organization.

## Results

### The transition from mitotic to interphase loop extruders occurs over 2 h after mitosis and requires nuclear import

To examine the time required to complete the switch from mitotic to interphase loop extruder genome organization (Fig. 1 A), we made use of human HeLa Kyoto (HK) homozygous knock-in cell lines in which all alleles of the endogenous genes for the kleisin subunits of Condensin I (NCAPH), Condensin II (NCAPH2), and the HEAT-repeat subunits of Cohesin-STAG1 (STAG1) and Cohesin-STAG2 (STAG2) have been tagged with GFP (Walther et al., 2018; Cai et al., 2018). After a single S-phase synchronization, we performed continuous FCS-calibrated 4D live-cell imaging (Politi et al., 2018, Fig. 1 B and Fig. S1 A) through two subsequent cell divisions with a 10-min time-resolution using SiR-Hoechst and extracellular Dextran to label nuclear and cell volumes, respectively (Fig. 1 C). Computational 3D segmentation of these cellular landmarks (Cai et al., 2018, Fig. S1 B), combined with automatic cell tracking, allowed us to align single-cell trajectories from one anaphase to the next and calculate absolute protein concentrations and copy numbers throughout a full cell cycle (Fig. 1 D and Fig. S1, C–G).

As expected, both Condensin isoforms were concentrated on mitotic chromosomes and Condensin II maintained a stable nuclear concentration after division (Fig. 1 D). Surprisingly, we found that while the high Condensin I concentration of 380 nM on chromosomes dropped sharply after segregation, it did not become completely cytoplasmic but maintained a concentration of 150 nM in the two newly formed interphase nuclei, where it then became diluted slowly with nuclear growth (Fig. 1 D). Photobleaching of this nuclear Condensin I pool in interphase

Figure 1. **FCS-calibrated imaging of Cohesin isoforms shows that reorganization of loop extrusion during mitotic exit takes about 2 h after anaphase onset. (A)** Schematic of current loop-extrusion–based models of mitotic and interphase genome organization. Condensin I and II build nested mitotic loops. Cohesins build topologically associating domains delimited by the boundary factor CTCF during interphase. **(B)** Fluorescence intensity calibration using fluorescence correlation spectroscopy (FCS). Fluorescence intensity and photon count fluctuation measurements are performed in cells expressing varying amounts of monomeric EGFP. An autocorrelation function simulating particle diffusion through the effective detection volume is fit to the autocorrelated photon count signal, enabling estimation of protein number in the effective detection volume and therefore the calibration of fluorescence intensities to absolute protein concentrations (Politi et al., 2018). **(C)** Imaging of genome-edited HK cells with homozygously EGFP-tagged Cohesin-STAG2 throughout two consecutive mitoses. Fluorescence intensities (FI) measured in the second mitosis were set to the average protein concentration (C[nM]) during metaphase as measured by FCS-calibrated imaging of the same cell line during mitotic exit imaging (see E). Protein concentrations of all other time points were adjusted based on relative differences in measured fluorescence intensities. **(D)** Nuclear concentration throughout an entire cell cycle ranging from one anaphase to the next displayed for genome-edited HK cells with homozygously (m)EGFP-tagged Cohesin-STAG1 (n = 20 cells) and Cohesin-STAG2 (n = 13 cells), as well as Condensin I (NCPAH, n = 8 cells) and Condensin II (NCAPH2, n = 21 cells). Inset shows the focus of imaging on the first 2 h after mitosis performed with higher temporal resolution. Error bands represent a 95% confidence interval. **(E)** FCS-calibrated imaging of genome-edited HK cells with homozygously EGFP-tagged Cohesin-STAG2 throughout mitotic exit. A total of 75 3D stacks with 2-min intervals is triggered after successful automated identification of metaphase cells based on SiR-Hoechst staining. Fluorescence intensity calibration by FCS allows for the conversion of measured fluorescence intensities (FI) to absolute protein concentrations and protein numbers (N) per unit volume. **(F)** Absolute protein numbers co-localizing with chromatin/the two daughter nuclei displayed for genome-edited HK cells with homozygously (m)EGFP-tagged Cohesin-STAG2 (n = 11 cells) and Condensin I (NCPAH, n = 14 cells). Reformation and full establishment of the nuclear envelope as determined by Lamin B receptor (Fig. S1 P) is indicated through grey background. Error bands represent 95% confidence interval.

revealed that it moves freely in the nucleus and does not exchange with the cytosolic pool (Fig. S1 H). Quantitative full cell cycle imaging showed that the nuclear pools of NCAPH/2 could in principle form complete Condensin complexes with the shared Condensin subunit SMC4, which is present inside the nucleus in sufficient numbers (Fig. S1 E).

Conversely to the sharp reduction in chromosomal Condensin I, the two Cohesin isoforms STAG1/2 that are key for interphase genome organization became enriched inside the nucleus after anaphase and reached essentially constant nuclear concentrations throughout the entire interphase (Fig. 1 D). This was also found to be true for CTCF (Fig. S1 E), revealing homeostatic stable nuclear concentrations of these factors and no doubling of interphase loop extruders with DNA replication, consistent with previous reports that showed uncoupling of nuclear growth from DNA replication (Otsuka et al., 2016). We found that the Cohesin isoform STAG1 only makes up 25% of the total Cohesin pool, consistent with previous studies (Fig. 1 D, Losada et al., 2000; Holzmann et al., 2019). Interestingly, Cohesin-STAG1 displayed rapid and complete nuclear localization shortly after mitosis, followed by the equilibration of its nuclear concentration upon nuclear expansion (Fig. 1 D and Fig. S1 D). The threefold more abundant Cohesin-STAG2, however, reached stable nuclear concentrations only about 3 h after mitosis in S-phase arrested cells (Fig. 1 D). Importantly, control experiments of asynchronous cells revealed that S-phase arrest resulted in increased cell volumes and protein numbers, delaying the complete import of Cohesin-STAG2 (Fig. S1, I–N). FCS-calibrated 4D live-cell imaging showed that nuclear import of Cohesin-STAG2 requires 2 h in asynchronous cells (Fig. S1, M and N).

Having characterized the chromosomal/nuclear concentration changes of the four loop extruders and CTCF throughout the cell cycle, we next focused our analysis on the transition between Condensin and Cohesin occupancy on the genome during the first 2 h after mitosis in asynchronous cells, increasing the time-resolution of our FCS-calibrated 4D imaging to 2 min (Fig. 1 E and Fig. S1 O). This detailed kinetic analysis revealed that the number of Condensin I proteins associating with chromatin rapidly dropped after its peak during anaphase (Fig. 1 F). This drop, however, ceased at the time of reformation of the nuclear envelope, 6 min after AO (Fig. S1 P), that creates a permeability barrier and apparently retains the remaining Condensin I molecules inside the newly formed nucleus. Cohesin-STAG2 started its nuclear enrichment precisely from the time of nuclear envelope assembly and required 2 h until complete nuclear enrichment (Fig. 1 F and Fig. S1 N). To test if the Cohesin accumulation required nuclear import, we acutely degraded degron knocked-in Nup153, an essential component of nuclear pore complexes (Fig. 2, A–E). We found that both Cohesin and CTCF levels inside the nucleus were significantly reduced in Nup153-depleted cells early after mitosis (Fig. 2, F and G), showing that both factors require functional nuclear pores to reach the genome.

## Condensins and Cohesins bind simultaneously, yet independently, to the early G1 genome at three complexes per megabase DNA

Given that a significant number of both Condensin complexes are still present inside the newly formed daughter cell nuclei when Cohesins start to be imported (Fig. 1 F and Table 1), we wanted to go beyond nuclear concentration and protein numbers and ask how many of the mitotic and interphase loop extruding complexes are bound to chromatin after mitosis and could thus be actively engaged in extrusion. To quantify binding, we used fluorescence recovery after photobleaching (FRAP, Fig. S2 A) of Condensin I and II on the metaphase plate and in the newly formed nucleus. Half nuclear photobleaching indicated that a significant fraction of Condensins remains chromatin-bound in early G1 (Fig. S2, B–D), while at the same time, a large fraction of the newly imported Cohesins are already bound (Fig. S2 F). To assay changes in the chromatin-bound fraction of Condensins and Cohesins quantitatively and in a highly time-resolved manner during mitotic exit, we used a rapid spot-bleach assay monitoring fluorescence depletion from a femtoliter-sized chromatin volume during a 30-s continuous illumination with a diffraction-limited focused laser beam (Fig. 3 A and Fig. S2, G–I). In this assay, the chromatin-bound protein fraction is bleached, while the unbound fraction recovers from the excess soluble nuclear pool outside the small bleach spot. This approach thus provides a rapid measure for the bound fraction of GFP-tagged proteins on chromatin that can be carried out repeatedly in a single living cell without interfering with mitotic progression (Tables 1 and 2).

We then used this assay to monitor changes in the chromatin binding of Condensin and Cohesin every 5 min after exit from mitosis. We found that while all Condensins (using the isoform-shared subunit SMC4-mEGFP) progressively dissociated from chromatin during telophase and early G1, they retained a significant chromatin bound fraction of around 25% 15 min after AO (Fig. 3 B). This reduction in bound fraction, also following nuclear envelope reformation, was consistent for both Condensin isoforms, as shown by time-resolved spot bleaching using isoform-specific NCAPH and NCAPH2 subunits (Fig. S2 K). By contrast, we found that the fraction of bound Cohesins (using isoform shared subunit RAD21-EGFP) increases continuously following nuclear envelope reformation (Fig. 3 B), reaching a bound fraction of about 40% 15 min after AO. Again, this increase in binding was consistent for both Cohesin isoforms (using isoform-specific STAG1/2 subunits (Fig. S2 K).

Our quantitative real-time analysis of chromatin binding in single dividing cells provides clear evidence for co-occupancy of chromatin by Condensin and Cohesin complexes throughout telophase and early G1. Combining the bound fraction measurements by FRAP with the protein numbers measured by FCS-calibrated imaging (e.g., Fig. 1 F and Fig. S1 Q) allows us to calculate the number of proteins bound to genomic DNA (Fig. 3 C and Fig. S2 L). This analysis shows that in early G1, 15 min after AO, the same number of around three Condensin and Cohesin complexes are simultaneously bound per megabase of genomic DNA (Fig. 3 C). Could this simultaneous binding of mitotic and interphase loop extruders be functionally interlinked, similar to Cohesin's eviction from chromatin by Condensin during prophase (Hirota et al., 2004; Samejima et al., 2024, Preprint)? To test this, we probed if the chromatin localization of Condensins and Cohesins in early G1 depends on each other's presence using AID-degron knock-in cell lines for the isoform-shared Condensin subunit SMC4 (Schneider et al., 2022) and the Cohesin-

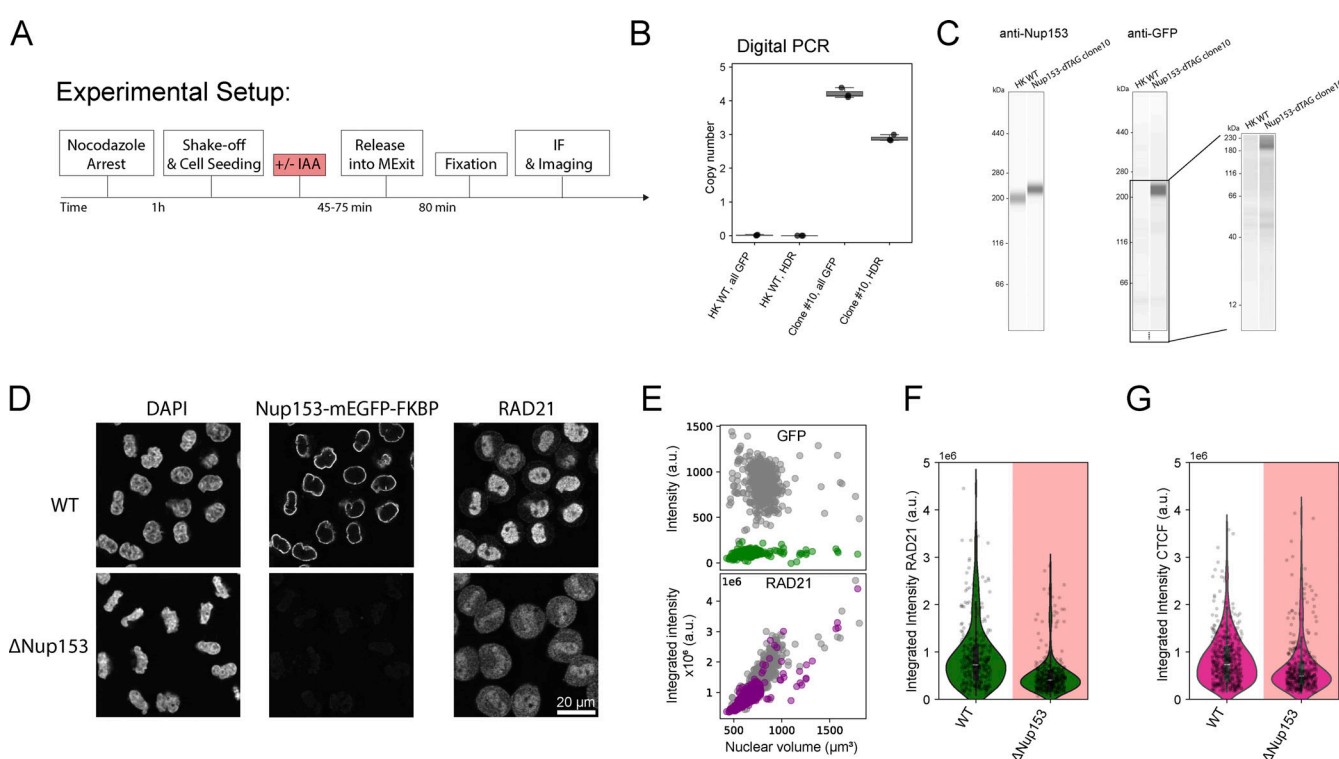

Figure 2. **Functional nuclear pores are required for nuclear import of Cohesin and CTCF. (A)** Experiment scheme for mitotic synchronization of genome-edited HK cells with homozygously mEGFP-FKBP12$^{F36V}$ tagged Nup153, followed by targeted protein degradation of Nup153, release into mitotic exit, and timed fixation during early G1. Cells are immunostained for RAD21 or CTCF and for diffraction-limited imaging. **(B and C)** Validation data for the generation of the homozygous Nup153-mEGFP-FKBP12$^{F36V}$ knock-in line, clone #10. **(B)** Digital PCR was performed to assess the copy number of GFP inserted into the HK genome and to assess the number of successful homology-directed repair (HDR) events at the Nup153 endogenous gene locus. While one copy of mEGFP-FKBP12$^{F36V}$ was inserted elsewhere in the genome, this copy is not expressed (C). Error bars represent minimum and maximum values of the three measured replicates. **(C)** Simple Western analysis of HK WT cells and the Nup153-mEGFP-FKBP12$^{F36V}$ #C10 cell line created and used within this study. The anti-GFP western blot shows no expression of free GFP. The anti-Nup153 western blot shows a clear shift of the Nup153 band, indicating successful and homozygous gene editing. **(D)** Fluorescence micrographs of early G1 cells (∼45 min past mitosis) stained with DAPI in WT or ΔNup153 condition. **(E)** Average fluorescence intensity plots per 3D-segmented nucleus in grey (WT) or colored (ΔNup153). ΔNup153 nuclei do not expand in size, show no residual Nup153 intensity, and show a clear reduction in RAD21 intensity inside the nuclear lumen. **(F)** Average fluorescence intensity of early G1 cells in WT (n = 539) or ΔNup153 (n = 463) condition stained for RAD21. 33–48% reduction in average fluorescence intensity after 45 min release time. Changes above/below 20% are considered a significant change. **(G)** Average CTCF fluorescence intensity upon immunostaining in early G1 cells in WT condition (n = 465) or after mitotic depletion of Nup153 (n = 400). ΔNup153 cells show a 25–40% reduction in average CTCF fluorescent intensity after 45 min release time. Changes above/below 20% are considered a significant change. Source data are available for this figure: SourceData F2.

chromatin-loader NIPBL (Mitter et al., 2020). In these cells, we could acutely degrade the degron-tagged proteins during mitosis (Fig. S2 J) and ask if they are required for the other complex to associate with chromatin by subsequent immunofluorescence staining for the non-degraded Condensin or Cohesin complex. This analysis did not show major differences in chromatin association of Condensin after NIPBL or Cohesin after SMC4 depletion, respectively (Fig. 3, D and E). While we cannot exclude a functional interaction of Condensins and Cohesins beyond chromatin binding, this result suggested that mitotic and interphase loop extruders bind to chromatin simultaneously but independently of one another during early G1.

### Cohesin-STAG1 and CTCF are simultaneously imported immediately after mitosis and sufficient to build the first interphase hallmarks in genome structure

Our full cell cycle data showed clear differences in the time required for complete nuclear import of the two Cohesin isoforms, with STAG1 reaching maximal nuclear concentration within only 10 min, while STAG2 reached steady state only after over 2 h (Fig. 1 D and Fig. S1 N). To get a first insight into which complex might functionally be more important for early G1 genome architecture, we compared these kinetics with the boundary factor CTCF using an endogenous CTCF-EGFP knock-in cell line (Cai et al., 2018). Calibrated full-cell cycle imaging showed a strikingly similar kinetic signature of its nuclear concentration changes compared with Cohesin-STAG1, reaching a ∼2.5 times higher steady-state concentration in interphase (Fig. S1 E). We therefore compared the nuclear import kinetics of Cohesin-STAG1 and CTCF relative to the slower accumulating Cohesin-STAG2 with high time-resolution after mitotic exit using our FCS-calibrated 4D imaging setup. Strikingly, we found that CTCF displayed indistinguishable import kinetics as Cohesin-STAG1 while Cohesin-STAG2 was imported at a much lower rate (Fig. 4 A and Fig. S3 A).

Table 1. **The absolute amount of Condensins, Cohesins, and CTCF inside the nucleus and bound to DNA, as well as their dynamic residence time during early G1**

| POI | eG1 (= 20 min past AO) | | | | | | | | |
|---|---|---|---|---|---|---|---|---|---|
| | Nuclear protein, relative to 2 h post AO | Absolute number in nucleus | % chromatin-bound (spot-bleach, average) | % bound (FRAP) | Absolute number, chromatin-bound | Mean residence time, dynamically bound pool (s) | Absolute number, chromatin-bound, per Mb | Fraction long term bound (%) | Absolute number, long term bound, per Mb |
| SMC4 | 104% | 178,281 ± 20,504 | 21 ± 9 | 19 ± 4 | 33,873 ± 7,661 | 28 ± 13 | 2.14 ± 0.48 | 3 ± 2.20 | 0.34 ± 0.25 |
| NCAPH | 103% | 127,174 ± 14,950 | 15 ± 11 | 11 ± 3 | 13,989 ± 4,155 | 45 ± 27 | 0.89 ± 0.26 | 1 ± 1.90 | 0.08 ± 0.15 |
| NCAPH2 | 101% | 25,430 ± 6,712 | 22 ± 9 | Optimal parameters for fitting not found | | Optimal parameters for fitting not found | | | |
| CTCF | 102% | 125,460 ± 21,359 | 50 ± 19 | 62 ± 8 | 77,785 ± 16,616 | 125 ± 35 | 4.92 ± 1.05 | 20 ± 5.30 | 1.59 ± 0.50 |
| RAD21 | 63% | 151,268 ± 39,019 | 41 ± 10 | 53 ± 7 | 80,172 ± 23,233 | 155 ± 32 | 5.07 ± 1.47 | 15 ± 3.90 | 1.44 ± 0.53 |
| STAG1 | 100% | 53,480 ± 13,770 | 37 ± 9 | 63 ± 3 | 33,692 ± 8,864 | 244 ± 42 | 2.13 ± 0.56 | 17 ± 3.60 | 0.57 ± 0.19 |
| STAG2 | 56% | 102,150 ± 20,097 | 37 ± 9 | 47 ± 5 | 48,010 ± 10,738 | 108 ± 12 | 3.04 ± 0.68 | 3 ± 2.90 | 0.19 ± 0.19 |

Estimated mean values and standard deviation are displayed. For proteins (stably) bound per megabase DNA, propagated standard deviation of mean nuclear protein numbers and mean FRAP bound fractions is provided. Calculations of per-megabase-DNA counts are based on the 7.9 Gb HeLa Kyoto genome reported by Landry et al. (2013).

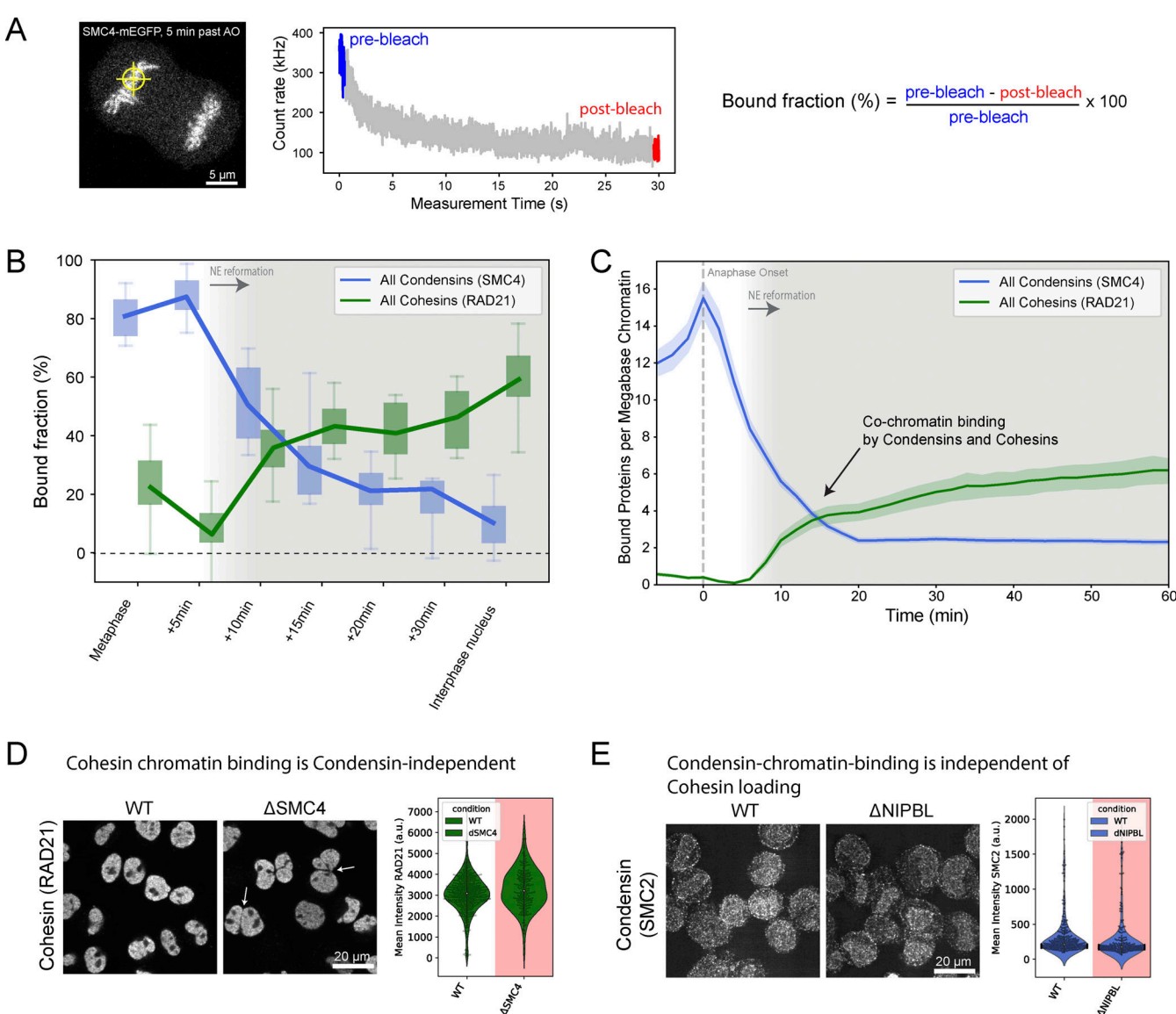

Figure 3. **Condensins and Cohesins co-occupy chromatin during telophase and early G1, as revealed by time-resolved bleaching. (A)** Illustration of the spot-bleach assay. Genome-edited HK cells homozygously expressing (m) EGFP-tagged Condensin and Cohesin subunits are illuminated at a single spot on chromatin for a total duration of 30 s and the resulting fluorescence intensity is continuously measured. The chromatin-bound fraction of a given protein of interest is calculated based on the mean fluorescence intensity of the first and last 500 ms. Exemplary image and bleach data are shown for the common Condensin subunit SMC4. **(B)** The fraction of chromatin-bound Condensins (SMC4) and Cohesins (RAD21) determined using the spot-bleach assay at different timepoints during mitotic exit. Every bar plot represents at least 10 individual datapoints measured in 10 separate cells. **(C)** Absolute number of proteins bound to chromatin was determined by multiplication of chromatin-bound fractions shown in B with absolute protein numbers colocalizing with chromatin ($n$[SMC4] = 21 cells, $n$[RAD21] = 18 cells) as determined in Fig. 1, E and F and displayed as per-megabase-count assuming an equal distribution of the proteins on the entire 7.9 Mb HeLa genome (Landry et al., 2013). Grey background indicates the reformation of the nuclear envelope. Error bands represent 95% confidence interval. **(D)** Fluorescence micrographs and quantification of early G1 cells in WT condition ($n$ = 496) or after degradation of the isoform-shared Condensin subunit SMC4 ($n$ = 278). Cells were pre-extracted for 1 min prior to fixation and were stained for RAD21. SMC4 depletion caused a delay in cell division as well as major cell division errors (see merged daughter nuclei in fluorescence micrograph indicated by arrow). Time of release from Nocodazole block had to be increased to 60–70 min to fix cells in early G1 stage. Difference in mean fluorescence intensity: 8–12.5%. Changes above 20% are considered a significant change. **(E)** Fluorescence micrographs and quantification of early G1 cells in WT condition ($n$ = 307) or after degradation of the Cohesin loader NIPBL ($n$ = 272). Cells were pre-extracted for 1 min before fixation and were stained for SMC2. The difference in mean fluorescence intensity: ∼15%. Changes above 20% are considered a significant change.

The simultaneous import of Cohesin STAG1 and CTCF is consistent with a functional interaction on chromatin immediately after nuclear reformation. We performed quantitative immunofluorescence after pre-extraction of soluble proteins and found that Cohesin-STAG1 and CTCF indeed bound chromatin already early in G1, while the majority of Cohesin-STAG2 bound chromatin only later in G1 (Fig. S3, B–F). In addition, we performed real-time spot-bleach, as well as FRAP measurements of CTCF, to compare its binding to chromatin with Cohesin STAG1 early after mitotic exit (Fig. S2, E and L).

**Table 2.** The absolute amount of Condensins, Cohesins, and CTCF inside the nucleus and bound to DNA, as well as their dynamic residence time during G1

| POI | G1 (= 2–4 h past AO) | | | | | | | | | |
|---|---|---|---|---|---|---|---|---|---|---|
| | Nuclear protein, relative to 2 h post AO | Absolute number in nucleus | % chromatin-bound (spot-bleach, average) | % bound (FRAP) | Absolute number, chromatin-bound | Mean residence time, dynamically bound pool (s) | Absolute number, chromatin-bound, per Mb | Fraction long term bound (%) | Absolute number, long term bound, per Mb |
| SMC4 | 100% | 174,165 ± 24,018 | 10.0 ± 8.5 | | | | | | |
| NCAPH | 100% | 124,990 ± 18,133 | 9 ± 14 | | | | | | |
| NCAPH2 | 100% | 25,597 ± 7,482 | 2.0 ± 9.4 | | | | | | |
| CTCF | 100% | 124,206 ± 23,317 | 72 ± 26 | 72.0 ± 3.3 | 89,428 ± 17,281 | 139 ± 22 | 5.66 ± 1.09 | 35.00 ± 3.70 | 1.98 ± 0.43 |
| RAD21 | 100% | 238,095 ± 56,897 | 59 ± 12 | 77.0 ± 4.8 | 183,333 ± 45,277 | 172 ± 36 | 11.60 ± 2.86 | 21.00 ± 3.70 | 2.44 ± 0.72 |
| STAG1 | 100% | 53,482 ± 13,535 | 0.59 ± 8 | 76.0 ± 5.9 | 40,646 ± 10,760 | 276 ± 55 | 2.57 ± 0.68 | 31.00 ± 7.50 | 0.79 ± 0.28 |
| STAG2 | 100% | 183,408 ± 33,855 | 50 ± 10 | 71.0 ± 5.3 | 132,054 ± 26,242 | 126 ± 28 | 8.36 ± 1.66 | 8.00 ± 5.00 | 0.67 ± 0.44 |

Estimated mean values and standard deviation are displayed. For proteins (stably) bound per megabase DNA, propagated standard deviation of mean nuclear protein numbers and mean FRAP bound fractions is provided. Calculations of per-megabase-DNA counts are based on the 7.9 Gb HeLa Kyoto genome reported by Landry et al. (2013).

This analysis revealed that when Cohesin-STAG1 and CTCF reach their maximum concentration in early G1, about 2 Cohesin-STAG1 and 5 CTCF molecules are bound per megabase of genomic DNA (Table 1). Two actively extruding Cohesin-STAG1 complexes per megabase of genomic DNA would in principle explain the frequency of compact TAD structures that have been estimated at 1.5 TADs/Mb using biochemical approaches previously (Wutz et al., 2020). To test directly if Cohesin STAG1 without Cohesin STAG2 is indeed sufficient to create the first more compactly folded G1 genome structures in single cells, we took advantage of our recently developed nanoscale DNA tracing method LoopTrace, enabling us to inspect individual 3D DNA folds as well as ensemble averages with precise physical distance measures (Beckwith et al., 2023, *Preprint*, Fig. 4 B and Fig. S3 G). We traced three independent 1.2 megabase long genomic regions predicted to contain TADs, in 3D at 12 kb genomic and 20 nm spatial resolution (Fig. 4 C; and Fig. S3, J and K). Our single-cell DNA traces could indeed readily identify compact 3D DNA folds already in single early G1 cells (Fig. 4 D). Depletion of Cohesin-STAG2 during the prior mitosis (Fig. S3, G and H) did not influence the overall genomic size of these domains, but led to some reduction in internal loop nesting and slight physical decompaction (Fig. 4 E and Fig. S3 I), which was also clear when comparing pairwise physical 3D distance maps of these regions from hundreds of control or STAG2 depleted cells (Fig. 4 F; and Fig. S3, L and M). We conclude that Cohesin-STAG1 and CTCF are imported with identical kinetics rapidly after mitosis and are sufficient to build the first compact looped interphase structures in single G1 cells, equivalent to biochemically detected TADs in cell populations.

### Cohesin-STAG1 and CTCF become increasingly stably bound to the genome throughout G1

To investigate the interplay of Cohesin-STAG1 and Cohesin-STAG2 at later times after mitosis, we performed FRAP measurements during G1 (2–5 h past AO) and compared them to our measurements shortly after mitosis (Fig. 5 A). We found that the chromatin-bound fraction for both Cohesin isoforms as well as CTCF significantly increased in later G1 (Fig. S4 A). A single exponential function with an immobile fraction fit the fluorescence equilibration kinetics of all proteins well (Fig. 5 B) and allowed us to determine the dynamically chromatin-bound protein fraction, its residence time, as well as the stably bound fraction that did not exchange dynamically during our measurement time (Fig. S4, B and C; and Tables 1 and 2). While the average residence time of the dynamically bound pool of Cohesin isoforms (STAG1: 4 min, STAG2: 2 min and CTCF: 2 min) remained unchanged from early to late G1 (Fig. S4 B), we measured a significant increase in the stably chromatin-bound fraction for Cohesin-STAG1 and CTCF, reaching up to 30–40% of the total protein (Fig. 5, C and E; and Fig. S4 C). Cohesin-STAG2 also displayed a significant increase in its stable chromatin-bound fraction, however reaching <10% of the total protein pool (Fig. 5 D).

Of note, our reported chromatin-bound fractions are slightly higher compared to recent studies while our chromatin residence times of Cohesins are shorter (Wutz et al., 2020;

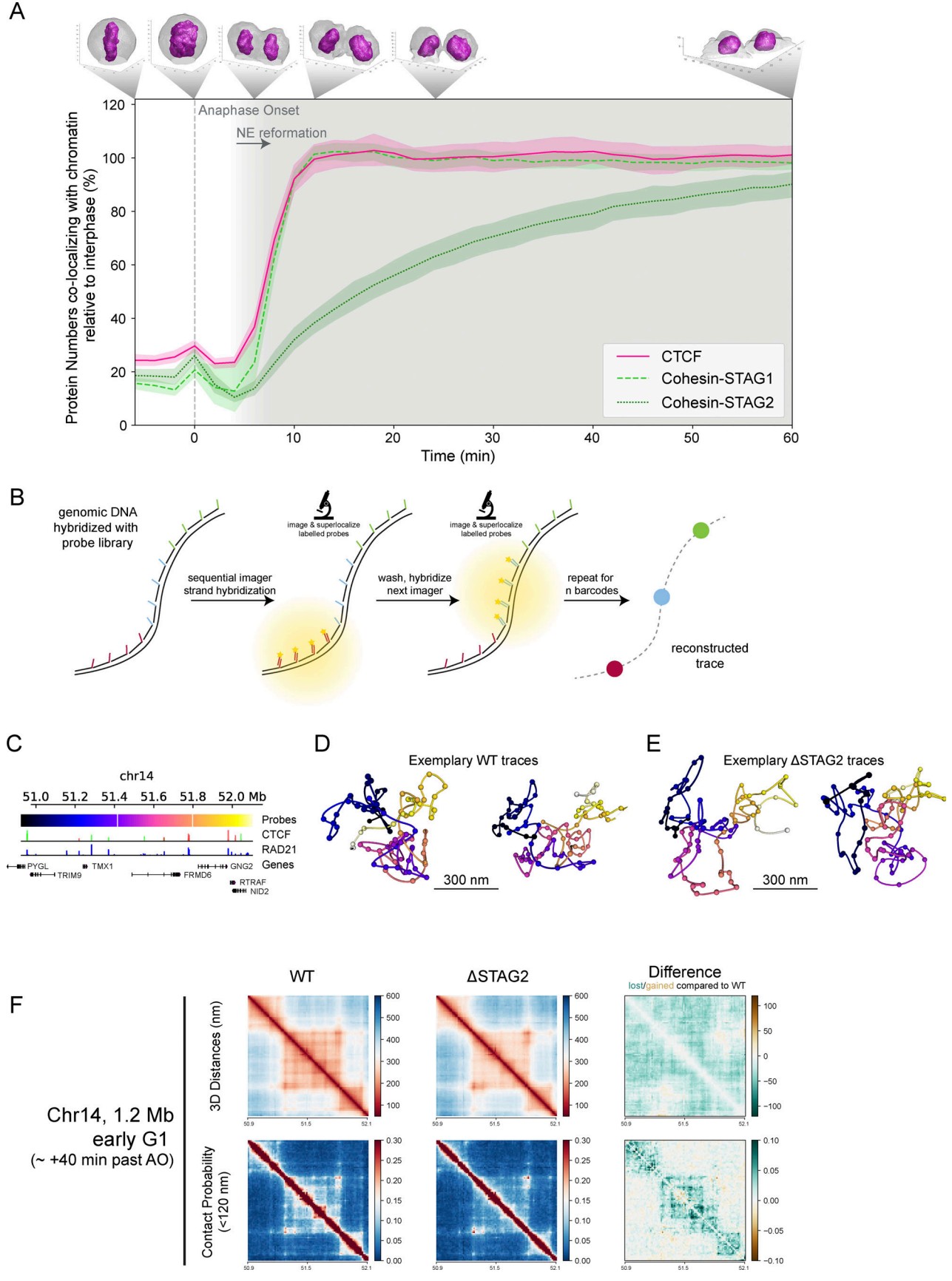

Figure 4. **Cohesin-STAG1 and CTCF cooperate to form interphase TAD structures after mitosis. (A)** FCS-calibrated protein numbers colocalizing with chromatin displayed for genome-edited HK cells with homozygously EGFP-tagged Cohesin-STAG1 (*n* = 25 cells), Cohesin-STAG2 (*n* = 11 cells), and CTCF (*n* = 15

cells) relative to the measurement 2 h after anaphase onset. Error bands represent 95% confidence interval. **(B)** Scheme explaining LoopTrace chromatin tracing workflow. Fixed cells were subjected to single-strand resection via exonuclease treatment (RASER) for maximal structure-preservation and subsequent hybridization with a tiled FISH library. Every FISH probe contains a non-genome-complementary docking handle that can be hybridized with a fluorescently labeled imager strand to read out the 3D location of a genomic locus (Beckwith et al., 2023, *Preprint*). **(C)** Overview of the traced 1.2 megabase locus on chromosome 14 with genes as well as ChIP-seq binding sites for RAD21 and CTCF (from the ENCODE portal [Sloan et al., 2016, https://www.encodeproject.org/] with the following identifiers: ENCFF239FBO [RAD21], ENCFF111RWV [CTCF]; CTCF directionality annotations from Rao et al. [2014]). **(D and E)** Exemplary chromatin traces of WT (D) or ΔSTAG2 (E) early G1 cells. **(F)** Distance and contact matrices of a 1.2 megabase region on chromosome 14 locus traced at a genomic resolution of 12 kb in early G1 cells with and without Cohesin-STAG2. Differences between WT and ΔSTAG2 are highlighted for distance and contact probability maps.

---

Holzmann et al., 2019; Hansen et al., 2017). This is likely due to differences in the experimental setup, as we have (1) ensured a minimal lag time (<30 s) between the FRAP's pre- and post-bleach measurements, thus classifying all proteins with a chromatin residence time above 30 s as "chromatin-bound" and (2) our overall shorter FRAP measurement period of 10 min, which lead to the fact that long-term chromatin bound proteins were classified as "immobile," thus precluding fits with longer residence times.

While it has been previously reported that Cohesin-STAG1 chromatin binding can be stabilized by CTCF (Wutz et al., 2020), whether CTCF's own binding is reciprocally affected by the presence of Cohesin has not been investigated. To test if CTCF's increasingly stable binding in G1 depends on Cohesin, we acutely depleted the isoform-shared subunit RAD21 (Fig. S4, D–G), which resulted in a significant reduction of stably chromatin-bound CTCF, which could be rescued by RAD21 overexpression (Fig. 5 F). This shows that Cohesin is necessary and sufficient to stabilize CTCF's interaction with chromatin in G1.

### Cohesin-STAG2 completes its nuclear import after 2 h and exhibits concentration-dependent co-localization on the genome in G1

To test directly whether the observed interdependent increase in stable binding of both CTCF and Cohesins is due to increased complex formation between these proteins on chromatin, we performed STED super-resolution imaging of CTCF and the Cohesin isoform-specific subunits STAG1 and STAG2 during early and late G1. To achieve high and comparable labeling efficiency of the different Cohesin isoforms, we used our homozygous knock-in cell lines for STAG1/2-EGFP and detected both with the same GFP-nanobody, while using a specific antibody to detect endogenous CTCF as a reference (Fig. 6, A and B; and Fig. S5, A–C). Calculating the number of chromatin complexes from our combined concentration imaging and FRAP data allowed us to estimate the labeling efficiency of our super-resolution imaging by counting the individual labeled fluorescent spots to about 60–70% for the two Cohesin-isoforms (Fig. S5, D–F), very similar to our previous labeling efficiencies of this GFP nanobody (Thevathasan et al., 2019). Given that we could resolve the expected number of Cohesin complexes as individual fluorescent spots, the large majority of the labeled STAG1/2 proteins in early G1 therefore most likely represent monomeric Cohesin complexes.

Colocalization of either STAG1 or STAG2 with CTCF resulted in comparable spatial correlations that in both cases increased

slightly but significantly from early to late G1, indicating an increase in Cohesin–CTCF complex formation for both isoforms (Fig. 6 C). This increased colocalization was not due to the still ongoing accumulation of STAG2 in the nucleus, as shown by image simulations with random protein distributions at realistic densities (Fig. S5, I and J). Our data thus suggest that CTCF associates with both STAG1, which enters the nucleus early, and STAG2, which completes its import later and eventually becomes the more abundant Cohesin isoform. This finding is in line with the fact that also Cohesin-STAG2 becomes more stably bound to chromatin in late G1 (Fig. 5 D).

The ability to detect both Cohesin isoforms at the single complex level with high labeling efficiency in early G1 also put us in a position to use the intensity of Cohesin spots to ask if we can detect multimerization of the Cohesins as the cell cycle progresses, which would be expected with the formation of closely stacked or nested loops. Interestingly, this analysis revealed that while the average spot intensity and total number of spots detected did not change for Cohesin-STAG1 between early and late G1, the STAG2 spot intensity increased about twofold between early and late G1 (Fig. 6 D and Fig. S5 G) and correlated with a twofold drop in the number of STAG2 spots detected compared to our expectation from quantitative live imaging (Fig. S5 F). When it has reached its maximum concentration in late G1, Cohesin-STAG2 thus appears to associate with the genome in pairs of molecules that are no longer resolvable individually by STED microscopy, which has a lateral precision of around 60 nm. With an estimated extension of single Cohesin complexes of 50 nm, they must therefore be very closely adjacent to each other or form dimers to result in single STED spots with doubled intensity.

Why might Cohesin-STAG2 form closely adjacent pairs of complexes only at the end of G1? To test if its self-association is concentration-dependent, we performed partial depletion of degron-tagged Cohesin-STAG2. This indeed shifted the average spot brightness in late G1 back to a value of nanobody monomers (Fig. S5 H), supporting a concentration rather than, for example, a cell cycle–driven dimerization of Cohesin-STAG2 on DNA. In fact, our quantitative imaging data of the increasing numbers of chromatin-bound Cohesins after mitosis provides a quantitative explanation for the concentration-dependent colocalization of chromatin-bound Cohesin-STAG2. During early G1, we found three Cohesin-STAG2 molecules to be on average bound for 120 s per megabase of DNA. Assuming they extrude loops with the estimated rate of 1 kb/s (Kim et al., 2019; Davidson et al., 2019), they would form 120 kb large loops and would thus be relatively unlikely to encounter each other within one megabase (Table 1).

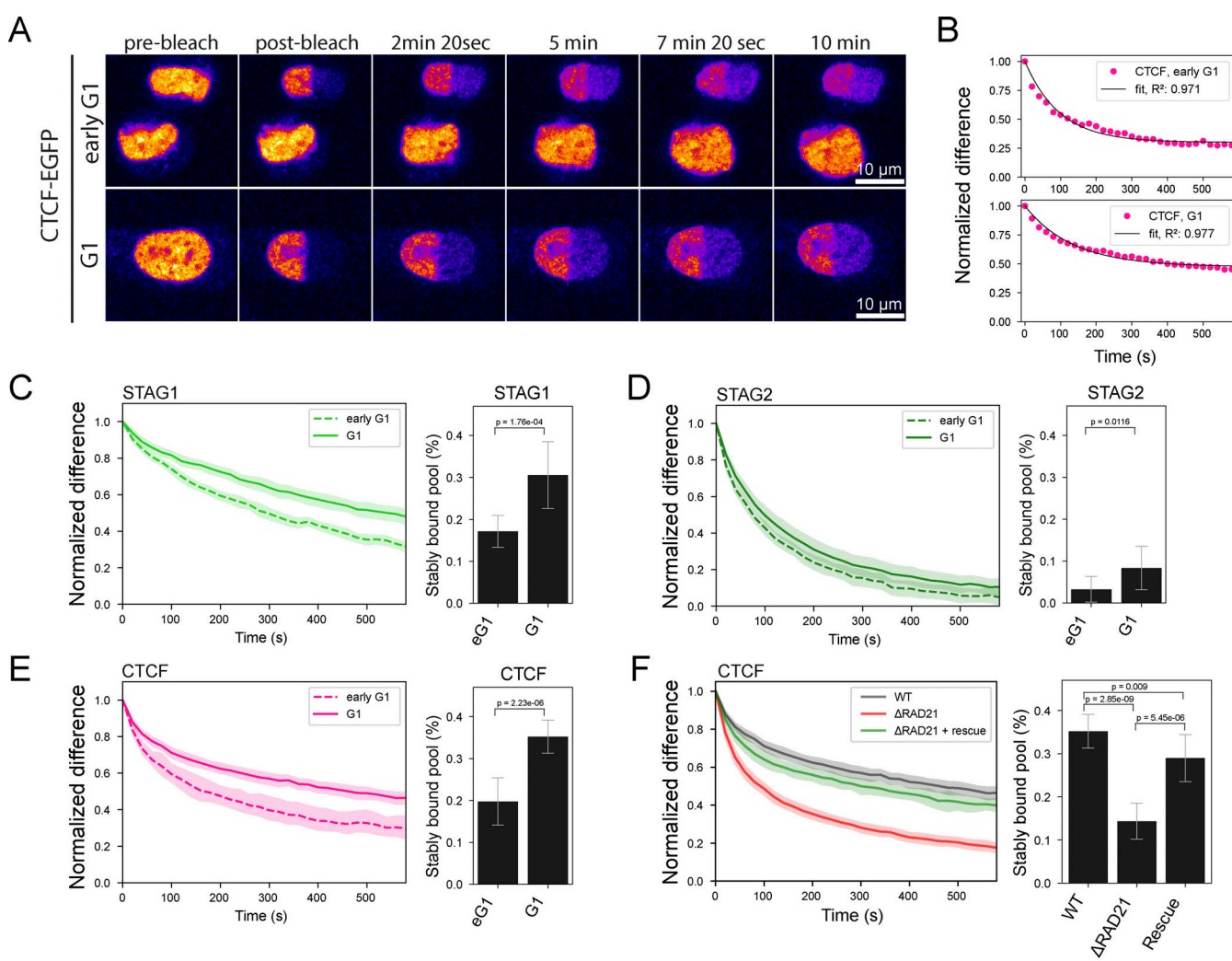

Figure 5. **Fluorescence recovery of Cohesin isoforms and CTCF. (A)** Fluorescence recovery after photobleaching (FRAP) was performed by bleaching half of a nucleus in early G1 cells (20–40 min after anaphase onset) or later G1 cells selected by nuclear volume. **(B)** FRAP shown for genome-edited HK cells with homozygously EGFP-tagged CTCF. The difference between the bleached and unbleached region is normalized by the maximal difference at time t = 0 after bleaching. Black line indicates the data fit by a single-exponential function with an immobile fraction. Single exponential functions with immobile fraction also fit the FRAP recovery of RAD21, STAG1/2 well. **(C–E)** FRAP measurements using homozygous EGFP-knock-in HK cell lines in early G1 and G1 cells, respectively. Bar plots display the mean fraction of protein that is stably bound to chromatin. Two-sample t test was used to calculate significance levels. Error bars show standard deviation. **(C)** Cohesin-STAG1 (early G1: n = 10 cells, G1: n = 9 cells). **(D)** Cohesin-STAG2. (early G1: n = 10 cells, G1: n = 13 cells). **(E)** CTCF. (early G1: n = 9 cells, G1: n = 10 cells). **(F)** FRAP measurements of endogenous CTCF with WT levels of RAD21, after degradation of endogenous RAD21, and after rescue of RAD21 degradation by exogenous RAD21 expression for at least 24 h. Bar plots display the mean fraction of protein that is stably bound to chromatin. Error bars indicate standard deviation (CTCF WT: n = 10 cells, CTCF dRAD21: n = 9 cells, CTCF dRAD21 rescue: n = 10 cells). Two-sample t test was used to calculate significance levels. Data from CTCF-EGFP knock-in line is used as WT reference as it displays WT expression levels of RAD21. The double-knock-in line Rad21-EGFP-AID CTCF-Halo-3xALFA #C7 displayed leaky degradation of RAD21, reducing CTCF-chromatin binding already in -IAA cells (see Fig. S4 G and Materials and methods).

In late G1, however, about eight Cohesin-STAG2 complexes are bound per megabase with a similar residence time (Table 2), making Cohesin-STAG2 encounters between eight 120-kb sized loops within one megabase much more likely. The fact that we observe a quantitative shift in the intensities of the Cohesin-STAG2 spot distribution from early to late G1 (Fig. 6 D) in fact suggests that Cohesin complexes not only encounter each other transiently but potentially stay associated with each other when they meet, which would induce stacking and nesting of loops. To test if such nested and stacked loops indeed form in late G1 in single cells in a

Cohesin-STAG2 dependent manner, we again made use of our nanoscale DNA tracing of interphase cells targeting the same three 1.2 megabase TAD regions as before. The 3D folds of these regions indeed revealed stronger nesting and stronger compaction of these regions compared with early G1 and again showed that this is largely dependent on Cohesin-STAG2 (Fig. 6, E–H and Fig. S5, K–P). In conclusion, due to its continuous nuclear import, Cohesin-STAG2 crosses a critical occupancy threshold on the genome within the first 1–2 h after mitosis that led to a high probability of encounters between Cohesin-STAG2 complexes, accompanied by increased

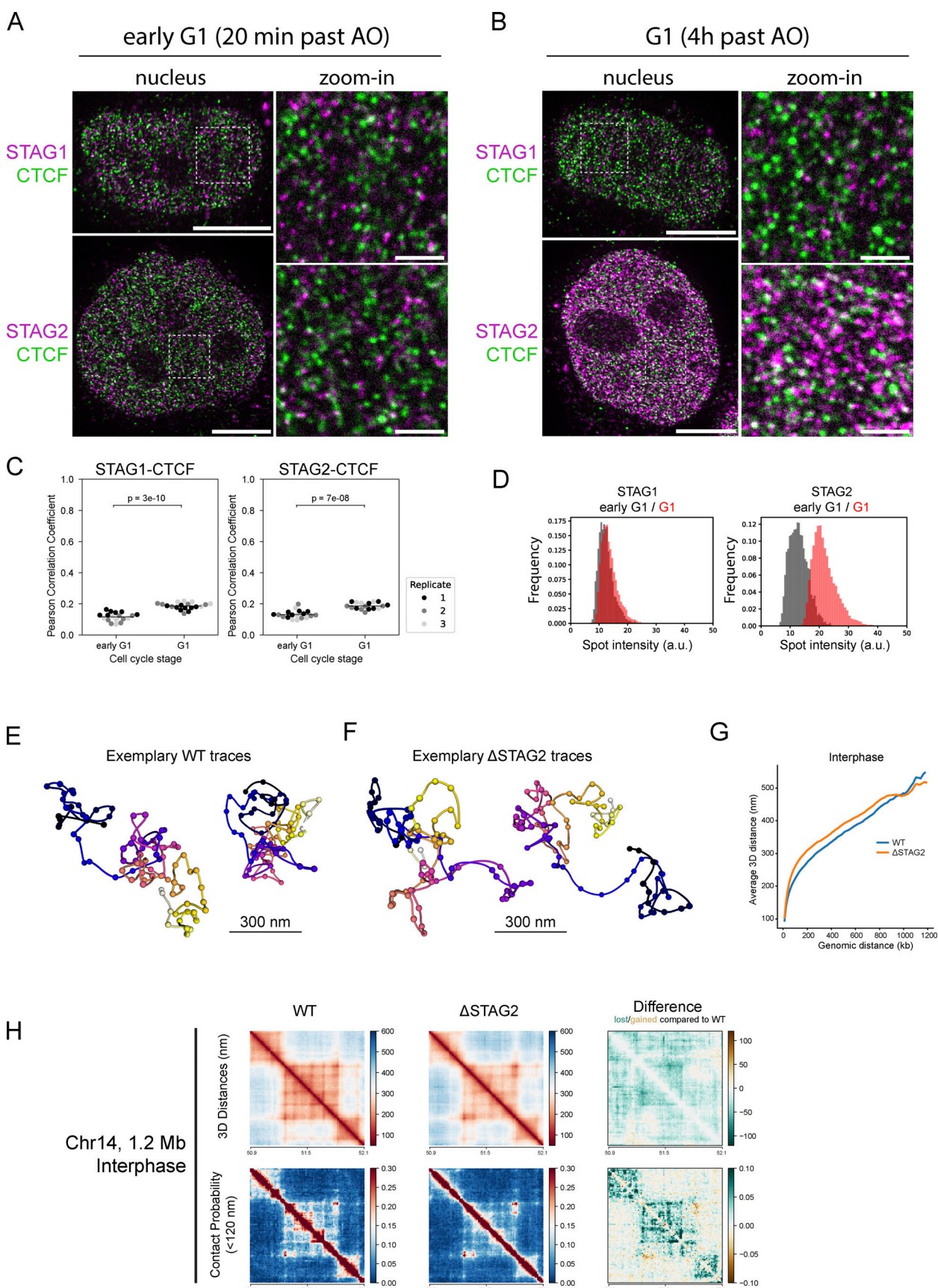

Figure 6. **STED super-resolution imaging of Cohesin isoforms and CTCF. (A and B)** Exemplary STED images showing Cohesin-STAG1/2 (magenta) and CTCF (green) in early G1 (A) and G1 (B) nuclei (scalebar: 5 μm) and zoom-ins (scalebar: 1 μm). **(C)** Colocalization analysis of Cohesin-STAG1/2 with CTCF using

the Pearson Correlation Coefficient of segmented nuclei ($n \geq 17$). Differences in STAG-CTCF colocalization in early G1 compared with G1 are significant as assessed by an independent two-sample *t* test. **(D)** Mean intensity of segmented STAG1/2 spots in STED images of replicate 3. The same results are observed in replicate 1 and 2. Formal significance tests are meaningless due to large sample size. Median intensities of the mean spot intensity distributions are: STAG1 eG1: 12.02, STAG1 G1: 13.06, STAG2 eG1: 12.62, STAG2 G1: 21.02 (arbitrary intensity units). **(E and F)** Exemplary chromatin traces of WT (E) or ΔSTAG2 (F) interphase cells. **(G)** Scaling plot showing the genomic versus Euclidian distance relationship of the three 1.2 Mb regions sampled at 12 kb in WT or ΔSTAG2 interphase cells. Traces from ΔSTAG2 are on average less compact compared with WT. **(H)** Distance and contact matrices of a 1.2 megabase region on chromosome 14 locus traced at a genomic resolution of 12 kb in interphase cells with and without Cohesin-STAG2. Differences between WT and ΔSTAG2 are highlighted for distance and contact probability maps. WT data from 1, ΔSTAG2 data from two independent technical replicates (392 and 610 traces, respectively).

formation of nested loops inside TAD-scale compact domains of the interphase genome.

### A double hierarchical loop model quantitatively explains the transition from mitotic to interphase loop extruder–driven genome organization

Our systematic, quantitative time-resolved mapping of mitotic and interphase loop extruders in single HeLa cells shows that the interphase genome is sequentially organized into compact TAD-scale regions which then compact further by internal stacking of nested loops. This is highly reminiscent of the previously proposed nested loop organization in mitosis (Walther et al., 2018), in which chromosomes are organized by the sequential action of two Condensin complexes (Walther et al., 2018; Gibcus et al., 2018). In this model of establishing mitotic architecture, the less abundant Condensin II loop extrusion motor first forms large DNA loops during prophase that become subsequently nested by the more abundant Condensin I, once it gains access to chromosomes during prometaphase.

Our study now shows that following chromosome segregation and nuclear envelope reformation, some Condensins are still bound to chromatin, while Cohesins and CTCF are rapidly imported into the newly formed nucleus, leading to a co-occupancy of the genome by 3 Condensins and 3 Cohesins per megabase of DNA in early G1. Very interestingly, when the Cohesin interphase loop extruders start binding the genome, they do so independently of Condensins, but like Condensins during mitotic entry also in a sequential manner during mitotic exit. First, the rapid and synchronous nuclear import of Cohesin-STAG1 and CTCF (completed within 10 min after AO) and their immediate chromatin binding at relatively low abundance (three complexes bound per megabase) with a long residence time (4 min) progressively build up the first compact interphase structures even in the absence of Cohesin-STAG2. In a second step, the slowly imported Cohesin-STAG2 (complete only 2 h after mitosis) then binds in higher abundance (eight complexes bound per megabase) and with a shorter residence time (2 min), leading to the generation of many smaller loops (~120 kb), and frequent encounters and likely stalling with neighboring Cohesin-STAG2 complexes, leading to stacking of nested loops inside the larger STAG1 defined domains. In line with a recent study (Wutz et al., 2020), we therefore propose a double hierarchical loop model for the transition from mitotic to interphase loop extruder–driven genome architecture in HeLa cells, in which the Condensin-based, randomly positioned nested loop architecture established during mitotic entry is replaced by a less compact, but conceptually similar Cohesin-driven nested loop architecture, positioned by CTCF, from mitotic exit to early G1 (Fig. 7).

## Discussion

This study provides comprehensive quantitative and time-resolved data on the chromatin-binding of Condensins and Cohesins throughout mitotic exit and G1. In addition to the new data, it furthermore allows us to integrate many previous more qualitative and individual observations into an overall, internally consistent, and quantitative model of how the loop extruder–based genome organization is handed over from mitosis to interphase in HeLa cells.

### A role for chromatin-bound Condensin during telophase?

The Condensin-driven mitotic chromosome organization, previously proposed to be best explained by an axial arrangement of nested DNA loops (Gibcus et al., 2018; Walther et al., 2018), is rapidly lost during telophase when 75% of Condensins unbind DNA. Consistent with chromatin fractionation data from a previous report (Abramo et al., 2019), we find that this removal of Condensins is followed by the import of Cohesin and CTCF into the newly forming nuclei, leading to a co-occupancy of only three Condensins and Cohesins per megabase during early G1. While we show that this is the lowest number of genome-associated loop extrusion complexes at any time during the cell cycle, we conclude that telophase chromatin is not devoid of loop extruders, contrary to what has been concluded recently (Abramo et al., 2019). Although Condensin helps evict chromatin-bound Cohesin during prophase (Hirota et al., 2004; Samejima et al., 2024, Preprint), we did not find evidence for a functional Condensin-Cohesin interaction that impacts their chromatin binding during telophase and early G1 (Fig. 3, D and E). The significant fraction of Condensins that remains chromatin-bound during telophase and early G1 leads to a so far unappreciated pool of Condensin I to be retained in the nucleus during interphase. However, in interphase, nuclear Condensin I is unlikely to be actively engaged in processive loop extrusion due to its mitosis-specific loading onto chromosomes (Hirano et al., 1997) regulated via phosphorylation of the NCAPH N-terminal tail (Tane et al., 2022) and mitotic activation by KIF4A (Cutts et al., 2024, Preprint). It could be that the retained fraction of Condensins has a transient role during telophase and early G1 to facilitate the removal of intrachromosomal catenations as suggested recently (Hildebrand et al., 2024).

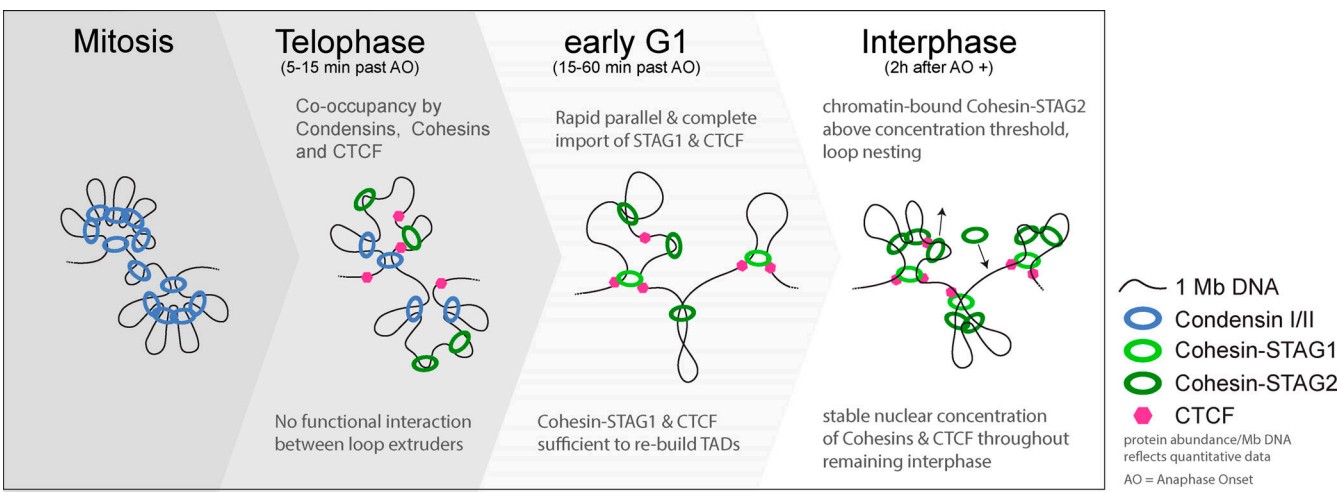

Figure 7. **A new model for the mitosis-to-interphase transition of genome organization by loop extrusion.** Mitotic chromosomes are majorly organized by the abundant Condensin complexes (12/Mb, of those ~1.5 are Condensin II), only a small fraction of CTCF (about 14,000 proteins, ~10% of cellular pool) binds chromatin very transiently and a small fraction of the cellular Cohesin holds together sister chromatids until anaphase. During telophase and early G1, not all Condensins dissociate right away, leading to a co-occupancy of three Condensins and three Cohesins per megabase DNA and a significant pool of Condensin I that remains nuclear until the next mitosis starts. Cohesin-STAG1 and CTCF are fully imported into the newly formed nucleus within 10 min after anaphase onset and are sufficient to build interphase TAD structures in the absence of Cohesin-STAG2. Cohesin-STAG2 only completes its import 2 h after anaphase onset and—due to its abundance on chromatin upon full import (eight complexes per megabase)—frequently crashes into neighboring loop extruders, leading to increased nesting of sub-TAD loops. While Cohesin-STAG1 is stabilized through CTCF and SMC3 acetylation (Wutz et al., 2020), CTCF is increasingly stabilized on chromatin through its association with Cohesin on chromatin. After completed import of Cohesins and CTCF, their nuclear concentrations remain stable irrespective of the DNA content of the nucleus.

### The two Cohesin isoforms bind sequentially and likely have different structural roles

Consistent with previous systems analysis of mitotic protein networks (Cai et al., 2018), we found that Cohesin-STAG1 and CTCF are rapidly imported into the newly formed daughter cell nuclei after cell division and are sufficient in numbers and loop extrusion processivity to form the first interphase TAD-scale loops shortly after mitosis. By contrast, we found that the more abundant second Cohesin isoform STAG2 is imported slowly over the course of 2 h and is dispensable for the generation of these TAD-scale compact structures in early G1 as well as later in interphase. Due to its later binding, higher abundance, and short residence time on chromatin, Cohesin-STAG2 leads to shorter and nested loops within the already established larger Cohesin-STAG1 loops. We speculate that this STAG2-dependent highly dynamic sub-structuring of the more stable TAD loops could promote cell-type-specific intra-TAD contacts between enhancers and promoters independently of CTCF, explaining cell-type dependent effects of Cohesin-STAG2 mutations or depletion (Kojic et al., 2018; Viny et al., 2019). While we found that the overall fraction of chromatin-bound proteins increases for both Cohesin isoforms and CTCF similarly from early to later G1, Cohesin-STAG1 and CTCF became specifically stabilized on chromatin during G1. Our observations are consistent with recent reports that Cohesin-STAG1 is stabilized through CTCF binding and acetylation of its SMC3 subunit by Esco1 (Wutz et al., 2020), suggesting a continued role of Cohesin-STAG1 in the generation and maintenance of long-range loops during interphase that are further sub-structured by Cohesin-STAG2.

### Chromatin binding of CTCF is stabilized by Cohesin

Our FRAP analysis early after mitosis and later during G1 enabled us to see a clear increase in the fraction of stably chromatin-bound CTCF. Interestingly, this stabilization was dependent on the presence on Cohesin and occurred progressively throughout G1 when we found both Cohesin isoforms to increasingly co-localize with CTCF. It is therefore likely that CTCF's stabilization is due to interaction with one or both chromatin-bound Cohesin isoforms, which may serve as additional anchors at CTCF sites. Combined with the fact that Cohesin-STAG1 is preferentially associated with CTCF (Kojic et al., 2018) and is stabilized in part through CTCF (Wutz et al., 2020), our data suggest that Cohesin-STAG1 and CTCF mutually stabilize each other on chromatin, which may be important to stabilize longer lived loop structures in interphase, equivalent of TADs.

### The oligomerization state of Cohesin

Using structured illumination microscopy of the isoform-shared Cohesin subunit RAD21, it was recently reported that the majority of the loop-extruding Cohesin is present in dimers or multimers (Ochs et al., 2024). Consistent with this finding, our STED super-resolution imaging of isoform-specific Cohesin subunits revealed that the less abundant Cohesin-STAG1 is present as a monomer, but that the more abundant isoform Cohesin-STAG2 undergoes dimerization on chromatin later in G1 in a concentration-dependent manner. While we cannot exclude that soluble nuclear Cohesin-STAG2 undergoes concentration-dependent dimerization itself, our quantitative imaging data point to a model according to which

Cohesin-STAG2 colocalizes on chromatin upon mutual encounters: We found that Cohesin-STAG2 is bound to chromatin for 120 s on average and increases its occupancy from three to eight complexes per megabase from early to late G1. If we assume the bound complexes to extrude loops with the reported speeds (i.e., 0.5–2 kb/s, Kim et al., 2019; Davidson et al., 2019), three randomly loaded Cohesin-STAG2 complexes per megabase DNA are very unlikely to encounter each other during early G1 due to the relatively small loops they can make for 2 min (around 120 kb). However, encounters and potential stalling between loops of this size become much more likely when Cohesin-STAG2 is fully imported and present at eight copies per megabase in late G1. While we cannot exclude that Cohesin-STAG2 dimers continue active loop extrusion, our data would be consistent with the view that the default state of Cohesin complexes in loop extrusion is monomeric and that dimers result from encounters and potential stalling events.

### A comprehensive and quantitative dataset to constrain next-generation polymer models

In summary, our systematic and quantitative assessment of Condensin and Cohesin loop extruder dynamics on chromatin in HeLa cells provides a comprehensive and integrated view of the transition from mitotic to interphase genome organization. Given the sequential import of Cohesin isoforms, their chromatin binding dynamics, their different abundance, as well as the impact on chromatin upon depletion, we propose a hierarchical nested loop model for the establishment of the interphase genome organization by the Cohesin loop extruders after mitosis, as has been done previously based on HiC and chromatin binding data of Cohesins during G1 (Wutz et al., 2020). This model is conceptually similar to the hierarchical nested loop architecture proposed for the establishment of Condensin driven mitotic organization (Gibcus et al., 2018; Walther et al., 2018). While the sub-structuring of large Condensin II loops in mitosis by Condensin I serves to laterally compact mitotic chromatin and confers additional mechanical rigidity to chromosomes (Ono et al., 2003; Shintomi and Hirano, 2011; Green et al., 2012; Houlard et al., 2015), we think that the sub-structuring of large STAG1 loops (Kojic et al., 2018; Wutz et al., 2020) by STAG2 aids TAD-scale compaction and specific intra-TAD contact enrichment, potentially in a cell type and species-specific manner (Dixon et al., 2012; Phillips-Cremins et al., 2013; Rao et al., 2014).

The quantitative data and understanding provided by our study are based on our measurements in the HK cell model system, a cancer cell line for which we have established a systematic collection of genome-edited tools to comprehensively analyze SMC complexes quantitatively and perturb them. We note that previous work from our group and others have reported clonal variation of expression in HeLa cells (Walther et al., 2018; Liu et al., 2019). Nevertheless, the relative dynamic changes observed between different proteins during mitotic exit have been very consistent in our hands. In addition, we note that in our studies of mitotic exit behavior of a different nuclear protein complex, the NPC, a systematic comparative analysis between HeLa cells (Otsuka et al., 2023) and normal rat kidney cells (Dultz and Ellenberg, 2010; Dultz et al., 2008) has shown a highly consistent behavior between these cell types. In our view, it is therefore reasonable to assume that the presented results on the behavior of the conserved SMC complexes during cell division measured in HeLa cells are likely to be similar in other human and animal cell lines. However, as for any other cell line, we cannot exclude that HeLa cells may exhibit cell type–specific aspects.

## Materials and methods
### Cell culture
HK cells (RRID: CVCL_1922) were obtained from S. Narumiya (Kyoto University, Kyoto, Japan) and cultured in high-glucose DMEM (41965-062; Thermo Fisher Scientific) supplemented with 10% FBS (10270-106, Lot. 42F2388K; Thermo Fisher Scientific), 100 U/ml penicillin-streptomycin (15140-122; Thermo Fisher Scientific), and 1 mM sodium pyruvate (11360-039; Thermo Fisher Scientific) at 37°C, 5% $CO_2$ unless otherwise stated. Cells were grown in cell culture dishes (Falcon) and passaged every 2–3 days via trypsinization with 0.05% Trypsin-EDTA (25300-054; Thermo Fisher Scientific) at 80–90% confluency. Mycoplasma contamination was checked regularly and confirmed negative.

### FCS-calibrated confocal time-lapse imaging
Cell samples for FCS-calibrated confocal time-lapse imaging were prepared according to Politi et al. (2018). Specifically, 2 days before the experiment, two 0.34-cm² wells of an 18-well chambered coverglass (μ-slide, 81817; Ibidi) were seeded with 3,750 HK WT cells. 24 h prior to the experiment, one well of HK WT cells was transfected with a plasmid expressing monomeric EGFP, and 2,000–4,000 genome-edited cells expressing the protein of interest (POI) endogenously tagged with (m)EGFP were seeded in a third well. 1.5 h prior to imaging, DMEM medium was exchanged to phenol-red free $CO_2$-independent imaging medium based on Minimum Essential Medium (M3024; Sigma-Aldrich) containing 30 mM HEPES (pH 7.4), 10% FBS, 1X MEM non-essential amino-acids (11140-050; Thermo Fisher Scientific), and 50–100 nM 5-SiR-Hoechst (gift from G. Lukinavičius, Bucevičius et al., 2018). In addition, after 1.5 h of DNA labeling by 5-SiR-Hoechst, 500-kDa dextran-Dy481XL (Cai et al., 2018) was added to the genome-edited cells to facilitate cell segmentation.

Fluorescence correlation spectroscopy (FCS)-calibrated imaging was performed on Zeiss LSM780 (equipped with ConfoCor 3 unit, controlled by ZEN 2.3 Black software, Version 14.0.18.201; Zeiss) and LSM880 (controlled by ZEN 2.1 Black software, Version 14.0.9.201; Zeiss) laser-scanning microscopes with an inverted Axio Observer microscope stand, equipped with an in-house constructed incubation chamber for temperature control set to 37°C (without $CO_2$ due to use of $CO_2$-independent imaging medium) and using a C-Apochromat 40×/1.2 W Korr UV-Vis-IR water-immersion objective (421767-9971-711; Zeiss). Microscope calibration by FCS was performed as described by Politi et al. (2018), but using 10 nM Atto488 carboxylic

acid (AD 488-21; ATTO-TEC, Kapusta, 2010) in ddH$_2$O instead of AF488 coupled to an H$_2$O-hydrolyzable NHS ester group to estimate the confocal volume in FCS measurements. This led to ~30% larger confocal volume estimates in better agreement with other methods for confocal volume determination (Buschmann et al., 2009). This change resulted in a systematic drop of the protein concentrations measured proportional to the change in confocal volume size compared with previous measurements using AF488-NHS (Politi et al., 2018). Ten FCS measurements of 1 min each were performed to estimate the effective confocal volume in the well with Atto488 solution. FCS measurements of 30 s were performed in the nucleus and cytoplasm in WT cells not expressing mEGFP to determine background fluorescence and photon counts. Experiment-specific calibration factors were obtained from interphase cells expressing mEGFP by correlating measured fluorescence intensities and absolute mEGFP concentration calculated from 30-s FCS measurements (Politi et al., 2018).

Calibrated four-dimensional confocal time-lapse imaging was performed on cells expressing the mEGFP-tagged protein of interest (POI) using a combination of MyPic macros for Zen-Black software (https://git.embl.de/grp-ellenberg/mypic), AutoMicTools library (https://git.embl.de/halavaty/AutoMicTools) for ImageJ (Schindelin et al., 2012), and ilastik (Berg et al., 2019). Specifically, metaphase cells were automatically identified in multiple predefined fields of view by low-resolution imaging of the DNA channel (5-SiR-Hoechst). Subsequently, cells of interest were imaged for the next 150 min with a time-resolution of 2 min to capture anaphase onset (AO) and 120 min of progression through mitotic exit with 31 z-slices with a voxel size of 250 nm in xy and 750 nm in z, covering a total of 75 × 75 µm in xy (300 × 300 pixels) and 22.5 µm in z, which was sufficient to cover the whole cell volume, in the GFP ((m)EGFP-tagged POI), DNA, Dextran-Dy481XL (extracellular space), and transmission channels. A previously developed computational pipeline (Cai et al., 2018) was adapted to track and segment dividing cells from high-zoom time lapses in 3D based on the nuclear (SiR-Hoechst) and cellular (Dextran-Dy481XL) landmarks. The third eigenvalue of the segmented chromatin mass, representing the thickness of the chromosomal volume, was utilized to detect AO as chromosomes begin to be segregated toward opposite cell poles. All mitotic exit time series were aligned to AO and set as the t = 0 min time point. All individually aligned time series displayed a very consistent increase in chromatin volume over time, rendering any further alignment dispensable.

### Estimation of protein numbers from FCS-calibrated images

Fluorescence intensities in image voxels were converted to absolute protein concentrations and numbers based on the experiment-specific calibration line (calibration factor [= slope] and background intensity) and the 3D binary masks of the nucleus and the cell. The average protein concentration was calculated by multiplying the calibration factor (slope of the calibration line) by the average background-corrected fluorescence intensity in all nuclear, cellular, or cytosolic pixels (cytosol = within the cell, but excluding the nucleus). The absolute protein number inside each compartment was achieved by integrating all background-corrected fluorescence intensities and multiplying them with the calibration factor.

### Full cell cycle imaging

About 750–1,000 genome-edited cells expressing the POI endogenously tagged with EGFP were seeded 2 days before the experiment into a 0.34-cm² well of an 18-well chambered cover glass (µ-slide, 81817; Ibidi) and incubated at 37°C, 5% CO$_2$. 20 h later, cells were arrested in S-phase for 15–16 h by changing the medium to DMEM supplemented with 2 mM thymidine (T1895; Sigma-Aldrich). Cells were subsequently released from S-phase arrest by washing three times with DMEM. 4 h after release, the medium was exchanged to phenol-red free, CO$_2$-independent imaging medium (see above) containing 50–100 nM 5-SiR-Hoechst, and 1 h later, 500-kDa dextran-Dy481XL was added as a cell outline marker (added later due to interference with efficient SiR-Hoechst staining). Imaging was started 6 h after release from S-phase, well before the first mitotic division. As a control of the effect of S-phase arrest, ~3,750 asynchronous cells were seeded 1 day before imaging into the well of an 18-well Ibidi µ-slide, and imaging was carried out 1.5 h after the addition of the imaging medium containing 5-SiR-Hoechst and addition of 500-kDa dextran-Dy481XL. Imaging was carried out on a Zeiss LSM780 and LSM880 using a C-Apochromat 40×/1.2 W Korr UV-Vis-IR water-immersion objective (421767-9971-711; Zeiss) with a custom-made objective cap for automated water dispension with a field of view (FOV) size of 177.12 × 177.12 µm covering a z-range of 22.5 µm with 253 nm pixel size in xy and 750 nm in z and a pixel dwell time of 0.76 µs. 0.2% laser power of the 488 nm Argon laser line was used to ensure minimal bleaching, and GFP fluorescence was recorded on the GaAsP detector (499–553 nm range, gain set to 1,100). Four FOVs were automatically imaged every 10 min with an autofocus step before every single 3D stack (based on peak reflection of 514 nm laser line at glass sample interface). Depending on the cell cycle length and whether synchronous or asynchronous cells were used, total imaging time varied from 25 to 40 h to capture two subsequent mitosis events for most cells present in the FOV. Image data was processed using an adapted computational pipeline (Cai et al., 2018) performing 3D segmentation based on chromatin (5-SiR-Hoechst) and cellular landmarks (500 kDa Dextran), as well as cell tracking of single cells using the 3D centroid of the chromatin mass. After manually filtering out duplicate or poorly segmented single-cell tracks, single-cell cycles were cropped out based on the cellular and nuclear volume information, resulting in a list of full-cell cycle tracks ranging from one anaphase/telophase to the next. These full cell cycle tracks were aligned to the first division and subsequently interpolated and fit to a common average cell cycle timing. Calibration of the measured fluorescence intensities was performed not through direct FCS-calibrated imaging, but by setting the concentration of proteins inside a cell (con_cell) in the second mitosis (when the S-phase arrest effect has ceased) to the mean concentration of proteins inside a cell measured in asynchronous FCS-calibrated metaphase cells, resulting in a conversion factor that was used to transform measured fluorescence intensities to absolute protein numbers and concentrations at all

other timepoints. While bleaching of GFP-tagged proteins was not tested over the course of an entire cell cycle, we assume it to be minimal due to low laser exposure (488 nm: 0.2%, pixel dwell: 0.76 μs, 1 stack every 10 min) and the fact that cellular concentrations of all proteins did not change from one mitosis to the next.

## Simple western

Protein separation, immunodetection, and quantification from cell lysates were performed in a Jess Automated western Blot System (Bio-Techne) using 12–230 kDa and 66–440 kDa fluorescence separation capillary cartridges (SM-FL004-1, SM-FL005-1; Bio-Techne). For this, total protein lysates were prepared for each cell line and condition of interest by growing cells in a 10-cm dish until ~80% confluency, subsequently washing with PBS and resuspending cells in 500 μl of lysis buffer (RIPA buffer [R0278; Sigma-Aldrich], 1 mM PMSF [P7626; Sigma-Aldrich], cOmplete EDTA-free Protease Inhibitor Cocktail [04693132001, 1 tablet/10 ml; ; Roche], and PhosSTOP [4906845001, 1 tablet/10 ml; Roche]) with the help of a cell scraper (on ice). Cells were then lysed by two cycles of freezing in liquid nitrogen and thawing at 37°C. After centrifugation for 10 min at ~16,000 × $g$ and 4°C, the supernatant containing soluble total protein extracts was separated and kept at –80°C until use. Total protein was quantified with a Pierce BCA Protein Assay Kit (23227; Thermo Fisher Scientific) and diluted to 0.4 μg/μl final concentration including 1x Master Mix (from EZ Standard Pack 1 (PS-ST01EZ-8; Bio-Techne). Loading of samples and detection reagents into the Simple Western (SW) microplate was conducted following the provider's instructions. Detection was achieved by ECL using anti-rabbit and anti-mouse secondary HRP antibodies (042-206/042-205; Bio-Techne) and Luminol-S/Peroxide solution (043-311/043-379; Bio-Techne). Capillary electrophoresis was run and analysis was conducted with the Compass for SW software (Bio-Techne) following the provider's guidelines.

## Preparation of homozygous endogenous knock-in cell lines

Genome-edited cell lines generated in this study (HK Rad21-EGFP-AID CTCF-Halo-3xALFA #C7 and HK Nup153-mEGFP-FKBP12$^{F36V}$ #C10 (dTAG technology: Nabet et al., 2018) were obtained by C-terminal tagging of CTCF and Nup153 in HK RAD21-EGFP-AID (Davidson et al., 2016) or HK WT parental cell lines, respectively, using the CRISPR/Cas9 method, as previously described (Koch et al., 2018; Callegari et al., 2024). In brief, a linear DNA donor sequence encoding for the tag of interest (and corresponding 50 base pair long homology arms) was electroporated into the parental cell line, together with the catalytic Cas9/gRNA ribonucleoparticle complex. For this, we used Alt-R S.p. HiFi Cas9 Nuclease V3 (1081061; IDT) and single gRNAs (see Supplementary Information). Edited cells expressing the tags of interest were selected by FACS sorting, and the correct tagging of all target copies was subsequently validated as described in Callegari et al. (2024). Expression of the POI at endogenous levels was confirmed by simple western analysis using specific anti-CTCF and anti-Nup153 antibodies combined with total protein normalization (Protein

Normalization Module, DM-PN02; Biotechne; see also Fig. S4, D–F; and Fig. 2, B, and C). Homozygous tagging of the POI was confirmed by PCR screening (with primer pairs spanning the insertion site), simple western, and digital PCR. Digital PCR (dPCR) allows us to quantify the copy number of specific sequences of interest in a template genome by partitioning the amplification reaction (including a primer pair and an internal fluorescent probe, per region to be quantified) into thousands of nanodroplets, each containing zero to a few DNA molecules. Upon amplification of the region of interest in a given droplet, the specific internal probe is released from the DNA, and fluorescence is detected. The count of fluorescent versus non-fluorescent droplets is read out and used to quantify the absolute amount of template DNA. The triple-color dPCR assay used in this work allowed us to quantify: the total number of tags ("allGFP" or "allHalo") integrated into the genome, the number of tags inserted at the intended target locus ("HDR," homologous-directed repair after Cas9-directed DNA cut), and the copy number of a reference sequence located in the vicinity of the target locus. This setup therefore allowed us to quantify how many endogenous alleles are tagged, as well as the detection of excess off-target tag integrations within the recipient genome. Finally, the correct sequence and positioning of the integrated tags were corroborated by PCR amplification and sequencing of the edited genomic regions.

## Fluorescence recovery after photobleaching

Cells for FRAP measurements were seeded at a density of 2.5 × 10$^5$ cells/ml into Ibidi glass bottom μ-Slide channels (80607; Ibidi) 1 day prior to imaging. DMEM was replaced by a CO$_2$-independent imaging medium (as above) containing 50–100 nM 5-SiR-Hoechst at least 1 h before imaging. FRAP experiments were performed on an LSM880 laser-scanning microscope with an inverted Axio Observer controlled by ZEN 2.1 Black software (Version 14.0.9.201; Zeiss), equipped with an in-house constructed incubation chamber for temperature control set to 37°C and using a C-Apochromat 40×/1.2 W Korr UV-Vis-IR water-immersion objective (421767-9971-711; Zeiss). Cells in metaphase and early G1 were selected manually based on their chromatin staining, and FRAP of metaphase cells was performed as described previously (Walther et al., 2018). Cells in the G1 stage were selected manually based on nuclear size and filtered out computationally based on a nuclear size threshold of <1,050 μm$^3$ corresponding to the size of cells about 5 h into the cell cycle according to full cell cycle data of asynchronous cells (exact nuclear size was derived from a 3D stack covering the whole chromatin mass, segmented with a previously developed script [Cattoglio et al., 2019]). A single image was recorded prior to bleaching, recording five z-planes in metaphase and early G1, three z-planes in G1 with a pixel size of 213 × 213 × 750 nm, pixel dwell 1.7 μs and a FOV size of 27.25 × 27.25 μm for metaphase and G1 cells and of 42.5 × 42.5 μm for early G1 cells, respectively in the EGFP (488 nm argon laser line, excitation power: 1%, GaAsP detection range set to 499–562 nm, gain set to 1,000) and SiR-Hoechst channels (633 nm diode laser, excitation power 0.2–0.4%, GaAsP detection range set to 641–696 nm, gain set to

1,000). Subsequently, a square region covering half of the chromatin/nucleus area in the middle z-plane was bleached using similar laser power for metaphase, early G1, and G1 cells (488 nm laser power: 100%). While metaphase plates were bleached with one bleach step (45 × 35 pixels, 150 repetitions), early G1 and G1 cells were bleached three times within 30 s to completely bleach the freely diffusion soluble pool (45 × 35 pixels for eG1, 60 × 50 pixels for G1, 3 × 50 repetitions), enabling the determination of chromatin-bound fractions. The fluorescent recovery was recorded by time-lapse imaging every 20 s for another 30 frames with the settings described for the pre-bleach image, resulting in minimal bleaching throughout the imaging period (<10%).

FRAP image analysis was performed using a previously developed custom-written ImageJ script (Walther et al., 2018), adapted to enable the analysis of metaphase, early G1 and G1 cells at the same time, as well as an R-script for downstream data processing (Walther et al., 2018). In brief, this analysis script aggregates the (m)EGFP-POI and SiR-Hoechst fluorescence intensity data along the major 2D chromatin axis (segmented using SiR-Hoechst channel) into a 1D profile. Using a gap of 14 pixels in the center of the 1D profile, the border of the bleaching ROI was omitted to avoid boundary effects. The weighted mean fluorescence intensities (using SiR-Hoechst) in the unbleached and bleached regions were computed as described in Walther et al. (2018). As in Gerlich et al. (2006); Walther et al. (2018), the weighted normalized difference between the unbleached and bleached region.

$$\frac{F_{ub}(t) - F_b(t)}{F_{ub}(0) - F_b(0)}$$

was used as a readout for the residence time and immobile fraction. A single exponential function

$$a + (1 - a) \ e^{-\left(k_{off}\right)t}$$

was employed to fit the normalized fluorescence recovery data. The parameter $a$ represents the immobile fraction and $k_{off}$ is the unbinding rate constant.

### FRAP to investigate Cohesin-dependence of CTCF chromatin association

FRAP measurements of CTCF after depletion of RAD21 were carried out in G1 cells of genome-edited HK cells in which all alleles of RAD21 were tagged with an AID degron and EGFP and all alleles of CTCF were tagged with Halo (see above). G1 cells were selected based on nuclear volume, but no stringent size filter was applied since the variance of individual measurements was found to be minimal and not dependent on nuclear volume. Complete depletion of RAD21 in these genome-edited cells was achieved by incubation with inole-3-acetic acid (IAA, I5148; Sigma-Aldrich) for at least 1.5 h. For rescue of RAD21 depletion, exogenous RAD21-EGFP was overexpressed for at least 24 h prior to the start of the experiment. FRAP measurements were carried out as described above, however bleaching and imaging of fluorescence recovery was performed using 561 nm excitation of the Halo-TMR (G8252; Promega) ligand coupled to endogenous CTCF-Halo (excitation power: 0.7%, GaAsP detection range

set to 570–624, gain set to 1,000) after 10 min of labelling with Halo-TMR at a concentration of 100 nM at 37°C in imaging medium. Interestingly, we found that CTCF-Halo displayed a reduced chromatin residence time and immobile fraction in the absence of IAA, unlike CTCF-EGFP endogenously tagged in a different cell line. We found that this correlated with a leaky degradation of RAD21 in the RAD21-EGFP-AID CTCF-Halo cell line, reducing RAD21 levels about 40% relative to our CTCF-EGFP line (using Simple Western of asynchronous cell lysates, RAD21 detected via anti-RAD21 antibody (05-908, 1:50; Merck Millipore, Fig. S4 G). Overexpression of RAD21 rescued this effect, bringing CTCF-Halo residence time and bound fraction almost back to WT levels (data not shown). For comparison with our ΔRAD21 and ΔRAD21+rescue conditions, we therefore decided to use our CTCF-measurements as WT reference condition.

### Cell synchronization by mitotic shake-off
To synchronize HK cells in mitosis for subsequent protein degradation or timed release into early G1 or G1, we used a combination of Nocodazole treatment and a mitotic shake-off. In brief, cells were regularly passaged (every second day) and seeded into a T-175 flask (353112; Corning) to reach a confluency of around 80% after 16–24 h of incubation. 1 h prior to mitotic shake-off, cells were incubated in 12 ml of DMEM complete medium supplemented with 82 nM Nocodazole (SML1665; Sigma-Aldrich) to enrich mitotic cells. The mitotic shake-off was conducted by banging five times the cell culture flask on a table covered with ~5 paper tissues. After confirming the detachment of most mitotic cells by inspection on a microscope, the mitotic cell suspension was transferred to a 15 ml Falcon tube and centrifuged for 3 min at 90 × g. The resulting cell pellet was resuspended in 150 μl DMEM + 82 nM Nocodazole and the cell density was counted. 35 μl of cells at a desired density (between 1.2 × 10^6 cells/ml and 2.5 × 10^6 cells/ml) were seeded into an Ibidi μ-Slide glass bottom slide (80607; Ibidi) with channels pre-coated for 15 min with poly-L-lysine (P8920; Sigma-Aldrich). Ibidi slides were incubated for 15 min at 37°C, 5% CO₂ to allow cells to attach. 100 μl of DMEM complete medium supplemented with 82 nM Nocodazole was added to cells in every Ibidi μ-Slide channel prior to any further treatment.

### Immunofluorescence
Fixed cells were prepared for immunostaining by permeabilization with 0.25% Tergitol (15S9; Sigma-Aldrich) in PBS for 15 min and subsequent incubation in blocking buffer (2% BSA, 0.05% Tergitol in PBS) for at least 30 min at room temperature (RT, 20–25°C in this work). Primary antibody incubation was performed in blocking buffer at 4°C in a humidified chamber overnight (16–24 h), followed by washing with blocking buffer (three times, 5 min). Secondary antibody hybridization was performed in blocking buffer for 1 h at RT. After washing with PBS (three times, 5 min), samples were post-fixed with 2.4% PFA (15710; EMS) in PBS for 15 min, quenched with 100 mM NH₄Cl in PBS for 10 min and washed in PBS. Samples used for LoopTrace-based chromatin tracing were permeabilized with Triton X-100 instead of Tergitol at

the same concentration for consistency with previous experiments.

## Protein depletion during mitosis

For the degradation of Nup153, SMC4, RAD21, and CTCF during mitosis, we used genome-edited HK cells in which all copies of the POI were endogenously tagged with a dTAG degron system (Nup153-mEGFP-FKBP12$^{F36V}$, Nabet et al., 2018, 2020), or an Auxin-inducible degron tag (SMC4-Halo-mAID [Schneider et al., 2022], RAD21-EGFP-AID [Davidson et al., 2016], CTCF-mEGFP-AID [Wutz et al., 2017]). Nocodazole-assisted mitotic shake-off was conducted as described above and 35 µl of mitotic cells were seeded into Ibidi glass bottom µ-slides (80607; Ibidi) pre-coated with poly-L-lysine (15 min) at a density of 2–2.5 × 10$^6$ cells/ml. Cells were allowed to attach for 15 min at 37°C, 5% CO$_2$. Subsequently, the depletion of degron-tagged proteins was conducted for 1.5 h in the presence of 82.5 nM Nocodazole and each specific degradation-triggering ligand (Nup153: 250 nM dTAG-13 [SML2601; Sigma-Aldrich] & 500 nM dTAG$^V$-1 [6914; Tocris]; SMC4: 1 µM 5-Ph-IAA [30-003; BioAcademia]; RAD21 & CTCF: 500 µM IAA). Afterwards, cells were released into mitotic exit by washing out Nocodazole through cell incubation for 45–90 min in fresh medium supplemented with dTAGs, 5-Ph-IAA or IAA, respectively. Then, cells were either pre-extracted by washing in PBS and then incubating with 0.25% Tergitol in PBS for 1 min followed by PFA-fixation (ΔSMC4, ΔNIPBL), or fixed directly with 2.4% PFA in PBS for 15 min (ΔNup153), followed by quenching of PFA with 100 mM NH$_4$Cl in PBS and washing with PBS. Immunofluorescence was performed as described above, using the following primary antibodies: mouse anti-RAD21 (05-908, 1:500; Merck Millipore), rabbit anti-SMC2 (ab10412, 1:1,000; Abcam); rabbit anti-CTCF (07-729, 1:2,000; Merck Millipore) or rabbit-anti CTCF (Wutz et al., 2020, Glycine Elution, 1;3,000). Secondary hybridization was performed using fluorescently tagged antibodies: AF647 goat anti-rabbit (A21245, 1:1,000; Invitrogen), AF594 goat anti-rabbit (A11037, 1:1,000; Life Technologies), AF555 goat anti-mouse (A28180, 1:1,000; Invitrogen) or AF594 goat anti-mouse (A11005, 1:1,000; Life Technologies). Stained and post-fixed cells were imaged on a Nikon Ti-E2 equipped with a Lasercombiner, a 60× SR P-Apochromat IR AC 60× 1.27 NA water immersion objective, a CSU-W1 SoRa spinning disk unit and an Orca Fusion CMOS camera in spinning disk mode, operated using NIS Elements 5.2.02 (Nikon). Per condition (WT/ΔPOI), at least 5 z-stacks covering a ROI size of 261.46 × 261.46 × 21 µm were acquired in the DAPI channel (405 nm excitation), GFP channel (488 nm excitation, degradation control), and immunofluorescence channels (561 or 640 excitation) with a pixel size of about 227 nm in xy and 500 nm in z.

## Pre-extraction of mitotic exit cells for quantitative immunofluorescence imaging

Cells were synchronized in mitosis using a mitotic shake-off in combination with Nocodazole, as described above. Cells were released into mitotic exit by washing out Nocodazole 3x with fresh and pre-warmed DMEM, and allowed to progress for 45 min (= early G1) or 4 h (= G1). Prior to fixation and immunofluorescence staining, cells were pre-extracted for 1 min using

0.25% Tergitol in PBS. Cohesin-STAG1/2 were stained in the respective homozygous EGFP knock-in cell line via the FluoTag-X4 anti-GFP Nanobody conjugated to AF647 (N0304-AF647-L; NanoTag), CTCF was stained through a specific primary antibody (Wutz et al., 2020) and a secondary goat anti-rabbit AF555 antibody (A-21428; Thermo Fisher Scientific).

## Image analysis of non-/pre-extracted mitotic exit samples

Image analysis of 3D stacks of stained mitotic exit cells was performed with a custom-written Python script. In brief, after a mild gaussian blur, the DAPI channel was converted to a 3D binary mask of nuclei used for 3D segmentation (method = triangle). Small objects and cropped nuclei at the image borders were removed automatically, and further quality control to remove poorly segmented, multinucleate or dead cells were removed manually using napari. Interactive viewing of the nuclei images and binary masks via napari was also used to classify cells as "mitosis" or "interphase" (representing all nuclei past anaphase). After classification, the nuclei mask and labels were used to extract fluorescence intensities of the endogenous POI-GFP, as well as stained proteins in the unprocessed 488, 561, and 647 nm (if applicable). Image background from regions devoid of cells was subtracted from mean nuclear pixel intensities in every image channel.

## Spot-bleach assay and analysis

Cells for spot-bleach measurements were seeded at a density of 2.5 × 10$^5$ cells/ml into Ibidi glass bottom µ-Slide channels (80607; Ibidi) and grown for 16–24 h. 1 h before imaging, DMEM was replaced by CO$_2$-independent imaging medium (as above) containing 50–100 nM 5-SiR-Hoechst. Spot-bleach experiments were performed on a LSM880 laser-scanning microscope with an inverted Axio Observer controlled by ZEN 2.1 Black software (Version 14.0.9.201; Zeiss), equipped with an in-house constructed incubation chamber for temperature control set to 37°C and using a C-Apochromat 40×/1.2 W Korr UV-Vis-IR water-immersion objective (421767-9971-711; Zeiss). Cells were screened at low-resolution live imaging in the SiR-Hoechst channel and image acquisition was started once a cell undergoing anaphase onset was identified. At 5, 10, 15, 20 and 30 min after anaphase onset, an image of the dividing cell in the GFP (488 nm emission) and DNA (SiR-Hoechst, 633 nm emission) was acquired and used to place and initiate a 30 s continuous illumination with a diffraction limited focused laser beam (488 nm, ~1.5 µW laser power, corresponding to 0.1% Argon laser power). This resulted in a clear depletion of the chromatin-bound (m)EGFP-tagged protein pool (chromatin-bound = bound for more than the measurement period of 30 s) and minor bleaching of the overall cellular pool that readily replaced the bleached soluble fraction at the measured spot. Measurement timepoints were distributed between the two daughter cells to further minimize light exposure of a single cell. During the 30 s illumination, emitted fluorescence was continuously measured using the GaAsP detector in photon counting mode. The mean of the first (prebleach) and last (postbleach) 500 ms of the fluorescence depletion trace was used to calculate the chromatin-

bound fraction for each measurement based on the following formula:

$$Bound\ fraction = \frac{prebleach - postbleach}{prebleach}\ 100$$

In addition to the measurements shortly after mitosis, chromatin-bound fractions of each POI were measured in asynchronous interphase cells. Measured bound fractions were calibrated using exogenous H2B-EGFP (low expression level, positive control representing ∼100% chromatin bound fraction) and freely diffusing mEGFP (unbound control, representing 0% chromatin bound fraction) expressed in a HK WT cell background and measured in asynchronous interphase cell nuclei. The average calibrated chromatin-bound fractions of 10 spot-bleach measurements per protein per timepoint was interpolated (the asynchronous interphase measurements were set to 300 min after anaphase onset for this purpose) and used to calculate the average number of chromatin-bound POIs at each timepoint during mitotic exit using the FCS-calibrated protein number information from Fig. 1, D and F; and Fig. S1 Q.

### Calculation of average copy numbers per megabase DNA
Average protein copy numbers per megabase DNA were calculated from FCS-calibrated imaging data of mitotic exit (Fig. 1 F and Fig. S1 Q). This data represents total protein copy numbers co-localizing with DNA for either (1) the entire chromosome mass during mitosis or (2) both daughter nuclei summed up (after anaphase onset). To calculate the protein copies that are bound to DNA, the average number of protein copies co-localizing with DNA during each given timepoint was multiplied with the respective chromatin-bound fraction estimate derived from spot-bleach or FRAP data.

The HK genome is hypotriploid with 7.9 Gb (Landry et al., 2013). Thus, the total genomic content of replicated mitotic cells is 2 × 7.9 Gb. Similarly, the total genomic content of both daughter cells is 2 × 7.9 Gb. The total protein copy numbers were divided by 15,800 Mb to achieve an average per megabase count, assuming that all investigated proteins occupy the mappable HeLa genome at equal frequency.

### Cell synchronization and immunofluorescence for chromatin tracing
To prepare HK cells expressing AID-EGFP-tagged Cohesin-STAG2 as well as HK WT cells for chromatin tracing in interphase, 120 µl of asynchronous AID-tagged and WT cells were seeded at a 1:1 ratio and a total density of 5 × 10⁵ cells/ml into PBS-washed channels of Ibidi µ-Slide glass bottom slides (80607; Ibidi) and cultured for 20 h at 37°C, 5% CO₂ in DMEM supplemented with 40 µM BrdU/BrdC (ratio 3:1, BrdU: B5002; Sigma-Aldrich, BrdC: sc-284555; Santa Cruz Biotech). Degradation of EGFP-AID-STAG2 was induced by the addition of 500 µM IAA (I5148; Sigma-Aldrich) for 2 h at 37°C, 5% CO₂ in DMEM. Cells were then fixed using 2.4% PFA (15710; EMS) in PBS for 15 min, followed by quenching of PFA with 100 mM NH₄Cl in PBS (5 min) and washing with PBS. To prepare cells in early G1, HK WT, and STAG2-AID, cells were grown for 20 h in a T-175 flask (353112; Corning) in the presence of 40 µM BrdU/BrdC (ratio 3:1)

to reach a confluency of around 80% suitable for mitotic shake off. Nocodazole-arrest, mitotic shake-off, and resuspension of mitotic cells were performed as described above. Enriched mitotic HK WT and STAG2-AID cells (Wutz et al., 2020) were diluted to 2.5 × 10⁶ cells/ml, mixed 1:1 and 35 µl of this cell suspension was seeded into Ibidi µ-Slide glass bottom slides (80607; Ibidi) precoated with poly-L-lysine, and incubated for 15 min at 37°C, 5% CO₂ to allow cells to attach. Degradation of STAG2 was induced upon the addition of 500 µM IAA in the presence of Nocodazole, ensuring near-complete degradation within 45 min. Release into mitotic exit was triggered by Nocodazole washout using DMEM containing 500 µM IAA. Cells were fixed 80 min after release. Live imaging of cells at this point showed that they are on average about 45 min past anaphase. After fixation, early G1 and asynchronous interphase cells were permeabilized for 15 min using 0.25% Triton X-100 (T8787; Sigma-Aldrich) in PBS, and 0.1 µm Tetraspec beads were added to the Ibidi channels (1:100 dilution from stock, T7279; Thermo Fisher Scientific) to be used as fiducials for drift correction. After blocking with 2% BSA in 0.05% Triton X-100 at RT for at least 30 min, primary labeling of STAG2 was performed overnight at 4°C in a humidified chamber (with rabbit-anti STAG2, Glycine Elution, 1:200, Sumara et al., 2000), followed by hybridization with an AF488-labeled secondary antibody (goat-anti-rabbit AF488, A-11034; Molecular Probes).

### Non-denaturing FISH (RASER-FISH)
Non-denaturing FISH (RASER-FISH) as well as FISH library design and amplification were performed as described previously (Beckwith et al., 2023, *Preprint*; 2024, *Preprint*). In brief, cells were incubated with 0.5 ng/µl DAPI in PBS at RT for 15 min to sensitize DNA for UV-induced single-strand nicking of the replicated strand containing BrdU/C. Subsequently, the cells were exposed (without Ibidi lid) to 254 nm UV light for 15 min (Stratalinker 2400 fitted with 15W 254 nm bulbs-part no G15T8). The nicked strand of DNA was then digested using Exonuclease (1 U/µl, M0206; NEB) in NEB buffer 1 at 37°C for 15 min in a humidified chamber. Cells were post-fixed using 5 mM Bis(NHS)PEG5 (803537; Sigma-Aldrich) in PBS for 30 min at RT to preserve cell fixation during primary FISH library hybridization at 37°C. Hybridization of primary FISH probe libraries targeting 1.2 Mb regions (Chr14 50.92–52.10 Mb, Chr5 149.50–150.70 Mb, Chr2 191.11–192.31 Mb) with 12 kb genomic resolution (one trace-spot = tiled set of ∼150 FISH probes with common docking handle) was performed by incubation with hybridization buffer (50% formamide [FA, AM9342; Thermo Fisher Scientific], 10% [wt/vol] dextran sulfate [D8906; Sigma-Aldrich] in 2xSSC [AM9763; Thermo Fisher Scientific]) containing the FISH probe libraries at a final concentration of 100–200 ng/µl DNA per library for one to two nights at 37°C in a humidified chamber. After primary hybridization, channels were rinsed three times with 50% FA in 2xSSC, washed again twice with 50% FA in 2xSSC for 5 min at RT, and finally washed with 2xSSC containing 0.2% Tween. RNA–DNA hybrids were removed by incubating cells with 0.05 U/µl RNAse H (M0297S; NEB) for 20 min at 37°C in RNAse H buffer (NEB). To image and segment whole 1.2 Mb tracing loci, secondary FISH probes

serving as bridges between all primary probes of a whole 1.2 Mb locus and a common imager strand were applied at a concentration of 100 nm in secondary hybridization buffer (20% ethylene carbonate [EC, E26258; Sigma-Aldrich], 2xSSC) for 20 min at RT rocking. Secondary probes were then washed with 30% FA in 2XSSC at RT (three washes, 5 min each) and two additional washes with 2xSSC. Prior to imaging, DNA was stained with 0.5 ng/µl DAPI in PBS for 5 min at RT.

## Chromatin tracing using LoopTrace

3D DNA trace acquisition using a custom-built automated fluidics setup was performed as described in Beckwith et al. (2023, *Preprint*) and in https://git.embl.de/grp-ellenberg/tracebot. In brief, 12-mer imager strands with 3′ or 5′-azide functionality (Metabion) complementary to the docking handles employed by the primary FISH probe library or the bridged regional barcode probes added during secondary hybridization were fluorescently labeled with Cy3B-alkyne (AAT Bioquest) or Atto643-alkyne (Attotec) using click chemistry (ClickTech Oligo Link Kit; Baseclick GmbH) according to the manufacturer's instructions to enable dual-color tracing. Fluorescently labeled 12-mer imagers were diluted to a final concentration of 20 nM in 5% EC 2X SSC in a 96-well plate and placed on the stage of a custom-built automated fluidics setup based on a GRBL controlled CNC stage (Beckwith et al., 2023, *Preprint*). Furthermore, a 3-well deep plate containing washing buffer (10% FA, 2X SSC) and stripping buffer (30% FA, 2XSSC) covered with parafilm, as well as a 24-well plate containing imaging buffer (0.2X Glucose Oxidase [G7141; Sigma-Aldrich], 1.5 mM TROLOX [238813; Sigma-Aldrich], 10% Glucose, 50 mM Tris, and 2X SSC, pH 8.0) were placed on the stage of the automated fluidics setup. A syringe needle mounted in place of the CNC drill head was connected to the sample and a CPP1 peristaltic micropump (Jobst Technologies, flow rate of 1 ml/min at maximal speed) using 1 mm i.d. PEEK and silicone tubing (VWR), allowing to pull liquids out of the well plates and through the sample channel in an automated manner. Imaging was performed on a Nikon Ti-E2 microscope equipped with a Lasercombiner, a 100× 1.35 NA silicon oil immersion objective, a CSU-W1 SoRa spinning disk unit, and an Orca Fusion CMOS camera in spinning disk mode, operated using NIS Elements 5.2.02 (Nikon) in combination with custom-made Python software for synchronization with automated liquid handling. Prior to sequential imaging, a 3D stack of DAPI-stained nuclei (405 nm excitation), STAG2-EGFP fluorescence (488 nm excitation), and the fiducial beads (561 or 640 excitation) was acquired as a reference stack for cell classification with a pixel size of 130 nm in xy and 300 nm in z at a total size of 149.76 × 149.76 µm in xy and covering a z-range of 14.1 (interphase)–18.3 µm (early G1). Subsequently, imager strands were sequentially hybridized for ~2 min at 20 nM concentration in 5% EC 2X SSC, washed for 1 min with washing buffer, imaged after the addition of GLOX-based imaging buffer as a 3D stack, stripped for ~2 min using stripping buffer, and washed again for 1 min. 3D stacks acquired during sequential imaging had equal pixel sizes and z-range as before but were acquired only in the 561- or 640-nm channels

(100% laser power, 100 ms exposure time, triggered acquisition mode) to image fiducial beads and Cy3B or Atto643-labeled imagers, respectively.

## Analysis of LoopTrace data

Processing of acquired tracing data was performed as described in Beckwith et al. (2023, *Preprint*) with code available under https://git.embl.de/grp-ellenberg/looptrace. In brief, nd2 image files were converted to OME-ZARR format. Images were drift-corrected based on cross-correlation, and sub-pixel drift was corrected by fitting the fiducial bead signal to a 3D Gaussian function and subsequent correction for calculated sub-pixel drift. Images were deconvolved using the experimental PSF extracted from fiducial beads. Identification of tracing regions was performed based on regional barcodes using an intensity threshold. Detected spot masks were then used to extract regions of interest for the 3D-superlocalization of individual trace spots by fitting with a 3D Gaussian. Finally, extracted traces were corrected for chromatic aberration between the 561 and 642 image channels by affine transformation obtained by least squares fitting of the centroid of fiducial beads imaged in both channels, and traces were assigned to nuclei classified as "interphase," "early G1," or "mitosis."

The resulting interphase and early G1 DNA traces were grouped into "WT" or "ΔSTAG2" based on their AF488 intensity, and the subsequent analysis was performed as described in Beckwith et al. (2023, *Preprint*). In brief, all fits were quality-controlled for their signal-to-background ratio, standard deviation of the fit, and fit center distance to the regional barcode signal. Traces containing <20 high-quality fitted positions were removed from further analysis. Median pairwise distances were calculated for all 3D coordinates within a single trace and used to display either pairwise-distance maps or contact maps by calculating the frequency of contacts below a certain 3D distance (set to 120 nm). Different matrices were achieved by subtraction of "dSTAG2" from "WT" pairwise distances. Scaling plots were generated from pairwise distance matrices as well, essentially plotting all measured 3D distances for every given genomic distance.

## Sample preparation for STED microscopy

To prepare genome-edited HK cells expressing endogenously EGFP-tagged Cohesin-STAG1/2 for STED microscopy in early G1 or G1, cells were synchronized in mitosis and subsequently released into mitotic exit, pre-extracted, PFA-fixed, and immunostained. Cell synchronization was performed by mitotic shake-off as described above. 35 µl of Nocodazole-arrested enriched mitotic cells were added at a density of $1.2 \times 10^6$ cells/ml (for G1) or $2.5 \times 10^6$ cells/ml (for early G1) to prewashed and poly-L-lysine coated channels of Ibidi µ-Slide glass bottom slides (80607; Ibidi) and incubated for 15 min at 37°C, 5% $CO_2$ to allow cells to attach. Subsequently, three washes with fresh DMEM were performed to wash out Nocodazole, and cells were allowed to exit mitosis for 45 min (for early G1 stage) or 4 h (for G1 stage) at 37°C, 5% $CO_2$. Pre-extraction was performed by washing cells once in PBS and then adding 0.25% Tergitol in 1X PBS for a total of 1 min. Cells were then immediately fixed using 2.4% PFA in

PBS for 15 min, followed by quenching of PFA (15710; EMS) with 100 mM $NH_4Cl$ in PBS and washing with PBS. Fixed cells were prepared for immunostaining by an additional 15-min permeabilization (standard IF protocol) in PBS with 0.25% Tergitol and subsequent blocking using a blocking buffer (2% BSA in 0.05% Tergitol in PBS) for at least 30 min at RT. Incubation with the anti-GFP nanobody (FluoTag-X4 anti-GFP conjugated to Abberior Star 635P, 1:250 dilution N0304-Ab635P; NanoTag) and rabbit anti-CTCF antibody (Glycine-Elution, 1:3,000, Wutz et al., 2020) was performed in blocking buffer at 4°C in a humidified chamber overnight. Secondary hybridization using AF594-conjugated goat-anti-rabbit antibody (1:1,000, A11037; Life Technologies) was performed for 1 h at RT. Samples were post-fixed for 15 min in 2.4% PFA in PBS, with subsequent quenching (100 mM $NH_4Cl$ in PBS) and PBS washing. Samples were imaged by STED super-resolution microscopy on the same day.

## STED microscopy

2D STED imaging was performed on a Leica Stellaris 8 STED Falcon FLIM microscope (Leica Microsystems) controlled by the Leica LAS X software (4.7.0.28176). Samples were imaged at RT using an HC PL APO 86×/1.2 W motCORR STED white water immersion objective. The microscope was equipped with the SuperK FIANIUM FIB-12 white light laser with laser pulse picker (440–790 nm; Leica Microsystems/NKT), 592 nm continuous wave (cw), 660 nm cw, and 775 nm pulsed lasers (MPB Communications), and the HyD S, HyD X, and HyD R detectors. Diffraction-limited as well as STED imaging of CTCF (AF594) and STAG1/2-EGFP (Abberior Star 635P) was performed with excitation at 590 and 645 nm using the white light laser (diffraction-limited/confocal: 3% each, STED: 590 nm: 9%, 645 nm: 6%). Fluorescence was detected with two HyD X detectors using a 601–619 and a 655–750 nm detection window, respectively. Imaging was performed in xy line sequential mode. The pinhole size was set to 1 airy unit and the pixel size was set to 18.88 × 18.88 nm in xy, resulting in images capturing a region of 19.31 × 19.31 μm with 1,024 × 1,024 pixels. STED imaging was performed using a 2D depletion doughnut and 50% power with a 775-nm depletion laser for super-resolved imaging of CTCF-AF594 and at 12% excitation power at 775 nm for imaging of STAG1/2-Abberior Star 635P. STED images were acquired using 16 line accumulations with a scan speed of 200 Hz resulting in a pixel dwell time of 3.85 μs. STED imaging was performed in FLIM mode and the images were post-processed using tau-STED enhancement with background suppression activated and tau-strength set to 0%. Crosstalk between fluorescence channels was quantified with the settings described above and found to be <5% (Fig. S5 C).

## STED image analysis

Preprocessing of diffraction-limited and STED images was performed using Fiji using a custom-made script. Nuclear masks were created by segmenting the CTCF-AF594 diffraction-limited image after Gaussian-blurring and used to crop out nuclei in all image channels. In addition, STED images were background subtracted using the rolling ball algorithm set to a radius of 50 pixels.

Colocalization analysis, as well as spot segmentation, was performed using custom Python scripts. For colocalization analysis, Pearson correlation coefficient of CTCF and STAG1/2 was computed based on cropped nuclei in the respective STED channel. Spot segmentation was performed by first coarsely segmenting spots inside the nuclear mask based on a common threshold (method: Otsu) after applying a mild Gaussian blur (sigma = 1). Image noise resulting in excess tiny spots was filtered out through binary mask erosion and filtering, followed by binary dilation of correctly detected spots. Coarsely segmented spots often represent clusters of spot signals and were further segmented using a combination of local peak finding and watershed. The resulting masks for individual spots were used to extract average pixel intensities in the STED and confocal images. Assuming a z-depth of about 500 nm, the number of detected spots per $\mu m^3$ was compared to protein number estimates derived by FCS-calibrated imaging of early G1 or G1 cells to estimate the overall labeling efficiency.

## STED image simulation

STED images were simulated by generating a desired number of randomly localized spots (single pixels) in an image representing 200 $\mu m^2$ (or 100 $\mu m^3$ assuming a z-depth of 500 nm), given the pixel size of 18.88 nm in the images acquired as described above. The randomly distributed spots were Gaussian blurred (sigma = 2.6) and their pixel intensity was enhanced sixfold to be distinguishable above a random background. Simulated images were analyzed for colocalization or segmented and analyzed to read out their average spot intensities as described above.

## Online supplemental material

Fig. S1 shows supplementary data related to the main Fig. 1. Fig. S2 shows supplementary data related to the main Fig. 3. Fig. S3 shows supplementary data related to the main Fig. 4. Fig. S4 shows supplementary data related to the main Fig. 5. Fig. S5 shows supplementary data related to the main Fig. 6. Table S1 lists antibodies: Antibodies used in this study for immunofluorescence imaging and Simple Western. Table S2 lists cell lines: Cell lines generated and used within this study. Table S3 lists gRNAs: gRNAs employed for cell line generation.

## Data availability

Quantitative and super-resolution imaging data generated in this study are available under BioImage Archive Accession Number S-BIAD1454. Custom code for image and downstream data analysis is available under https://git.embl.de/grp-ellenberg/mitotic-exit-code.

## Acknowledgments

We thank the EMBL Advanced Light Microscopy Facility (ALMF) and the EMBL Imaging Center for microscope support; we thank EMBL's Centre for Bioimage Analysis for support related to image analysis and the EMBL HPC Cluster for use of its computer infrastructure. We thank Gordana Wutz (Research Institute of Molecular Pathology, Vienna BioCenter, Vienna, Austria) for valuable discussions and supply with antibodies and cell lines

related to Cohesin and CTCF. We thank Daniel Gerlich (Institute of Molecular Biotechnology [IMBA], Vienna BioCenter, Vienna, Austria) for sharing the SMC4-mAID-Halo cell line with us.

This work was supported by grants from the National Institutes of Health Common Fund 4D Nucleome Program (grant U01 EB021223/U01 DA047728) to J. Ellenberg, the Deutsche Forschungsgemeinschaft Priority Programme "Spatial Genome Architecture in Development and Disease" (SPP 2202) to J. Ellenberg, as well as by the Paul G. Allen Frontiers Group through the Allen Distinguished Investigator Program to J. Ellenberg. A. Brunner has received a PhD fellowship from the Boehringer Ingelheim Fonds and K.S. Beckwith was supported by the Alexander von Humboldt foundation. Work in the laboratory of J.-M. Peters has received funding from Boehringer Ingelheim, the Austrian Research Promotion Agency (Headquarter grant FFG-852936), the European Research Council (ERC) under the European Union's Horizon 2020 research and innovation program (grant agreements no. 693949 and no. 101020558), the Human Frontier Science Program (grant RGP0057/2018) and the Vienna Science and Technology Fund (grant LS19- 029). Funded by the European Union. Views and opinions expressed are however those of the author(s) only and do not necessarily reflect those of the European Union or the ERC Executive Agency. Neither the European Union nor the granting authority can be held responsible for them. This work is supported by an ERC grant (MITOFOLD, 101142430) to J. Ellenberg.

Author contributions: A. Brunner: Conceptualization, Data curation, Formal analysis, Funding acquisition, Investigation, Methodology, Project administration, Software, Validation, Visualization, Writing - original draft, Writing - review & editing, N.R. Morero: Investigation, Resources, Validation, W. Zhang: Formal analysis, Methodology, Resources, Validation, M.J. Hossain: Data curation, Formal analysis, Software, Supervision, Visualization, Writing - review & editing, M. Lampe: Investigation, Resources, H. Pflaumer: Investigation, A. Halavatyi: Methodology, Software, Writing - review & editing, J.-M. Peters: Conceptualization, K.S. Beckwith: Software, Supervision, Visualization, Writing - review & editing, J. Ellenberg: Conceptualization, Funding acquisition, Methodology, Project administration, Resources, Supervision, Visualization, Writing - original draft, Writing - review & editing.

Disclosures: The authors declare no competing interests exist.

Submitted: 29 May 2024

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

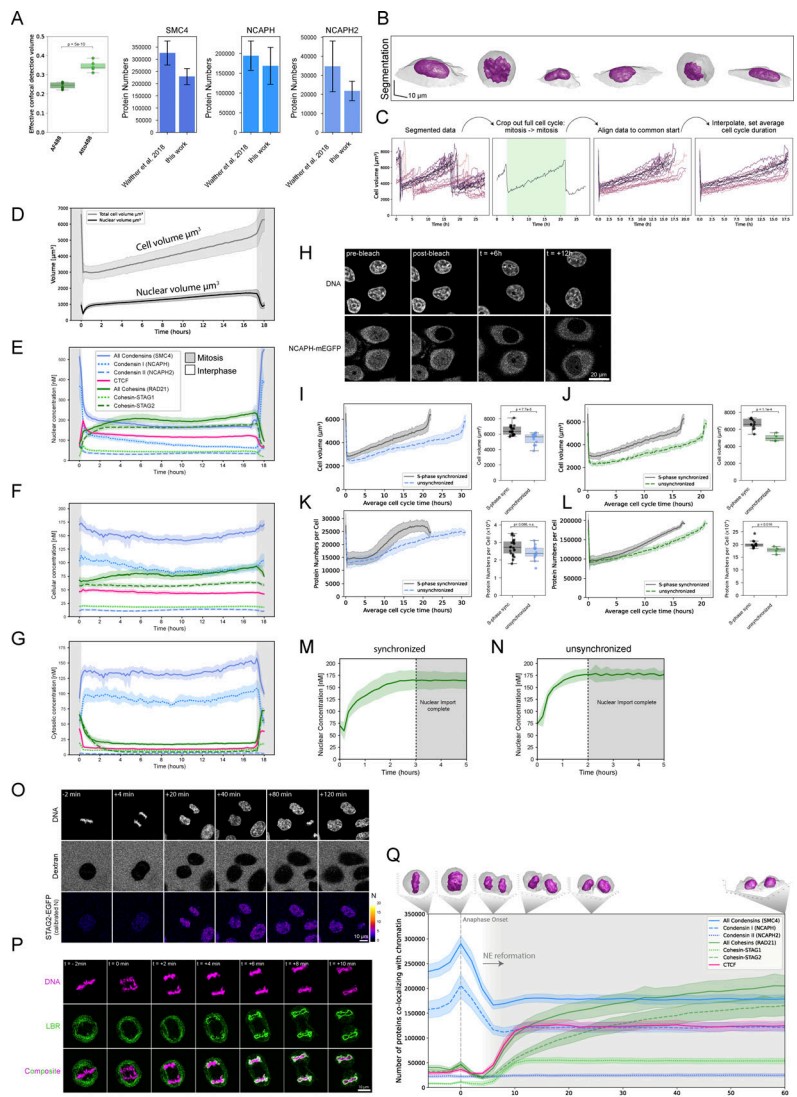

Figure S1. **Quantification of loop extruders throughout the cell cycle and during mitotic exit. (A)** Effective confocal detection volume ($V_{eff}$) determined using AF488-NHS dye (Politi et al., 2018) and Atto488 (this study) at low concentrations (10 nM). 5 30-s-long FCS measurements are performed for $V_{eff}$ determination. Autocorrelation analysis of these measurements and fitting of the diffusion time parameter $\tau_D$ is performed to calculate a 3D-gaussian volume. Differences between AF488-NHS and Atto488 are significant as determined by Student's $t$ test. The updated $V_{eff}$ determination routine resulted in a systematic drop in protein numbers measured in living cells. Protein numbers bound to chromatin (with cytosolic background correction performed as in Walther et al. [2018]) are displayed for SMC4-mEGFP, NCAPH-mEGFP, and NCAPH2-mEGFP, measured in the same HeLa Kyoto cell lines under similar culture conditions at anaphase onset at Walther et al. (2018) and this study ($n_{SMC4}$ = 13, $n_{NCAPH}$ = 11, $n_{NCAPH2}$ = 16). Error bars represent standard deviation of the mean. **(B)** Exemplary segmentation of a full cell cycle track. Segmentations correspond to the top right cell in Fig. 1 C. **(C)** Illustration of full cell cycle data processing based on cell volume information. **(D)** Average cell and nuclear volume for all single-cell trajectories combined. Error bands represent standard deviation. **(E–G)** Mean nuclear (E), cellular (F), and cytosolic (G) concentrations of HeLa Kyoto homozygous knock-in cell lines. Error bands represent a 95% confidence interval. **(H)** The nuclear NCAPH pool of cells endogenously expressing NCAPH-EGFP was photobleached and monitored for up to 12 h. Full bleaching could even be achieved with bleach ROIs that do not target the entire nuclear volume, indicating fast and freely moving protein. The bleached nuclear pool was not recovered by unbleached cytosolic pool throughout the measurement period. **(I–L)** Comparison of cell volume and absolute protein numbers in S-phase synchronized versus asynchronous cells. Bar plots compare cell volume or cellular protein content in the first mitosis after release from S-phase arrest. S-phase arrest resulted in notably increased cell size and protein abundance, influencing protein abundance, production and duration of the next cell cycle. See L and M for the influence of synchronization on Cohesin-STAG2 protein import. **(I and J)** $n_{sync}$ = 21, $n_{async}$ = 12; K and L $n_{sync}$ = 13, $n_{async}$ = 4. Boxes indicate the quartiles of the dataset and the whiskers show the rest of the distribution. **(M and N)** Comparison of Cohesin-STAG2 protein import kinetics in S-phase synchronized versus asynchronous cells. In contrast to S-phase synchronized cells, we found that WT cells required only about 2 h for full Cohesin-STAG2 import into the newly formed nuclei. Error bands represent standard deviation. **(O)** FCS-calibrated imaging of genome-edited HK cells with homozygously EGFP-tagged Cohesin-STAG1 throughout mitotic exit. **(P)** Live cell imaging of the nuclear envelope marker LBR-GFP (Ellenberg et al., 1997) to assess the timing of nuclear envelope reformation after mitosis. While LBR-GFP accumulated on chromatin (stained via 5-SiR-Hoechst) as early as +4 min after anaphase onset (AO), chromatin was almost fully engulfed at +6 min past AO and fully covered at +10 min past AO. **(Q)** Absolute protein numbers co-localizing with chromatin/the two daughter nuclei displayed for genome-edited HK cells with homozygously (m)EGFP-tagged proteins (SMC4: $n$ = 21 cells, NCAPH: $n$ = 14 cells, NCAPH2: $n$ = 19 cells, CTCF: $n$ = 15 cells, RAD21: $n$ = 18 cells, STAG1: $n$ = 25 cells, STAG2: $n$ = 11 cells). Reformation and full establishment of the nuclear envelope as determined by Lamin B receptor (Fig. S1 P) is indicated through a grey background. Error bands represent 95% confidence interval.

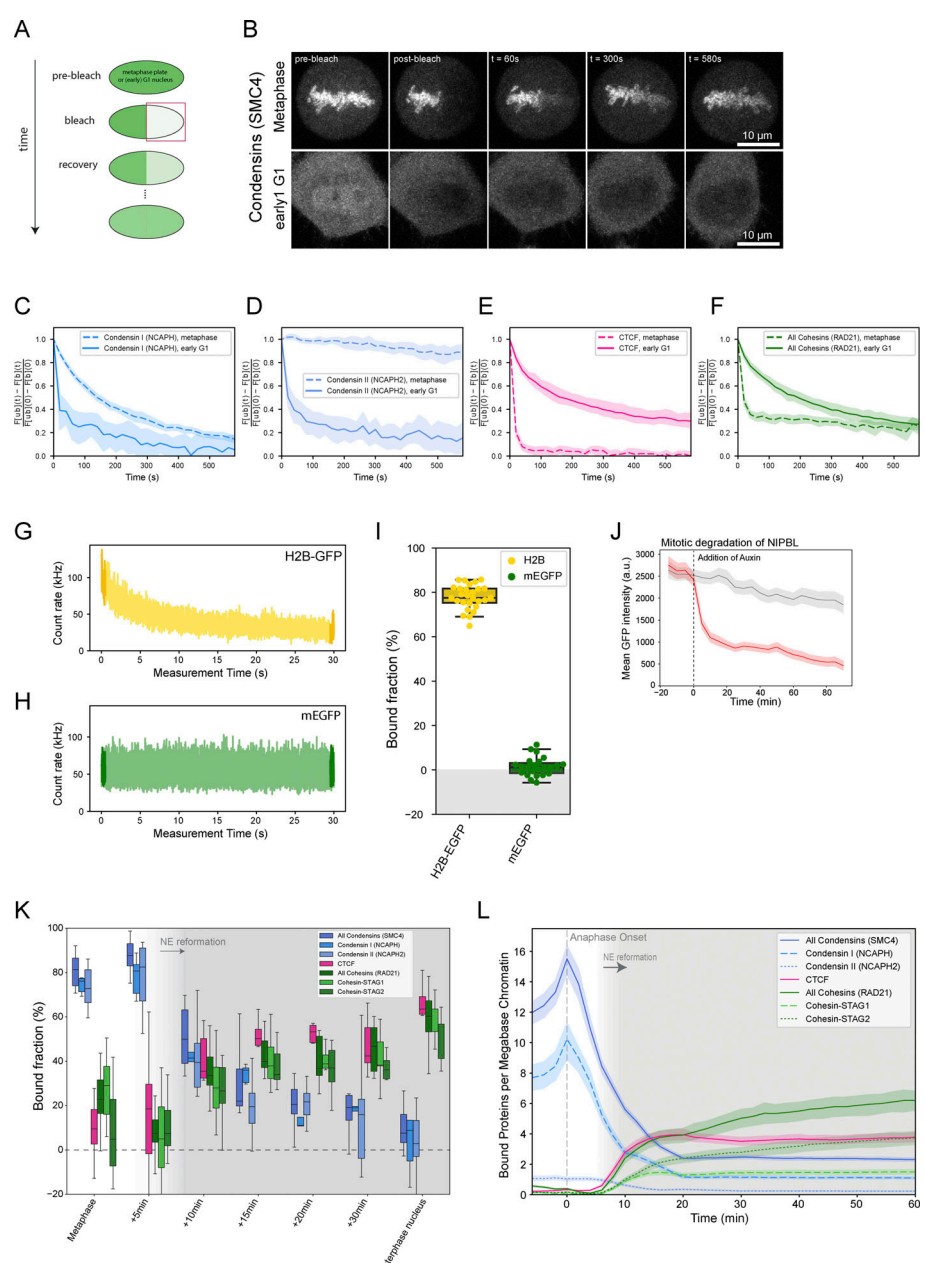

Figure S2. **Fluorescence photobleaching reveals the dynamic transition of chromatin-bound loop extruders from mitosis to interphase. (A)** Scheme of FRAP experiments. Half of the metaphase plate/nucleus was bleached and fluorescent recovery was monitored in bleached and unbleached regions. **(B)** Exemplary FRAP data of SMC4-mEGFP in metaphase and early G1 cells. While one bleach step (150 repetitions 100% laser power) is performed in metaphase cells, three bleach steps (50 repetitions, 100% laser power) are performed in early G1 cells to bleach the entire soluble pool and allow for the determination of the total chromatin-bound fraction. **(C–F)** Metaphase and early G1 FRAP measurements of Condensin I (C, NCAPH-mEGFP, $n_{meta}$ = 16, $n_{eG1}$ = 10), Condensin II (D, NCAPH2-mEGFP, $n_{meta}$ = 12, $n_{eG1}$ = 9), CTCF-EGFP (E, $n_{meta}$ = 15, $n_{eG1}$ = 9) and Cohesin (F, RAD21-EGFP, $n_{meta}$ = 11, $n_{eG1}$ = 10). Error bands represent a 95% confidence interval. **(G)** Representative example of a spot-bleach measurement of HK WT cells exogenously expressing low concentration of stably chromatin-bound H2B-EGFP. H2B-EGFP chromatin bound fraction was used to calibrate spot-bleach measurements. **(H)** Representative example of a spot-bleach measurement of HK WT cells exogenously expressing the low concentration of monomeric EGFP. mEGFP chromatin-bound fraction was used to calibrate spot-bleach measurements. **(I)** Chromatin-bound fractions of H2B-EGFP (used as a calibration reference for 100% chromatin-bound, $n$ = 27) and mEGFP (used as a calibration reference for 0% chromatin-bound pool, $n$ = 23). Chromatin-bound fractions of all other proteins of interest were scaled accordingly. **(J)** Mean cellular fluorescence intensity of endogenous NIPBL tagged with AID-EGFP in WT condition or upon addition of auxin measured in Nocodazole-arrested mitotic cells. Depletion of the protein pool happened within 20 min. Low signal/background ratio required high light-doses leading to bleaching of NIPBL-EGFP and autofluorescence. **(K)** The fraction of chromatin-bound Condensin and Cohesin isoforms as well as CTCF determined using the spot-bleach assay at different time points during mitotic exit. Every bar plot represents at least 10 individual datapoints measured in 10 separate cells. Boxes indicate quartiles and error bars show the rest of the sample distribution. **(L)** Absolute number of proteins bound to chromatin was determined by multiplication of chromatin-bound fractions shown in K with absolute protein numbers colocalizing with chromatin as determined in Fig. 1, E and F; and Fig. S1 Q and displayed as per-megabase-count assuming an equal distribution of the proteins on the entire 7.9 Mb HeLa genome (Landry et al., 2013). Grey background indicates the reformation of the nuclear envelope. Error bands represent 95% confidence interval.

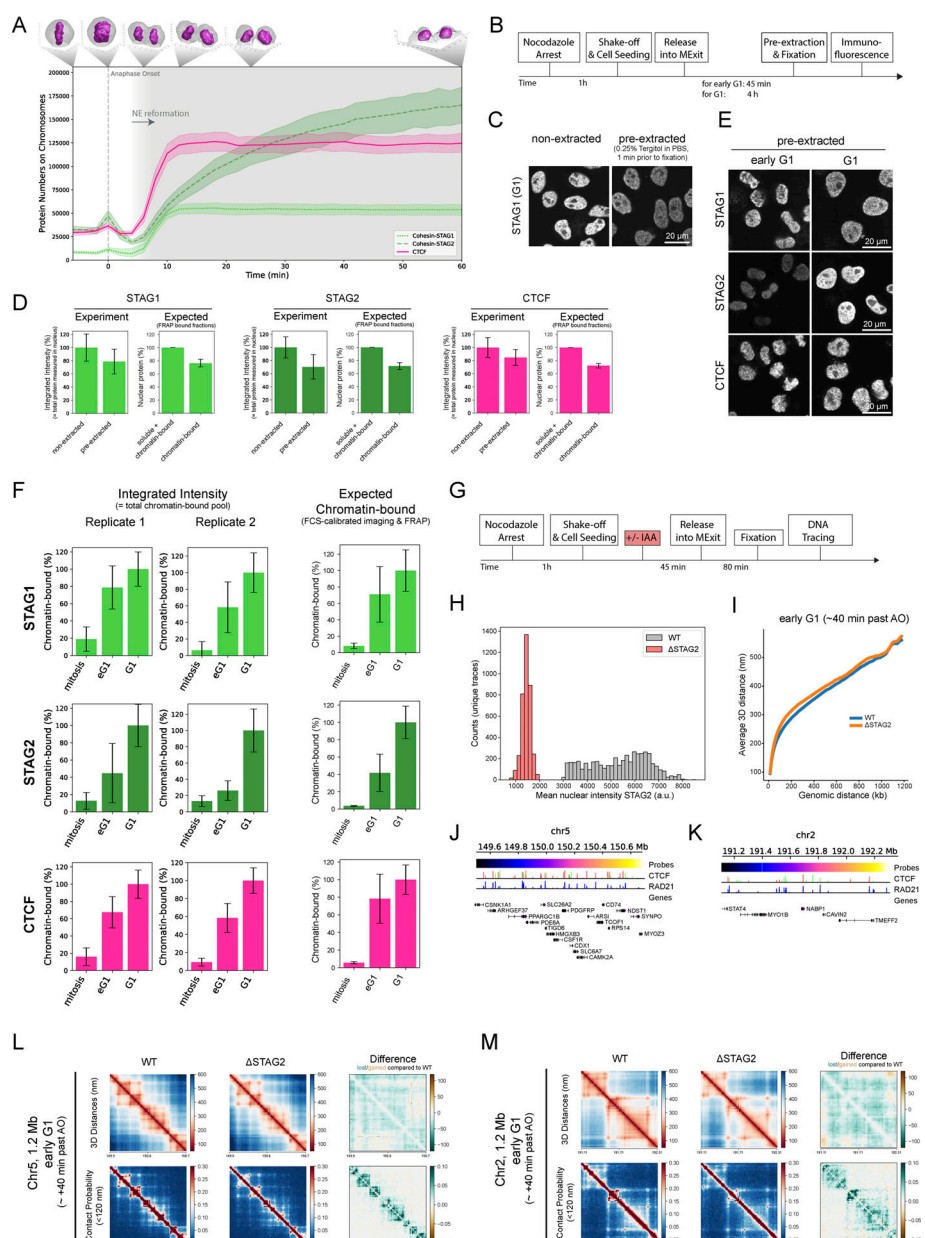

Figure S3. **Cohesin-STAG1 and CTCF co-bind chromatin early after mitosis and cooperate in TAD reformation. (A)** FCS-calibrated protein numbers colocalizing with chromatin are displayed for genome-edited HK cells with homozygously EGFP-tagged Cohesin-STAG1 (n = 25 cells), Cohesin-STAG2 (n = 11 cells), and CTCF (n = 15 cells). Error bands represent 95% confidence interval. **(B)** Experimental scheme for the synchronization of cells in early G1 and G1, with subsequent pre-extraction of soluble protein and immunofluorescence to visualize chromatin-bound proteins using specific antibodies. **(C)** Exemplary microscopy images of non-extracted and pre-extracted G1 cells in endogenous STAG1-EGFP knock-in cell lines, STAG1-EGFP was detected via GFP nanobody. **(D)** Validation of pre-extraction of the soluble Cohesin-STAG1 ($n_{non-extracted}$ = 80, $n_{extracted}$ = 230), Cohesin-STAG2 ($n_{non-extracted}$ = 16, $n_{extracted}$ = 92), and CTCF ($n_{non-extracted}$ = 16, $n_{extracted}$ = 92) pools. Pre-extraction results are compared with bound-fraction estimates derived from half-nuclear photobleaching FRAP experiments (Fig. 5). Error bars represent the standard deviation of the mean. **(E)** Exemplary microscopy images of pre-extracted early G1 and G1 cells, stained for Cohesin-STAG1, Cohesin-STAG2, or CTCF. **(F)** Integrated fluorescent intensity of pre-extracted cells in mitosis, early G1 and G1 stained for Cohesin-STAG1, Cohesin-STAG2 or CTCF represent the total chromatin-bound pool. >150 cells were analyzed for early G1 and G1, respectively. Total chromatin-bound pool measured by pre-extraction & IF is compared to expected chromatin-bound protein numbers estimated by FCS-calibrated imaging and spot-bleach (Fig. S2 L). Error bars represent standard deviation of the mean. **(G)** Experimental scheme for mitotic degradation of Cohesin-STAG2 using genome-edited HK cells with homozygously AID-EGFP tagged Cohesin-STAG2, followed by release into mitotic exit and chromatin tracing using LoopTrace (Beckwith et al., 2023, *Preprint*). **(H)** Mean nuclear intensity of immuno-stained STAG2 in WT condition or after depletion of endogenously tagged STAG2. Nuclei and the corresponding traces could be clearly classified into WT or ΔSTAG2. **(I)** Scaling plot showing the genomic versus Euclidian distance relationship of the three 1.2 Mb regions sampled at 12 kb resolution in WT or ΔSTAG2 cells. Traces from ΔSTAG2 are slightly less compact compared to WT. **(J and K)** Overview of the traced 1.2 megabase locus on chromosome 5 (J) and chromosome 2 (K) with genes as well as ChIP-seq binding sites for RAD21 and CTCF (from the ENCODE portal [Sloan et al., 2016], https://www.encodeproject.org/) with the following identifiers: ENCFF239FBO (RAD21), ENCFF111RWV (CTCF); CTCF directionality annotations from Rao et al. (2014). **(L and M)** Distance and contact matrices of a 1.2 megabase region on chromosome 5 (L) and chromosome 2 (M) traced at a genomic resolution of 12 kb in early G1 cells with and without Cohesin-STAG2. Differences between WT and ΔSTAG2 are highlighted for distance and contact probability maps.

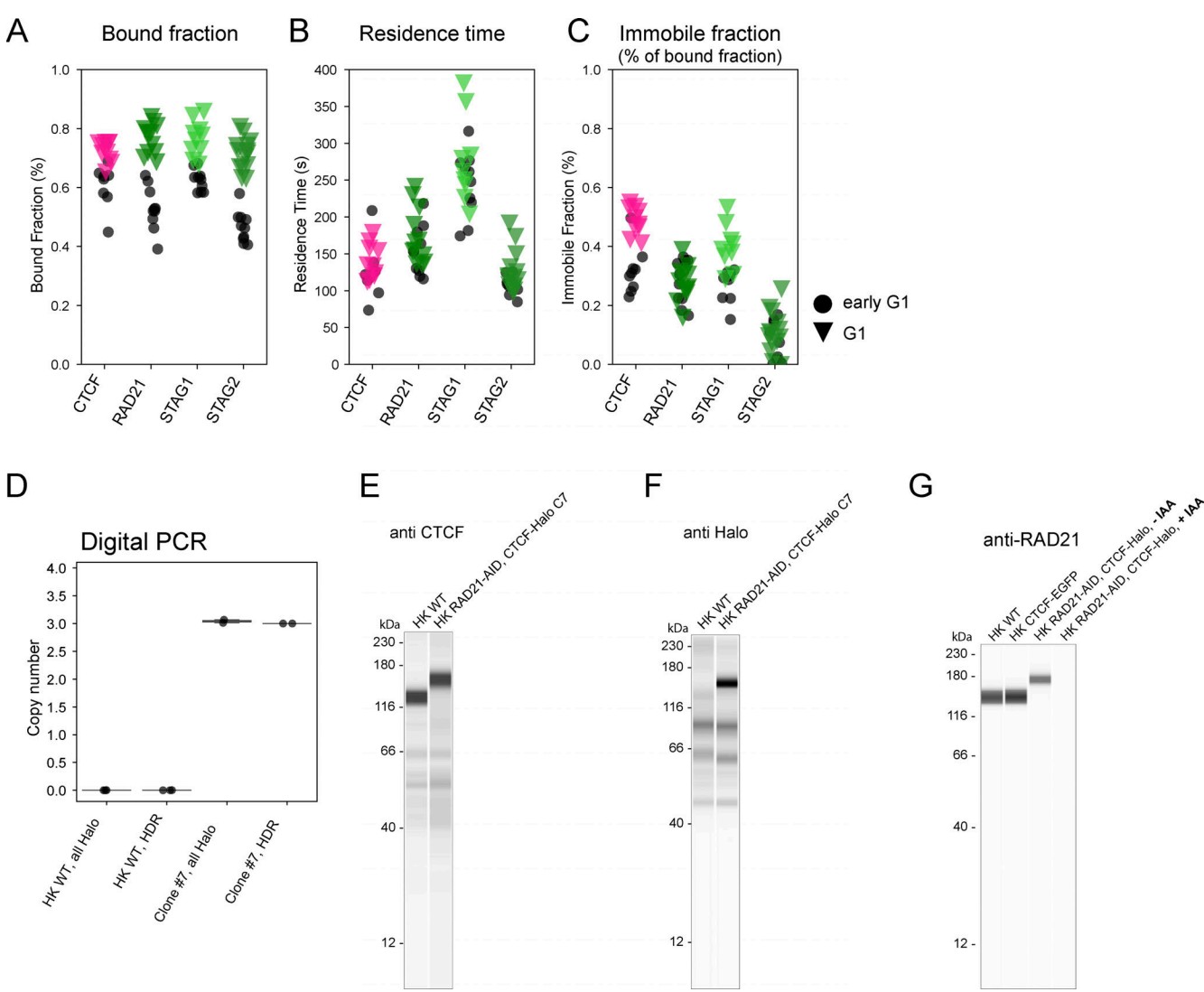

Figure S4. **FRAP reveals increased chromatin association of Cohesins and CTCF from early to late G1. (A–C)** Chromatin-association metrics derived from FRAP measurements in HK cells in which CTCF, RAD21, STAG1, and STAG2 were endogenously tagged with EGFP. Comparison between early G1 and later G1 measurement timepoint. **(A)** Chromatin bound fractions were calculated based on the remaining fluorescent intensity in the unbleached region after bleaching. After the third bleach iteration the entire soluble nuclear protein pool was bleached. Differences between early G1 and G1 were significant for all proteins tested given a significance level of P = 5% (Kolmogorov-Smirnov test, CTCF: P = 0.0037, RAD21: P = 3.09 × 10$^{-6}$, STAG1: P = 0.0002, STAG2: P = 1.75 × 10$^{-6}$). **(B)** Chromatin residence times were derived by fitting FRAP recovery with a single exponential function with an immobile fraction component. Differences between early G1 and G1 were non-significant for all proteins tested given a significance level of P = 5% (Kolmogorov-Smirnov test, CTCF: P = 0.39, RAD21: P = 0.28, STAG1: P = 0.54, STAG2: P = 0.22). **(C)** Immobile fractions were derived by fitting FRAP recovery with a single exponential function with an immobile fraction component. Differences between early G1 and G1 were significant for CTCF and STAG1 given a significance level of P = 5% (Kolmogorov-Smirnov test, CTCF: P = 0.0004, RAD21: P = 0.5577, STAG1: P = 0.0037, STAG2: P = 0.1497). **(D–F)** Validation data for the correct tagging of CTCF in the HK RAD21-EGFP-AID CTCF-Halo-3xALFA (#C7) cell line generated in this study. **(D)** The copy number of Halo-3xALFA tags integrated at the target locus (HDR assay) and within the whole recipient genome (all-Halo assay) was determined in HK WT and edited (clone #7) cell lines by digital PCR. The complete tagging of all three endogenous CTCF copies was confirmed by PCR-amplification of the target locus and sequencing analysis. Digital PCR results indicate that no extra off-target copies of the tag are present at the genome of edited cells. **(E and F)** Simple western analysis of protein extracts from HK WT cells and the RAD21-EGFP-AID CTCF-Halo-3xALFA #C7 line created and used in this study. For each condition, 3 µl of total protein lysate at 0.4 µg/µl was loaded into the assay's microplate. **(E)** Immunolabeling of CTCF shows a clear shift of the CTCF band to higher molecular weight in the edited cell line, indicating successful and homozygous gene tagging. Anti-CTCF antibody (07-729; EMD Millipore) was used at 1:40 dilution. **(F)** Immunolabeling of Halo shows correct tagging of a protein of the expected MW for CTCF-Halo-3xALFA, and no expression of free Halo tag. Anti-Halo Antibody (G9211; Promega) was used at 1:50 dilution. **(G)** Simple western analysis of protein extracts from HK WT cells, HK CTCF-EGFP cells and the RAD21-EGFP-AID CTCF-Halo-3xALFA #C7 line created and used in this study, the later grown in the absence (–IAA) and presence (+IAA) of auxin in the last 3 h of culture. For each condition, 3 µl of total protein lysate at 0.4 µg/µl was loaded into the assay's microplate. Anti-RAD21 antibody (05-908; Sigma-Aldrich) was used at 1:50 dilution. Complete depletion of RAD21 in the genome-edited cell line was achieved by the addition of auxin. In the absence of auxin, this cell line showed a reduced expression of RAD21 compared to HK WT or the HK CTCF-EGFP cell line due to leaky degradation of RAD21. Nonetheless, the effect of the loss of the remaining RAD21 in the double-knock-in cell line still led to a more dynamic interaction of CTCF with chromatin (assessed by FRAP, data not shown). Source data are available for this figure: SourceData FS4.

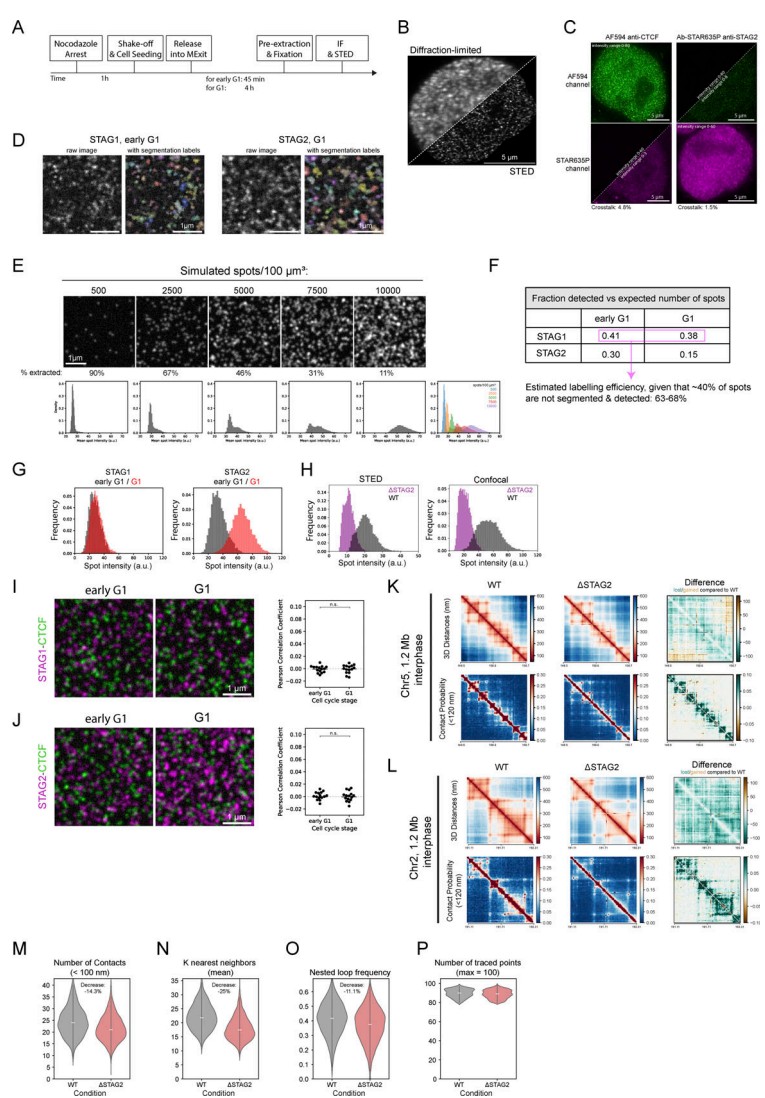

Figure S5. **Quantitative STED image analysis reveals pairwise co-localization of Cohesin-STAG2 after complete nuclear import. (A)** Experimental scheme for mitotic synchronization of genome-edited HK cells with homozygously EGFP tagged Cohesin-STAG1/2, followed by release into mitotic exit, fixation during early G1 or G1, immunostaining for CTCF and STAG1/2-EGFP and subsequent STED super-resolution imaging. **(B)** Exemplary G1 cell expressing STAG1-EGFP, with Cohesin-STAG1 stained via a GFP nanobody. Imaged in confocal or STED mode, respectively. **(C)** Cross-correlation control of STED image channels. Cells in which either CTCF-AF594 or STAG1/2-EGFP-Abberior-STAR-635P were labelled. Cross-talk between imaging channels was assessed and found to be minimal (<5%). **(D)** Illustration of the STED image spot segmentation. **(E)** Benchmarking of spot-counting via the segmentation workflow (D). At higher spot-densities the true spot number is underestimated. Our STED imaging data appears to have very similar spot densities as the simulated images with 2,500–5,000 spots/μm³ (for STAG1 and STAG2 in early G1). Given that only 67–46% of all spots are segmented and counted in these simulated images, we are likely also undercounting protein numbers in our acquired STED data by ∼40%. **(F)** The number of segmented (= detected) spots per cubic micrometer (assuming a typical z-depth of 500 nm for STED, averaged for all three replicates) compared to the expected number of chromatin-bound STAG1/2 proteins per cubic micrometer as measured by FCS-calibrated imaging and FRAP (Tables 1 and 2). Given that the segmentation pipeline underestimates the number of proteins (E) by about 40% (compare density to images in Fig. 6 A) the labelling efficiency can be estimated to about 63–68%. Note the twofold decrease in detected proteins spots for STAG2 from early to later G1. **(G)** Mean intensity of segmented STAG1/2 spots in confocal images of replicate 3. Same results are observed in replicate 1 and 2. Formal significance tests are meaningless due to large sample size. **(H)** Mean intensity of segmented STAG2 spots in STED and images of cells without or with partial depletion of Cohesin-STAG2. **(I)** Image simulation based on detected spot densities in acquired STED images for STAG1 and CTCF, multiplied by 1.67 to account for the fact that the spot-segmentation workflow underestimates STED spot densities around 40%. Differences in Pearson's correlation coefficient between simulated STAG1 and CTCF channels are insignificant based on Student's *t* test. **(J)** Image simulation based on detected spot densities in acquired STED images for STAG2 and CTCF, multiplied by 1.67 to account for the fact that the spot-segmentation workflow underestimates STED spot densities around 40%. Differences in Pearson's correlation coefficient between simulated STAG2 and CTCF channels are insignificant based on Student's *t* test. **(K and L)** Distance and contact matrices of a 1.2 megabase region on chromosome 5 (K) and chromosome 2 (L) traced at a genomic resolution of 12 kb in interphase cells with and without Cohesin-STAG2. Differences between WT and ΔSTAG2 are highlighted for distance and contact probability maps. WT data from 1, ΔSTAG2 data from two independent technical replicates (>400 traces, respectively). **(M–P)** Trace metric analysis from single chromatin traces of 1.2 Mb regions on chromosome 14, chromosome 5, and chromosome 2 (data from all three loci were combined to show one average plot per trace metric). Number of contacts below a physical distance of 100 nm (M), the mean number of k nearest neighbors (N) and the frequency of nested loops (requirement: two base loops have a common anchor point and thereby form a bigger stacked loop, O) are shown for >1,000 individual traces per condition that are at least 80% or more complete (P).

Provided online are Table S1, Table S2, and Table S3. Table S1 list of antibodies: Antibodies used in this study for immunofluorescence imaging and Simple Western. Table S2 list of cell lines: Cell lines generated and used within this study. Table S3 list of gRNAs: gRNAs employed for cell line generation.

