## [Peer Review File · The Journal of Cell Biology]

Quantitative imaging of loop extruders rebuilding interphase genome architecture after mitosis

Andreas Brunner, Natalia Morero, Wanlu Zhang, M. Julius Hossain, Marko Lampe, Hannah Pflaumer, Aliaksandr Halavatyi, Jan-Michael Peters, Kai Beckwith, and Jan Ellenberg

Corresponding Author(s): Jan Ellenberg, European Molecular Biology Laboratory

Review Timeline:

Submission Date:	2024-05-29
Editorial Decision:	2024-07-22
Revision Received:	2024-09-14
Editorial Decision:	2024-10-09
Revision Received:	2024-11-27

Monitoring Editor: Ana Pombo

Scientific Editor: Tim Fessenden

Transaction Report:

DOI: <https://doi.org/10.1083/jcb.202405169>

July 22, 2024

Re: JCB manuscript #202405169

Dr. Jan Ellenberg
European Molecular Biology Laboratory (EMBL)
Cell Biology and Biophysics Unit
Meyerhofstraße 1
Heidelberg, Baden-Württemberg 69117
Germany

Dear Dr. Ellenberg,

Thank you for submitting your manuscript entitled "Quantitative imaging of loop extruders rebuilding interphase genome architecture after mitosis". The manuscript was assessed by expert reviewers, whose comments are appended to this letter. Thank you for your patience during the rather lengthy review process. We invite you to submit a revision if you can address the reviewers' key concerns, as outlined here.

You will see that reviewers uniformly appreciated the conceptual advance made in this work by quantitative measurements of chromatin architecture re-formation after mitosis. Reviewers differed somewhat in the improvements they felt were needed to make this work suitable for publication. We agree with Reviewer 1 that an orthogonal measure of chromatin occupancy, such as chromatin pulldowns, to corroborate measurements of occupancy by STAG1/2 and CTCF must be included in a revision. We encourage you to address other points raised by this reviewer with new data, if possible. Concerning point 3 by this reviewer, we concur with other reviewers that the estimates of protein numbers are a strength of this work and should remain with acknowledgement of limitations of the approach taken. Reviewers 2 and 3 were more enthusiastic, and offered several suggestions to refine the text and claims made. We agree that details on methodology must be included in a revision, and that limitations of this study should be acknowledged where indicated. Finally, in agreement with Reviewer 3, JCB policy does not permit "data not shown."

GENERAL GUIDELINES:

Text limits: Character count for an Article is < 40,000, not including spaces. Count includes title page, abstract, introduction, results, discussion, and acknowledgments. Count does not include materials and methods, figure legends, references, tables, or supplemental legends.

Figures: Articles may have up to 10 main text figures. Figures must be prepared according to the policies outlined in our Instructions to Authors, under Data Presentation, <https://jcb.rupress.org/site/misc/ifora.xhtml>. All figures in accepted manuscripts will be screened prior to publication.

Supplemental information: There are strict limits on the allowable amount of supplemental data. Articles may have up to 5 supplemental figures. Up to 10 supplemental videos or flash animations are allowed. A summary of all supplemental material should appear at the end of the Materials and methods section.

Please note that JCB now requires authors to submit Source Data used to generate figures containing gels and Western blots with all revised manuscripts. This Source Data consists of fully uncropped and unprocessed images for each gel/blot displayed in the main and supplemental figures. Since your paper includes cropped gel and/or blot images, please be sure to provide one Source Data file for each figure that contains gels and/or blots along with your revised manuscript files. File names for Source Data figures should be alphanumeric without any spaces or special characters (i.e., SourceDataF#, where F# refers to the associated main figure number or SourceDataFS# for those associated with Supplementary figures). The lanes of the gels/blots should be labeled as they are in the associated figure, the place where cropping was applied should be marked (with a box), and molecular weight/size standards should be labeled wherever possible. Source Data files will be made available to reviewers during evaluation of revised manuscripts and, if your paper is eventually published in JCB, the files will be directly linked to specific figures in the published article.

The typical timeframe for revisions is three to four months. While most universities and institutes have reopened labs and allowed researchers to begin working at nearly pre-pandemic levels, we at JCB realize that the lingering effects of the COVID-19 pandemic may still be impacting some aspects of your work, including the acquisition of equipment and reagents. Therefore, if you anticipate any difficulties in meeting this aforementioned revision time limit, please contact us and we can work with you to find an appropriate time frame for resubmission. Please note that papers are generally considered through only one revision cycle, so any revised manuscript will likely be either accepted or rejected.

Thank you for this interesting contribution to Journal of Cell Biology. You can contact us at the journal office with any questions at cellbio@rockefeller.edu.

Sincerely,

Ana Pombo
Monitoring Editor
Journal of Cell Biology

Tim Fessenden
Scientific Editor
Journal of Cell Biology

Reviewer #1 (Comments to the Authors (Required)):

In this manuscript, Brunner et al. investigate the spatiotemporal dynamics of chromosome extruders (condensin I, condensin II, cohesin-STAG1, and cohesin-STAG2) during the mitotic exit and G1 phase transition. The authors employed a suite of techniques, including FCS-calibrated 4D live-cell imaging, FRAP analysis, sequential DNA-FISH, and super-resolution microscopy, to make the following observations:

- (1) Leveraging FCS-calibrated 4D live-cell imaging, the authors observed a peak association of condensin I with chromatin at anaphase/telophase, followed by a rapid decline that plateaued upon reformation of the nuclear envelope during mitotic exit.
- (2) The authors observed a gradual enrichment of both cohesin isoforms within daughter nuclei following mitosis. Notably, cohesin-STAG1 exhibited rapid and robust nuclear localization within a short timeframe (~10 minutes post-mitosis). In contrast, cohesin-STAG2 required a significantly longer period to achieve stable nuclear concentration, reaching it approximately 2 hours after mitosis. These findings reveal distinct re-import dynamics for cohesin-STAG1 and cohesin-STAG2. Importantly, functional nuclear pores were found to be essential for the post-mitotic nuclear import of both cohesin isoforms.
- (3) FRAP analysis revealed that a portion of condensin complexes remained chromatin-associated during early G1, a cell cycle phase where cohesin had already been loaded onto the genome. This observation suggests the co-localization of cohesin and condensin on chromatin in early G1. Furthermore, the data indicate independent chromatin binding mechanisms for these two SMC protein complexes.
- (4) Cohesin-STAG1 and CTCF exhibited concurrent and rapid import, followed by their co-localization on chromatin within daughter nuclei. Throughout G1 progression, both cohesin-STAG1 and CTCF displayed a gradual increase in their chromatin binding stability. Interestingly, the presence of cohesin stabilized CTCF during G1 phase.
- (5) Sequential DNA FISH experiments revealed that cohesin-STAG1 and CTCF are sufficient to establish compactly folded G1 structures like TADs. Interestingly, deletion of STAG2 did not significantly impact the overall TAD architecture. However, it did lead to a reduction in the signals associated with smaller nested domains within these TADs.
- (6) Cohesin-STAG2 progressively bind to the chromatin after mitosis and displayed a tendency to form dimer-complexes in late-G1 due to the relatively high concentration (8 complexes/Mb of genome).

The process of how the genome reassembles its interphase structures following mitosis has been a topic of intense investigation. Primarily, chromatin conformation capture techniques, such as in-situ Hi-C, have been used to study this

phenomenon in chemically synchronized bulk cell populations. This current study offers a novel approach by utilizing live-cell imaging to examine the question. The authors leverage their established expertise in absolute quantification of intracellular protein molecules to characterize the dynamic behavior of key chromatin extruders during mitotic exit. By analyzing these protein behaviors, the authors infer the subsequent consequences for chromatin architecture. Notably, the study employs a significant number of endogenously tagged and degron cell lines, involving a substantial effort. The manuscript is well-organized and clearly written, effectively presenting the research. However, I do reserve concerns about potential over-interpretation of some data and potential contradictions with previous findings.

Major points:

1. The FCS 4D live-cell imaging data are convincing. However, the use of S-phase arrested cells raises concerns about potential artifacts introduced by the synchronization process. To mitigate these concerns, while a complete re-analysis of the entire cell cycle may not be necessary, it would be beneficial for the authors to track the behavior of key molecules (e.g., condensin, cohesin-STAG1, and cohesin-STAG2) within an asynchronous population of cells throughout a full cell cycle. This would provide valuable context and strengthen the generalizability of the observed dynamics during mitotic exit.

2. The interpretation of the chromatin-bound protein fraction using FRAP analysis (Fig. 2A) appears to be overly simplistic. The authors' assumption that proteins exist in only two states - chromatin-bound (immobile) and unbound (freely mobile) - might not fully capture the cellular reality.

To strengthen the analysis, it would be valuable for the authors to consider alternative approaches that corroborate the FRAP results. For example, chromatin isolation at defined time points after mitosis (e.g., 5 minutes to 1 hour) followed by quantitative western blotting could provide a more precise quantification of chromatin-bound protein levels for STAG1 and STAG2. This would offer a complementary perspective on their chromatin association dynamics.

3. I have reservations about the absolute quantification of chromatin-bound SMC molecule numbers presented in the manuscript. This quantification relies on the FRAP analysis results, which, as previously discussed, might be an oversimplification. Therefore, presenting specific molecule counts based on this method could be misleading. The authors should tone down these claims and refrain from including precise numbers.

4. The interpretation in Fig. 2D and E, suggesting independent chromatin binding of condensin and cohesin in early-G1, needs further exploration. While the immunofluorescence (IF) staining confirms cohesin's overall nuclear presence unaffected by condensin depletion, it doesn't definitively show if chromatin-bound cohesin levels are impacted. The same applies to the condensin staining.

To definitively assess a potential co-dependence between cohesin and condensin in early-G1, techniques like FRAP or WB are needed. These methods quantify the chromatin-bound fraction of each protein, allowing for a clearer picture of their interaction in the presence or absence of the other.

Interestingly, a recent biorxiv preprint (<https://pubmed.ncbi.nlm.nih.gov/38659940/>) suggests condensin can potentially disrupt cohesin's chromatin association during mitotic entry. This raises the possibility of a similar interplay occurring in the early-G1 phase as well. The authors should discuss these data.

5. The authors observed co-presence of condensin and cohesin on chromatin in early-G1 (Supp Fig. 2). This finding appears to contradict a previous Hi-C study on HeLa cells (PMID: 31685986), which suggested a transient post-mitotic state devoid of both condensin and cohesin, potentially reflecting loop-less chromosome states. How would the authors reconcile these seemingly contradictory observations?

6. The observation that cohesin stabilizes CTCF binding in interphase cells is interesting. However, as discussed above, the authors should demonstrate this finding using an alternative approach. For example, WB or single-molecule tracking could provide more quantitative measurement on CTCF-chromatin association upon cohesin depletion.

7. Several Hi-C studies (PMID: 31776509, PMID: 33730542) observed a sequential formation of loops, with smaller nested loops appearing first, followed by larger loops encompassing them. This aligns with the loop extrusion model, where cohesin travel distance might influence loop size. However, the current study proposes a contrasting model where cohesin-STAG1 establishes larger loops first in early-G1, followed by cohesin-STAG2 mediated formation of smaller nested loops in late-G1. How to reconcile these two models?

8. The authors' sequential DNA FISH data on STAG2-depleted cells provides valuable insights. However, including data from late-G1, in addition to early-G1 (or interphase), would strengthen their findings. STAG2 is known to be most functional during late-G1, and including this stage could potentially reveal a more pronounced effect of depletion on loop formation.

Minor points:

1. Fig. 1C and E should also include STAG1 data to show its rapid increase in nuclear concentration.

2. Supplement Fig. 2E suggested that CTCF remains largely chromatin bound during mitosis, given the sharp decay of the FRAP curve. However, it is well established that CTCF is dramatically evicted from mitotic chromosomes. How to explain this discrepancy.

3. Supplement Fig. 2F measured the fraction of chromatin bound cohesin during interphase vs. mitosis. I'm curious how was this measured in the mitotic samples, since cohesin is completely evicted from mitotic chromosomes.

Reviewer #2 (Comments to the Authors (Required)):

This paper from Brunner et al. combines a very impressive range of cutting-edge approaches - including quantitative live-cell imaging, lots of genome-editing to tag numerous proteins, degron-perturbations, FRAP, tiling-chromosome tracing experiments super-resolution microscopy and more - to dissect the role of SMC complexes, condensin and cohesin, CTCF and associated factors as the genome is being reshaped from mitosis-to-G1. The Ellenberg lab is arguably one of the world's leading labs when it comes to quantitative microscopy - units of proteins per cell or protein concentrations instead of arbitrary units - and as a resource alone, the present paper is incredibly valuable. Beyond this, the paper provides great insights into genome-refolding and makes a case that we should think of STAG1-cohesin and STAG2-cohesin akin to Condensin I and II, making smaller and larger loops. The authors also make a surprising observation that cohesin seems to stabilize the residence time of CTCF and discuss whether SMC extrusion complexes are likely to be monomeric or dimeric. The figures look beautiful and the manuscript is well-written. Properly one of the more impressive papers that integrate diverse and complementary methods that I have seen in a while.

Overall, I think this is a fantastic paper that makes several major and original contributions to the field and I do believe it will be of broad and general interest including to the wide readership of the Journal of Cell Biology.

Nevertheless, there are some issues that need to be corrected in my opinion before this paper is ready for publication. I list these below.

MAJOR POINTS

PROTEINS PER MEGABASE

I very much like putting the numbers in terms of proteins per megabase, e.g. 3 cohesin extruders per average megabase. However, no information of how these calculations were made is provided (unless I missed it, in which case I do apologize). Can you please include a very detailed methods section on these calculations?

These calculations are more subtly than they sound. I can indirectly guess how they did it. In Table 2, authors say that there are 89280 CTCF proteins bound corresponding to 5.7 per megabase. This implies a genome size of $89280 \text{ proteins} / 5.7 \text{ proteins/Mb} = 15,663 \text{ Mb}$ or 15.6 billion basepairs. These numbers are for G1, so there should be no replication.

The haploid human genome is 3.3 Gb, so this would suggest that these HeLa cells are $15.6\text{Gb}/3.3\text{Gb} = 4.7\text{-ploid}$, so somewhere between tetraploid and pentaploid.

How were these numbers arrived at? I think including an SI figure showing how the ploidy was measured is required, as is a full methods description.

When the authors studied mitosis, did they then use a 9.4-ploid genome?

Just to be clear, I am not saying that what the authors did was incorrect, just that it is very important and necessary for them to fully describe what, why and how they did this.

Finally, I would like to see error propagation. Currently, the authors generally provide numbers without confidence intervals.

HELA CELLS AND GENERALITY

Throughout the paper the tone is that the numbers and discoveries described in this paper are "universal truths" akin to laws of physics. But most likely, if the authors were to repeat these studies in diverse mouse and human cell lines, both some of the observations (e.g. telophase behavior) and numbers (e.g. density of SMC complexes) would change substantially. This is fine, and it would be impossible for the authors to perform these studies in all cell types. However, I do think it is important for the authors to describe the results in a less general and more open-minded manner. They should explicitly state that these results may be specific to HeLa cells and that it is no clear how different other cell types and species might be. Throughout the manuscript, I would also ask the authors to tone down their tone - right now, when I read it the tone implies that these are universal truths rather than numbers specific to HeLa cells cultured in a certain way.

The authors should also explicitly comment on the known issues with clone-to-clone variation and cancer cell line variation,

which is well-known, see e.g. <https://www.nature.com/articles/s41592-019-0375-1>

ABRAMO PAPER

The Abramo 2019 paper claims that telophase is completely devoid of SMCs and loop extrusion. However, this conclusion is somewhat controversial, and I know several people in the field who believe this Abramo conclusion is not correct. The authors devote a whole section in the Discussion to this, but the writing and tone is so indirect, that even after reading the paper twice, I cannot tell for sure what the authors believe (though I think I can infer it). If I am not mistaken, the data in this paper disproves the Abramo conclusion and says that although there is reduced SMC complexes in telophase, there is no time when there are no SMC complexes on DNA.

I of course understand that the authors want to be polite and collegial, but I think it would be better if the authors could just clearly state if their results disagree with Abramo instead of the current phrasing which I found quite vague.

BOUND FRACTION CALCULATIONS

The bound fraction calculation appears to be just the fraction of pre-post/pre (Fig 2A). But the methods section says that they calibrate this against exogenous H2B-EGFP which they say represent 100% chromatin binding. This sounds reasonable, but I believe this is incorrect. Natural H2B is expressed during replication (S-phase). Exogenous H2B is expressed also in M, G1, G2 leading to a significant fraction of unbound H2B.

The extent of unbound H2B is typically around 20-25% (see Fig 4H in <https://elifesciences.org/articles/33125>), though no doubt depends on the expression system. Therefore, I anticipate this calibration - although intuitively reasonable - might lead to the wrong answer.

Moreover, the calculation in Fig 2A seems too simplistic when the authors can instead do proper reaction-diffusion modeling of spot-FRAP. Indeed, the bound fractions listed in SI Table 2 for CTCF and cohesin appear too high and inconsistent with several papers from several labs that have used SMT/SPT/FRAP to study these complexes in diverse cell types.

Can the authors re-visit this calculation and their approach?

MONOMERIC VS. DIMERIC SMC COMPLEXES

The authors do some very nice and quantitative STED imaging in Fig 5 and discuss the controversial issue of monomeric vs. dimeric SMC complexes. I think the authors make a compelling case that the dimeric and multimeric complexes they observe may indeed be adjacent, stacked extruders. However, I think they overstate things. They do at some point state that they cannot fully exclude dimeric SMC extruders, but overall I would like to ask the authors to tone down this a little bit, since I do not believe they can fully exclude concentration-dependent dimerization (indeed, the in vitro single-molecule extrusion papers also find concentration dependent dimerization).

NESTED LOOPS AND COHESIN AND CONDENSIN ANALOGIES

On the topic of nested loops and loop density, the difference in compaction between mitotic and interphase chromosomes is currently thought to result largely from differences in loop-extruder density. If you do a polymer simulation, high densities of loop extruders make compacted mitotic chromosomes and lower densities make interphase-like less compacted chromosomes, see e.g. <https://elifesciences.org/articles/14864>

E.g. in Table 2, they say they have 11.6 cohesins per megabase in G1. If you have >10 extruders per Mb would this not result in mitosis-like chromosome compaction? I would like to ask the authors to perform quantitative comparisons between their data and polymer simulation studies to see if the extruder densities make sense with interphase chromosome compaction?

Along these lines the authors make a nice analogy between Condensin I and II and cohesin STAG1 and STAG2 making long loops and nested shorter loops inside. This analogy is highly intellectually appealing, since it suggests a universal nested looping model for SMC extrusion in both mitosis and interphase. However, we also know that the mitosis compaction level is much higher due to much higher loop-extruder density, suggesting that the nested loops inside loops frequency is likely much much lower in interphase. Can the authors more quantitatively compare these including in terms of frequency of nested looping in mitosis and interphase based on their quantitative measurements? I am not suggesting a very extensive analysis, but the polymer simulations studies have typically processivity/separation ratios that produce interphase and mitotic chromosomes.

CHROMOSOME TRACING

The chromosome conformation tracing experiments are really nice and the precision and resolution (12kb, 20 nm) seems very impressive. The fact that they see sub-domain structure, akin to high-resolution Micro-C, e.g. in Fig 5H is quite astonishing and not something I have seen before. All other chromosome tracing papers I have read generally produce quite low quality and blurry contact maps that look substantially worse than Hi-C and Micro-C. I don't really have a question or concern, just want to comment on how high-quality the data appears to be. Maybe this is also partially because they use a better FISH protocol (RASER-FISH) that does not disrupt structure so much.

MINOR POINTS:

1. Line numbers would have been helpful
2. "seamlessly" feels like an inappropriate word for the abstract. Too anthropomorphic.
3. "Zink" should be "Zinc" on page 2
4. Page 2, the Zuin 2022 paper is very nice but not really relevant to the sentence as it does not deal with TADs. A better

reference would be only of the 2016, 2019, 2024 Bernstein papers on Glioma or GIST and CTCF insulator loss leading to aberrant E-P interactions and oncogene overexpression.

5. Page 3 top, the authors state that STAG2 makes up 75% of cohesin. This is not universally true, but varies enormously from cell type to cell type. Please correct.

6. Page 7, authors should compare their cohesin and CTCF residence times to other studies and discuss. E.g. CTCF result seems similar to Hansen 2017, but STAG1 result seems much shorter than Wutz 2020.

7. Page 11 "... longer lived loop structures in interphase, equivalent of TADs", could this stabilization perhaps explain why the protein residence time reported here is a bit shorter than the recently measured loop lifetime of ~20 min (Gabriele 2022).

Reviewer #3 (Comments to the Authors (Required)):

Brunner et al. investigate the localization of cohesin and condensin through mitosis and early interphase. The authors determine that cohesin and condensin are both on the DNA in early G1, and show that cohesin-STAG1 is rapidly imported into the nucleus, while the more abundant cohesin-STAG2 slowly accumulates in the nucleus. This leads the authors to a hypothesis that cohesin-STAG1 and cohesin-STAG2 form a nested-loop structure reminiscent of the structures that condensin II and condensin I can form. The authors use super-resolution imaging to show that STAG2-cohesin seems to cluster together as it starts to accumulate in the nucleus, and propose that when these complexes encounter each other, they may dimerize.

I much enjoyed reading this manuscript. This is a very thorough study, and the figures are clear and appealing. Though some of the data was already known (Figures 3F and 4C/D come to mind), the manuscript also provides valuable new insights into possible dimerization of cohesin complexes. I have only minor comments. Most of them textual:

Textual comments

- Throughout the results section, the authors are careful not to use the term dimerization, but rather colocalization. There are still instances where dimerization is used, like in the results section heading. I think colocalization is a more accurate term in this case. Fine of course to in the discussion consider that colocalization may reflect dimerization, but probably best to adjust the relevant heading in the results section.
- The 2018 paper from the same lab also contains quantifications of condensin complexes on DNA. It would be good to include a direct comparison of the quantifications from 2018 and the new quantifications.
- The nested loop hypothesis of STAG1 and STAG2 was previously proposed by Wutz et al. (2020). Though the authors do extensively reference this paper, it might be good to add that this hypothesis is not entirely novel, and to thus also reference Wutz in the context of this model.
- In their model as well as in the text, the authors claim that there are "no functional interactions between extruders" cohesin and condensin. Though they show that the levels of cohesin are not influenced by condensin depletion and vice versa, I do not think that this proves that there is no functional interaction between the complexes at all. So best to rephrase.

In my opinion, some of the data could be moved to the supplemental figures. The reason being that some of the main figures are currently a bit dense. I would suggest moving the following panels to the supplements:

- Nup153-dependent import of cohesin.
- Effects of cohesin depletion on condensin levels, and vice versa.

Finally, I have some questions regarding the presented data:

- Condensin I levels in the interphase nucleus remain higher than the condensin II levels. This is surprising, as it was previously thought that condensin I is excluded from the nucleus in interphase. I would like to invite the authors to elaborate a bit on this interesting observation.
- For the spot-brightness measurements of a single nanobody, the manuscript states "preliminary data, not shown". In my view, this data should be included.
- Ochs et al. (2024) show that cohesin might form dimers, and that this could make up a large proportion of the looping cohesin. I would like to see an estimate of the proportion of dimers that the authors see in their data.
- It is hypothesized that colocalizing STAG2-cohesin dots do not actively extrude anymore. Does this mean that the complexes are at or near CTCF at this point?
- Do putative 'heterodimers' of cohesin-STAG1 and cohesin-STAG2 also occur, or is it exclusively cohesin-STAG2 that colocalizes with itself?

Reviewer comments are in *black italic*, author responses are in blue.

Reviewer #1 (Comments to the Authors (Required)):

We thank Reviewer 1 for the thorough and critical review of our paper. We very much appreciated the detailed inspection of our presented data and address the Reviewer's comments one by one below.

In this manuscript, Brunner et al. investigate the spatiotemporal dynamics of chromosome extruders (condensin I, condensin II, cohesin-STAG1, and cohesin-STAG2) during the mitotic exit and G1 phase transition. The authors employed a suite of techniques, including FCS-calibrated 4D live-cell imaging, FRAP analysis, sequential DNA -FISH, and super-resolution microscopy, to make the following observations:

(1) *Leveraging FCS-calibrated 4D live-cell imaging, the authors observed a peak association of condensin I with chromatin at anaphase/telophase, followed by a rapid decline that plateaued upon reformation of the nuclear envelope during mitotic exit.*

(2) *The authors observed a gradual enrichment of both cohesin isoforms within daughter nuclei following mitosis. Notably, cohesin-STAG1 exhibited rapid and robust nuclear localization within a short timeframe (~10 minutes post-mitosis). In contrast, cohesin-STAG2 required a significantly longer period to achieve stable nuclear concentration, reaching it approximately 2 hours after mitosis. These findings reveal distinct re-import dynamics for cohesin-STAG1 and cohesin-STAG2. Importantly, functional nuclear pores were found to be essential for the post-mitotic nuclear import of both cohesin isoforms.*

(3) *FRAP analysis revealed that a portion of condensin complexes remained chromatin-associated during early G1, a cell cycle phase where cohesin had already been loaded onto the genome. This observation suggests the co-localization of cohesin and condensin on chromatin in early G1. Furthermore, the data indicate independent chromatin binding mechanisms for these two SMC protein complexes.*

(4) *Cohesin-STAG1 and CTCF exhibited concurrent and rapid import, followed by their co-localization on chromatin within daughter nuclei. Throughout G1 progression, both cohesin-STAG1 and CTCF displayed a gradual increase in their chromatin binding stability. Interestingly, the presence of cohesin stabilized CTCF during G1 phase.*

(5) *Sequential DNA FISH experiments revealed that cohesin-STAG1 and CTCF are sufficient to establish compactly folded G1 structures like TADs. Interestingly, deletion of STAG2 did not significantly impact the overall TAD architecture. However, it did lead to a reduction in the signals associated with smaller nested domains within these TADs.*

(6) *Cohesin-STAG2 progressively bind to the chromatin after mitosis and displayed a tendency to form dimer-complexes in late-G1 due to the relatively high concentration (8 complexes/Mb of genome).*

The process of how the genome reassembles its interphase structures following mitosis has been a topic of intense investigation. Primarily, chromatin conformation capture techniques, such as in-situ Hi-C, have been used to study this phenomenon in chemically synchronized bulk cell populations. This current study offers a novel approach by utilizing live-cell imaging to examine the question. The authors leverage their established expertise in absolute quantification of intracellular protein molecules to characterize the dynamic behavior of key chromatin extruders during mitotic exit. By

analyzing these protein behaviors, the authors infer the subsequent consequences for chromatin architecture. Notably, the study employs a significant number of endogenously tagged and degran cell lines, involving a substantial effort. The manuscript is well-organized and clearly written, effectively presenting the research. However, I do reserve concerns about potential over-interpretation of some data and potential contradictions with previous findings.

Major points:

1. The FCS 4D live-cell imaging data are convincing. However, the use of S-phase arrested cells raises concerns about potential artifacts introduced by the synchronization process. To mitigate these concerns, while a complete re-analysis of the entire cell cycle may not be necessary, it would be beneficial for the authors to track the behavior of key molecules (e.g., condensin, cohesin-STAG1, and cohesin-STAG2) within an asynchronous population of cells throughout a full cell cycle. This would provide valuable context and strengthen the generalizability of the observed dynamics during mitotic exit.

We thank the reviewer for this comment and agree with the importance to control for the effect of S-phase arrest. Indeed, we had already performed control experiments of asynchronous cell populations expressing (m)EGFP-tagged Cohesin-STAG2 and Condensin II and displayed them in the submitted manuscript in supplementary figure 1I-N. These control experiments showed that S-phase arrest (likely due to the cell's prolonged time in interphase) leads to an increase in cell volume and corresponding increase in protein abundance. We showed that the increased protein abundance of Cohesin-STAG2 indeed influences the time needed until nuclear import is complete, as S-phase synchronized cells required approximately 3 hours until the nuclear concentration of Cohesin-STAG2 plateaued (SFig. 1M), while asynchronous cells complete nuclear import of Cohesin-STAG2 within 2 hours (SFig. 1N), in line with our mitotic-exit-focused FCS-calibrated 4D imaging data of asynchronous cells presented in figure 1F. Our control experiments of tagged NCAPH2 expressing cells confirmed that S-phase arrest does not change the chromatin-colocalization dynamics of Condensin II during mitotic exit.

An interesting additional point of these control experiments is that while protein expression of Condensin II and Cohesin-STAG2 was increased following S-phase arrest (SFigs. 1K&L), this however, appears to be corrected by the cell right before entry into the next mitosis, when a level very similar to asynchronous cells is reached again (SFig. 1K).

Thanks to the concern of Reviewer 1, we realized that these critical control experiments were poorly referenced in the submitted manuscript, making it very easy to be overlooked. We thus adapted their referencing and changed the text from

“The 3-fold more abundant Cohesin-STAG2, however, reached stable nuclear concentrations only about two hours after mitosis (Fig. 1D, Suppl. Fig. 1I-N).”

to

“The 3-fold more abundant Cohesin-STAG2, however, reached stable nuclear concentrations only about three hours after mitosis in S-phase arrested cells (Fig. 1D). Importantly, control experiments of asynchronous cells revealed that S-phase arrest resulted in increased cell volumes and protein numbers, delaying the complete import of Cohesin-STAG2 (Suppl. Fig. 1I-N). FCS-calibrated 4D live-cell imaging showed that nuclear import of Cohesin-STAG2 requires two hours in asynchronous cells (Suppl. Fig. 1M,N).”

In addition, to make it clear that the highly-resolved 4D FCS-calibrated imaging focused on mitotic exit has been performed in asynchronous cells, we have added **the following** in the text:

“...we next focused our analysis on the transition between Condensin and Cohesin occupancy on the genome during the first 2 hours after mitosis in asynchronous cells, increasing the time-resolution of our FCS-calibrated 4D imaging to 2 minutes (Fig. 1E).”

2. The interpretation of the chromatin-bound protein fraction using FRAP analysis (Fig. 2A) appears to be overly simplistic. The authors' assumption that proteins exist in only two states - chromatin-bound (immobile) and unbound (freely mobile) - might not fully capture the cellular reality.

To strengthen the analysis, it would be valuable for the authors to consider alternative approaches that corroborate the FRAP results. For example, chromatin isolation at defined time points after mitosis (e.g., 5 minutes to 1 hour) followed by quantitative western blotting could provide a more precise quantification of chromatin-bound protein levels for STAG1 and STAG2. This would offer a complementary perspective on their chromatin association dynamics.

We agree with the Reviewer that the interpretation of the spot-bleach data does not capture all possible complexities of dynamically chromatin-bound proteins. Our assay classifies proteins that are chromatin-bound for more than 30 seconds as “chromatin-bound” while freely diffusing proteins and proteins with a chromatin residence time below 30 seconds are classified as “unbound”.

We want to emphasize that the strength of the small spot-bleaching assay is to provide a rapid and reproducible measure for chromatin binding in single cells during mitotic exit, which can be repeated in the same cell over time without arresting mitosis and allows a quantitative estimate of the chromatin-bound fractions of Condensins and Cohesins during this very dynamic period of the cell cycle. However, the assay is not able to fully characterize the intricacies of chromatin residence times of potentially present multiple bound fractions as is possible with classical FRAP, that analyzes larger areas over longer times. Classical FRAP based on half-nuclear bleaching (as shown in Figure 4) prohibits rapid and repeated measurements due to its inherent bleaching of a large fraction of the protein pool and uses higher total illumination dose which perturbs mitotic progression.

We would like to note that the main text explicitly acknowledges the chromatin-bound fraction measurements as estimates. In addition, we have now provided further detail in the material and methods section on how these estimates are calculated from the data.

We appreciate the suggestion of the Reviewer to validate our findings regarding Cohesin binding using an orthogonal approach. The suggestion to perform chromatin isolation at defined timepoints after mitosis followed by quantitative western blotting is difficult to compare to real time photobleaching, as the synchronization efficiency during mitotic exit is still rather limited and, similar to previous experiments performed by Abramo et al. 2019, postmitotic chromatin isolates represent a mixture of mitotic and mitotic exit cells, which cannot be aligned with the minute scale time resolution data we obtained from living single cells. The single cell dynamic data in our view is an important strength of our microscopy approach, as we tracked living cells during mitotic exit and perform repeated spot-bleach measurements in these cells with highly accurate timing.

To use an orthogonal method to examine Cohesin binding to chromatin that can achieve similar temporal precision, we therefore staged individual cells during mitotic exit and correlatively pre-extracted unbound protein from them, then fixed them and determined the bound Cohesins retained on chromatin by quantitative immunofluorescence with specific antibodies. Quantifying the chromatin-bound pool of Cohesin-STAG1/2 and CTCF in extracted and fixed cells, indeed replicated our findings in living cells using FCS-calibrated imaging and the spot-bleach assay (see separate validation data document for more technical detail). We have added this new data to the manuscript, which is now Suppl. Fig. 3B-F) and correspondingly extended our methods section.

3. I have reservations about the absolute quantification of chromatin-bound SMC molecule numbers presented in the manuscript. This quantification relies on the FRAP analysis results, which, as previously discussed, might be an oversimplification. Therefore, presenting specific molecule counts based on this method could be misleading. The authors should tone down these claims and refrain from including precise numbers.

In line with our comments above, we acknowledge that the spot-bleach assay does simplify the complex chromatin-binding landscape of loop extruders. Nonetheless, given our careful analysis of protein co-localization with DNA provided in Figure 1D,F and Figure 3A, in combination with the spot-bleach assay in Figure 2A and complementary corroborating FRAP measurements provided in SFig. 2A-F, Figure 4 summarized and compared with the spot-bleach assay's result in Tables 1&2, we do think our chromatin-binding estimates are well controlled and reproducible and provide a reasonable quantitative estimate of their cellular concentrations. We therefore suggest that they remain in the manuscript, clearly stated as estimates and with an acknowledgement of the experimental limitations of the assay we used to derive them.

4. The interpretation in Fig. 2D and E, suggesting independent chromatin binding of condensin and cohesin in early-G1, needs further exploration. While the immunofluorescence (IF) staining confirms cohesin's overall nuclear presence unaffected by condensin depletion, it doesn't definitively show if chromatin-bound cohesin levels are impacted. The same applies to the condensin staining.

To definitively assess a potential co-dependence between cohesin and condensin in early-G1, techniques like FRAP or WB are needed. These methods quantify the chromatin-bound fraction of each protein, allowing for a clearer picture of their interaction in the presence or absence of the other.

Interestingly, a recent biorxiv preprint (<https://pubmed.ncbi.nlm.nih.gov/38659940/>) suggests condensin can potentially disrupt cohesin's chromatin association during mitotic entry. This raises the possibility of a similar interplay occurring in the early-G1 phase as well. The authors should discuss these data.

We thank the Reviewer for this thoughtful comment and reference to recent new results.

With regards to the IF-experiment, we would like to clarify that the cells were first pre-extracted to remove soluble protein, and then fixed. Retained bound protein was then quantified by immunofluorescence staining. The imaged nuclear protein pool thus represents the chromatin-bound fraction of proteins.

In line with the Reviewers paper reference, Condensin has indeed also been shown to support the eviction of Cohesin from chromatin within the so-called "prophase-pathway" in a previous study by two senior authors of this study (<https://pubmed.ncbi.nlm.nih.gov/15572404/>) and the experiments in Figure 2D and E have been motivated by these findings. We now referenced both the older and very recent findings in the results text which now reads as follows:

"Could this simultaneous binding of mitotic and interphase loop extruders be functionally interlinked, similar to Cohesin's eviction from chromatin by Condensin during prophase (Hirota et al., 2004; Samejima et al., 2024)?"

And in the discussion:

"Although Condensin helps evict chromatin-bound Cohesin during prophase (Hirota et al., 2004; Samejima et al., 2024), we did not find evidence for a functional Condensin-Cohesin interaction that impacts their chromatin binding during telophase and early G1 (Fig. 2D,E)."

5. *The authors observed co-presence of condensin and cohesin on chromatin in early-G1 (Supp Fig. 2). This finding appears to contradict a previous Hi-C study on HeLa cells (PMID: 31685986), which suggested a transient post-mitotic state devoid of both condensin and cohesin, potentially reflecting loop-less chromosome states. How would the authors reconcile these seemingly contradictory observations?*

We thank the Reviewer for raising this comment as our results indeed seemingly contradict the conclusions of the Abramo et al., 2019 paper.

In fact, our results are actually well in line with the Abramo et al. 2019 paper's biochemical chromatin fractionation results. Both Fig. 7a and Extended Data Fig. 8d of Abramo et al. 2019 show a clear co-binding of Condensin I and II isoforms with Cohesin's RAD21. While the chromatin binding of Condensins was reduced in (early)G1 compared to mitosis, a significant pool of Condensins remained chromatin-bound.

While the HiC results of Abramo et al. 2019 did not show clear loop-structure signatures in their P(s) plots, indicative of a lack of chromatin loop extrusion, their chromatin-fractionation data in fact indicates a co-presence of the two complexes. In our reading, the data of this study does in fact not clearly support the interpretation that post-mitotic chromatin is completely devoid of both Condensin and Cohesin complexes. We have addressed this topic now more thoroughly and explicitly in the updated discussion section.

6. *The observation that cohesin stabilizes CTCF binding in interphase cells is interesting. However, as discussed above, the authors should demonstrate this finding using an alternative approach. For example, WB or single-molecule tracking could provide more quantitative measurement on CTCF-chromatin association upon cohesin depletion.*

We also find this finding interesting. That being said, it really is a side observation in our study, whose focus is on mitotic exit. Validating it with orthogonal assays and establishing more detailed cell cycle tracking protocols within interphase is therefore in our view beyond the scope of the revision of this work and rather the subject of future studies on interphase genome architecture. Given that we could validate our postmitotic findings with an orthogonal approach, we feel that it is legitimate to mention this side observation to motivate such future studies.

7. *Several Hi-C studies (PMID: 31776509, PMID: 33730542) observed a sequential formation of loops, with smaller nested loops appearing first, followed by larger loops encompassing them. This aligns with the loop extrusion model, where cohesin travel distance might influence loop size. However, the current study proposes a contrasting model where cohesin-STAG1 establishes larger loops first in early-G1, followed by cohesin-STAG2 mediated formation of smaller nested loops in late-G1. How to reconcile these two models?*

We thank the Reviewer for raising this point and the opportunity to explain our model in more detail.

According to recent in vitro data of Cohesin loop extrusion (<https://doi.org/10.1126/science.aaz3418>, <https://doi.org/10.1126/science.aaz4475>), Cohesin can loop-extrude DNA with a rate of approximately 1kb/s. If the in vivo loop extrusion rate is roughly similar to the rate measured in vitro, large TAD loops of the size of 500kb may thus be generated within ~500 seconds, i.e. in less than 10 minutes. With Cohesin-STAG1's early chromatin binding starting from telophase, it is thus expected that long-range TAD-scale loops exist already early in G1. Our chromatin-tracing data of early G1 cells selected from a mildly synchronized cell population confirms the presence of these loops.

It would also be expected that small-scale loops begin to be visible during telophase. Although our data was primarily focused on earlyG1 and asynchronous interphase, our tracing experiments also covered few traces from cells in metaphase and telophase. Compared to earlyG1, smaller-scale average looping structures are visible. Please note that the average contact maps are off less quality due to the low numbers of traces sampled for the mitotic stages compared to earlyG1.

The publications referenced by the Reviewer, as well as the above mentioned Abramo et al. 2019 paper use extensive Nocodazole-mediated mitotic arrest, and subsequent release into interphase. This leads to a reduced dynamic sampling capability, as early G1 timepoints contain significant fractions of mitotic and telophase cells. In agreement with the loop extrusion model, according to which Cohesin travel distance correlates with loop size, we think that the small loops observed in the early G1 fractions are population averages across mitotic, early G1 and G1 cells, thus producing rather weak and smaller-scale loops. Only later after release into interphase when the cell populations are pure enough and devoid of telophase cells, we would anticipate to see clear long-range loops consistent with our purely early G1 and G1 measurements.

8. The authors' sequential DNA FISH data on STAG2-depleted cells provides valuable insights. However, including data from late-G1, in addition to early-G1 (or interphase), would strengthen their findings. STAG2 is known to be most functional during late-G1, and including this stage could potentially reveal a more pronounced effect of depletion on loop formation.

We thank the Reviewer for the appreciation of our chromatin tracing data and this interesting suggestion. Indeed, most loop-extruding (i.e. dynamically chromatin-bound) Cohesin-STAG2 should be available at the end of G1 as no cohesive Cohesin is formed yet and chromosomes are not yet replicated. To explore this, we have stratified cells in our tracing data set by nuclear size in G1 (small nuclei indicating earlier and large nuclei later G1 stage). Unfortunately, we only found a marginally more pronounced depletion effect in G1 cells with large nuclei and thus refrained from including this analysis into the revised manuscript.

Minor points:

1. Fig. 1C and E should also include STAG1 data to show its rapid increase in nuclear concentration.
2. Supplement Fig. 2E suggested that CTCF remains largely chromatin bound during mitosis, given the sharp decay of the FRAP curve. However, it is well established that CTCF is dramatically evicted from mitotic chromosomes. How to explain this discrepancy.

We appreciate the Reviewer's comment on CTCF binding dynamics during mitosis. As it is not central to the key messages of the manuscript, we did not detail it further in the manuscript, but we are happy to explain it here:

Indeed, most of CTCF is evicted from mitotic chromosomes and only a minor pool of CTCF co-localizes with mitotic chromosomes (Suppl. Figure 2P, Figure 3). However, CTCF is still mildly enriched over the cytosolic background, which is apparent in live cell imaging of mitotic CTCF-EGFP-tagged cells (see below).

Metaphase cell expressing CTCF-EGFP, image of CTCF-EGFP acquired in 488 nm channel is depicted on the left. Chromatin was stained using SiR-Hoechst and is displayed on the right.

Immediately after half-metaphase-plate bleaching, all soluble protein is depleted and a “post-bleach” z-stack is acquired to measure the maximal fluorescence difference between the bleached and unbleached half of the chromosome mass. This maximal difference thus reflects only the small amount of bound material and is normalized to 1 in the data analysis. This is similar for Cohesin, which is also only mildly enriched on metaphase chromatin (SFig. 2F).

3. *Supplement Fig. 2F measured the fraction of chromatin bound cohesin during interphase vs. mitosis. I'm curious how was this measured in the mitotic samples, since cohesin is completely evicted from mitotic chromosomes.*

See the comment to minor point 2 above.

A minor fraction of Cohesin is bound to mitotic chromosomes, which represents the cohesive Cohesin that is mostly concentrated at the chromosome's centromeres.

Reviewer #2 (Comments to the Authors (Required)):

We very much appreciate the enthusiastic assessment and comments of Reviewer 2. As for Reviewer 1, we have addressed the comments point-by-point below.

This paper from Brunner et al. combines a very impressive range of cutting-edge approaches - including quantitative live-cell imaging, lots of genome-editing to tag numerous proteins, degran-perturbations, FRAP, tiling-chromosome tracing experiments super-resolution microscopy and more - to dissect the role of SMC complexes, condensin and cohesin, CTCF and associated factors as the genome is being reshaped from mitosis-to-G1. The Ellenberg lab is arguably one of the world's leading labs when it comes to quantitative microscopy - units of proteins per cell or protein concentrations instead of arbitrary units - and as a resource alone, the present paper is incredibly valuable. Beyond this, the paper provides great insights into genome-refolding and makes a case that we should think of STAG1-cohesin and STAG2-cohesin akin to Condensin I and II, making smaller and larger loops. The authors also make a surprising observation that cohesin seems to stabilize the residence time of CTCF and discuss whether SMC extrusion complexes are likely to be monomeric or dimeric. The figures look beautiful and the manuscript is well-written. Properly one of the more impressive papers that integrate diverse and complementary methods that I have seen in a while. Overall, I think this is a fantastic paper that makes several major and original contributions to the field and I do believe it will be of broad and general interest including to the wide readership of the Journal of Cell Biology.

Nevertheless, there are some issues that need to be corrected in my opinion before this paper is ready for publication. I list these below.

MAJOR POINTS

PROTEINS PER MEGABASE

I very much like putting the numbers in terms of proteins per megabase, e.g. 3 cohesin extruders per average megabase. However, no information of how these calculations were made is provided (unless I missed it, in which case I do apologize). Can you please include a very detailed methods section on these calculations?

These calculations are more subtly than they sound. I can indirectly guess how they did it. In Table 2, authors say that there are 89280 CTCF proteins bound corresponding to 5.7 per megabase. This implies a genome size of $89280 \text{ proteins} / 5.7 \text{ proteins/Mb} = 15,663 \text{ Mb}$ or 15.6 billion basepairs. These numbers are for G1, so there should be no replication.

The haploid human genome is 3.3 Gb, so this would suggest that these HeLa cells are $15.6\text{Gb}/3.3\text{Gb} = 4.7\text{-ploid}$, so somewhere between tetraploid and pentaploid.

How were these numbers arrived at? I think including an SI figure showing how the ploidy was measured is required, as is a full methods description.

When the authors studied mitosis, did they then use a 9.4-ploid genome?

Just to be clear, I am not saying that what the authors did was incorrect, just that it is very important and necessary for them to fully describe what, why and how they did this.

Finally, I would like to see error propagation. Currently, the authors generally provide numbers without confidence intervals.

We thank the Reviewer for raising this point. To address it, we have added a more detailed description to the methods section, which we copy here for information:

“Calculation of Average Copy Numbers per Megabase DNA

Average protein copy numbers per megabase DNA were calculated from FCS-calibrated imaging data of mitotic exit (Fig. 1F, Suppl. Fig. 1Q). This data represents total protein copy numbers co-localizing with DNA for either 1) the entire chromosome mass during mitosis or 2) both daughter nuclei summed up (after anaphase onset). To calculate the protein copies that are bound to DNA, the average number of protein copies co-localizing with DNA during each given timepoint was multiplied with the respective chromatin-bound fraction estimate derived from spot-bleach or FRAP data.

The HeLa Kyoto genome is hypotriploid with 7.9 Gb (Landry et al., 2013). Thus, the total genomic content of replicated mitotic cells is 2x7.9 Gb. Similarly, the total genomic content of both daughter cells is 2x7.9 Gb. The total protein copy numbers were divided by 15,800 Mb to achieve an average per megabase count, assuming that all investigated proteins occupy the mappable HeLa genome at equal frequency.”

We thank the Reviewer for making us aware to report standard deviation of the reported means and the propagated errors of the reported per-megabase protein counts that are based on multiplication of nuclear protein numbers and bound fractions determined by FRAP. We have updated Table 1 and 2 accordingly, and removed supplementary tables 1 & 2 as we noted that the reported comparison between FRAP and spot-bleach bound fractions are already listed in tables 1& 2 as well.

HELA CELLS AND GENERALITY

Throughout the paper the tone is that the numbers and discoveries described in this paper are "universal truths" akin to laws of physics. But most likely, if the authors were to repeat these studies in diverse mouse and human cell lines, both some of the observations (e.g. telophase behavior) and numbers (e.g. density of SMC complexes) would change substantially. This is fine, and it would be impossible for the authors to perform these studies in all cell types. However, I do think it is important for the authors to describe the results in a less general and more open-minded manner. They should explicitly state that these results may be specific to HeLa cells and that it is no clear how different other cell types and species might be. Throughout the manuscript, I would also ask the authors to tone down their tone - right now, when I read it the tone implies that these are universal truths rather than numbers specific to HeLa cells cultured in a certain way.

The authors should also explicitly comment on the known issues with clone-to-clone variation and cancer cell line variation, which is well-known, see e.g. <https://www.nature.com/articles/s41592-019-0375-1>

We thank the reviewer for raising this important point. We have adapted our language by regularly mentioning that these measurements are from the HeLa cell model.

We also stated the use of the HeLa model system and its potential limitations explicitly at the end of the discussion section:

“The quantitative data and understanding provided by our study is based on our measurements in the HeLa Kyoto cell model system, a cancer cell line for which we have established a systematic collection of genome edited tools to comprehensively analyze SMC complexes quantitatively and perturb them. We note that previous work from our group and others have reported clonal variation of expression in HeLa cells (Walter et al., 2018; Liu et al., 2019). Nevertheless, the relative dynamic changes observed between different proteins during mitotic exit have been very consistent in our hands. In addition, we note that in our studies of mitotic exit behavior of a different nuclear protein complex, the NPC, a systematic comparative analysis between HeLa cells (Otsuka et al., 2023) and normal rat kidney cells (Dultz et al. 2008,2010) has shown a highly consistent behavior between

these cell types. In our view it is therefore reasonable to assume that the presented results on the behavior of the conserved SMC complexes during cell division measured in HeLa cells are likely to be similar in other human and animal cell lines. However, as for any other cell line, we cannot exclude that HeLa cells may exhibit cell type specific aspects.”

ABRAMO PAPER

The Abramo 2019 paper claims that telophase is completely devoid of SMCs and loop extrusion. However, this conclusion is somewhat controversial, and I know several people in the field who believe this Abramo conclusion is not correct.

The authors devote a whole section in the Discussion to this, but the writing and tone is so indirect, that even after reading the paper twice, I cannot tell for sure what the authors believe (though I think I can infer it). If I am not mistaken, the data in this paper disproves the Abramo conclusion and says that although there is reduced SMC complexes in telophase, there is no time when there are no SMC complexes on DNA.

I of course understand that the authors want to be polite and collegial, but I think it would be better if the authors could just clearly state if they results disagree with Abramo instead of the current phrasing which I found quite vague.

We thank the Reviewer for raising this point and we agree that we should explain this discrepancy more clearly.

As stated in the manuscript, our chromatin binding data is indeed consistent with the Abramo et al. 2019 chromatin fractionation data presented in Fig. 7a and Extended Data Fig. 8d. Both the Abramo et al. 2019 study and our results show a clear chromatin co-occupancy of Condensins and Cohesins during telophase.

However, indeed the authors of the Abramo et al. 2019 derived a different conclusion (i.e. telophase chromatin is devoid of Condensin and Cohesin loop extruders), likely because HiC data of telophase stages did not show a clear signature of looping structures. We thus suggest a different interpretation of this data and have made it more clear in the discussion section.

BOUND FRACTION CALCULATIONS

The bound fraction calculation appears to be just the fraction of pre-post/pre (Fig 2A). But the methods section says that they calibrate this against exogenous H2B-EGFP which they say represent 100% chromatin binding. This sounds reasonable, but I believe this is incorrect. Natural H2B is expressed during replication (S-phase). Exogenous H2B is expressed also in M, G1, G2 leading to a significant fraction of unbound H2B.

The extent of unbound H2B is typically around 20-25% (see Fig 4H in <https://elifesciences.org/articles/33125>), though no doubt depends on the expression system. Therefore, I anticipate this calibration - although intuitively reasonable - might lead to the wrong answer.

We agree with Reviewer 2 that not all nuclear H2B is fully chromatin-bound, and depending on the expression system, more or less H2B is chromatin-bound.

Bearing this in mind, we have chosen cells expressing H2B-EGFP at low levels, and we have quantified the chromatin-bound fraction of H2B using the orthogonal approach via half-nuclear photobleaching. The chromatin-bound fraction is derived through an estimate of the soluble protein fraction. This soluble protein fraction is derived as follows:

We measure fluorescence on chromatin pre- and post-bleach, then calculate the loss of fluorescence in the unbleached half of chromatin. Unbleached soluble protein of the unbleached half will be partially bleached due to its diffusion into the bleach region, or diluted through its diffusion inside the whole nucleus.

These measurements have shown that our cell line expressing H2B-EGFP has chromatin-bound fractions of ~90%.

We decided that it is not useful to attempt to correct for this small fraction of unbound H2B, as it is negligible given the precision of our spot-bleach measurements. In these, the “pre-bleach” and “post-bleach” measurements are averages of the first and last 500 milliseconds of the continuous FCS-like spot-bleach measurement. The “pre-bleach” measurement represents the unbleached starting point and is a sum of chromatin-bound and unbound fractions of the protein. The “post-bleach” represents only the unbound fraction. During the first 500 milliseconds, a small portion of the protein of interest bleaches, thus leading to a slight underestimation of the chromatin-bound pool.

Since the magnitude of H2B unbound fraction is on a similar scale as this minor time averaging effect, we decided not to correct them mathematically. Rather, we confirmed the validity of the spot-bleach assay’s measurements through the orthogonal half-nuclear bleaching FRAP method, as outlined in Supplementary Table 1 & 2.

Moreover, the calculation in Fig 2A seems too simplistic when the authors can instead do proper reaction-diffusion modeling of spot-FRAP. Indeed, the bound fractions listed in SI Table 2 for CTCF and cohesin appear too high and inconsistent with several papers from several labs that have used SMT/SPT/FRAP to study these complexes in diverse cell types.

Can the authors re-visit this calculation and their approach?

This comment address two different topics, so we address them one by one below:

Regarding reaction-diffusion modeling of the spot-bleach data:

Indeed, our analysis approach of the spot bleach data is a simplistic one. Our aim with this assay was to establish a rapid and reproducible measure of chromatin-binding during early G1 that can be repeatedly performed in single cells with accurate timing during mitotic exit. However, this assay also has limitations in the degree to which the details of binding can be studied and we therefore did not apply more complex reaction diffusion models to analyze it. Rather, we performed more precise investigations of the protein’s residence time on chromatin during early G1 and G1 using half-nuclear photobleaching. This can however not be repeated many times in the postmitotic cell where binding changes dynamically, as unbound material is almost completely depleted in the first half-nuclear bleach and the laser intensities used would interfere with mitotic progression.

Regarding the bound fractions listed in SI2 for CTCF and Cohesin:

The relatively high chromatin-bound fractions measured in the present study by spot-bleach are explained by the short time intervals between pre-bleach and post-bleach. In the spot-bleach assay, chromatin-bound proteins with an average residence time above 30 seconds are effectively considered as “chromatin-bound” as they do not leave the bleached spot in the 30 second measurement period of the assay.

Similarly, for half-nuclear bleaching, we ensured that the time between the pre-bleach and the post-bleach measurement is minimal (photobleaching process occurs 3x with 10 second wait times in between, thus leading to ~30 seconds time between pre- and post-bleach acquisitions). This ensures that chromatin-bound proteins, especially the rather short-lived dynamically bound fraction, do not have the chance to unbind and diffuse off.

When comparing our data to previous FRAP measurements performed on interphase Cohesin, one should bear in mind that they have been performed on many cells at the same time (personal communication with Dr. Kota Nagasaka), leading to longer and variable lag times between the pre- and post-bleach image acquisition. Dynamically chromatin-bound proteins thus had more time to unbind and diffuse off, explaining the lower overall chromatin-bound fractions derived from this data.

When comparing our data to single particle tracking data, the rapid bleaching of tracked molecules in SPT should be borne in mind, which often lead to loss of focus or tracking errors. These errors cannot be distinguished from protein unbinding from chromatin, thus typically leading to an underestimation of the chromatin-bound fraction, as well as the dynamic residence time on chromatin. This issue is recognized, see for example <https://doi.org/10.1042/BST20221242> :

“In slow SMT, a residence time distribution is obtained by classifying ‘bound’ molecules that move less than a certain distance and tabulating how many frames they persist. Interpreting this distribution is complicated by the fact that fluorophores may disappear not only due to dissociation but also due to photobleaching, defocalization, or tracking errors [62]”

And <https://www.nature.com/articles/s41598-021-88802-7> (cited above as [62])

“We [...] found that tracking errors, similar to fluorophore photobleaching, have to be considered for reliable analysis.”

MONOMERIC VS. DIMERIC SMC COMPLEXES

The authors do some very nice and quantitative STED imaging in Fig 5 and discuss the controversial issue of monomeric vs. dimeric SMC complexes. I think the authors make a compelling case that the dimeric and multimeric complexes they observe may indeed be adjacent, stacked extruders. However, I think they overstate things. They do at some point state that they cannot fully exclude dimeric SMC extruders, but overall I would like to ask the authors to tone down this a little bit, since I do not believe they can fully exclude concentration-dependent dimerization (indeed, the in vitro single-molecule extrusion papers also find concentration dependent dimerization).

We thank the Reviewer for the appreciation of our quantitative STED imaging and appreciate this comment and word of caution. We adapted our discussion section accordingly.

NESTED LOOPS AND COHESIN AND CONDENSIN ANALOGIES

On the topic of nested loops and loop density, the difference in compaction between mitotic and interphase chromosomes is currently thought to result largely from differences in loop-extruder density. If you do a polymer simulation, high densities of loop extruders make compacted mitotic chromosomes and lower densities make interphase-like less compacted chromosomes, see e.g. <https://elifesciences.org/articles/14864>

E.g. in Table 2, they say they have 11.6 cohesins per megabase in G1. If you have >10 extruders per Mb would this not result in mitosis-like chromosome compaction? I would like to ask the authors to perform quantitative comparisons between their data and polymer simulation studies to see if the extruder densities make sense with interphase chromosome compaction?

Along these lines the authors make a nice analogy between Condensin I and II and cohesin STAG1 and STAG2 making long loops and nested shorter loops inside. This analogy is highly intellectually appealing, since it suggests a universal nested looping model for SMC extrusion in both mitosis and interphase. However, we also know that the mitosis compaction level is much higher due to much higher loop-extruder density, suggesting that the nested loops inside loops frequency is likely much

much lower in interphase. Can the authors more quantitatively compare these including in terms of frequency of nested looping in mitosis and interphase based on their quantitative measurements? I am not suggesting a very extensive analysis, but the polymer simulations studies have typically processivity/separation ratios that produce interphase and mitotic chromosomes.

In line with the Reviewer's comment, we were also surprised about the relatively high number of Cohesins relative to Condensins.

We assume that compaction differences between mitosis and interphase can be partially attributed to 1) more stable chromatin binding of Condensins relative to Cohesins and thus potentially larger genomic loop sizes and 2) the gapless arrangement of loops on chromatin during mitosis. In interphase, chromatin most likely is not gaplessly extruded due to the presence of CTCF as boundary factor.

Further, a recent study convincingly showed that histone modifications add to the compaction of mitotic chromosomes (PMID: 35922507).

CHROMOSOME TRACING

The chromosome conformation tracing experiments are really nice and the precision and resolution (12kb, 20 nm) seems very impressive. The fact that they see sub-domain structure, akin to high-resolution Micro-C, e.g. in Fig 5H is quite astonishing and not something I have seen before. All other chromosome tracing papers I have read generally produce quite low quality and blurry contact maps that look substantially worse than Hi-C and Micro-C. I don't really have a question or concern, just want to comment on how high-quality the data appears to be. Maybe this is also partially because they use a better FISH protocol (RASER-FISH) that does not disrupt structure so much.

We very much thank the Reviewer for the appreciation of our chromatin tracing workflow. Indeed, RASER-FISH substantially improves the structural preservation and trace quality (Beckwith et al. 2023, SFig. 1)

MINOR POINTS:

1. *Line numbers would have been helpful*

We apologize if the absence of line numbers caused an inconvenience during review and have added line numbers in the revised manuscript.

2. *"seamlessly" feels like an inappropriate word for the abstract. Too anthropomorphic.*

To contrast to the the main conclusion of Abramo et al. 2019, we tried to emphasize the continuous and overlapping nature of the transition between Condensins and Cohesins we observed during mitotic exit. "Seamless" appeared appropriate to us in this context.

3. *"Zink" should be "Zinc" on page 2 thank you for the identification of this typo*

4. *Page 2, the Zuin 2022 paper is very nice but not really relevant to the sentence as it does not deal with TADs. A better reference would be only of the 2016. 2019, 2024 Bernstein papers on Glioma or GIST and CTCF insulator loss leading to aberrant E-P interactions and oncogene overexpression.*

We thank the Reviewer for this better fitting suggestion which we have included in the revised manuscript.

5. *Page 3 top, the authors state that STAG2 makes up 75% of cohesin. This is not universally true, but varies enormously from cell type to cell type. Please correct.*

We added "...in HeLa cells" to make this clear.

6. Page 7, authors should compare their cohesin and CTCF residence times to other studies and discuss. E.g. CTCF result seems similar to Hansen 2017, but STAG1 result seems much shorter than Wutz 2020.

We extended the results section on FRAP to note the differences with other published papers and explained why our FRAP measurements result in higher bound fractions for Cohesins and CTCF and shorter residence times for Cohesins. Please also see our explanation on this topic provided above.

7. Page 11 "... longer lived loop structures in interphase, equivalent of TADs", could this stabilization perhaps explain why the protein residence time reported here is a bit shorter than the recently measured loop lifetime of ~20 min (Gabriele 2022).

We think that the residence times measured in our FRAP measurements are biased towards shorter residence times due 1) immediate imaging after bleaching and 2) the focus on the recovery only for the first 10 minutes after bleaching. Experimentally, it becomes very challenging to perform longer stable FRAP measurements of the same nuclear z-slices as the cell settles down after division, thus limiting the maximal possible length of our early G1 measurements. For best comparability, we kept the G1 measurements exactly the same as for early G1. With more data points for longer recoveries, we would expect to see somewhat longer residence times. Our fitting procedure assigns the long-term bound proteins mostly to the "immobile fraction".

Reviewer #3 (Comments to the Authors (Required)):

We thank the Reviewer for the appreciation of our work and the constructive comments that we have addressed as outlined below.

Brunner et al. investigate the localization of cohesin and condensin through mitosis and early interphase. The authors determine that cohesin and condensin are both on the DNA in early G1, and show that cohesin-STAG1 is rapidly imported into the nucleus, while the more abundant cohesin-STAG2 slowly accumulates in the nucleus. This leads the authors to a hypothesis that cohesin-STAG1 and cohesin-STAG2 form a nested-loop structure reminiscent of the structures that condensin II and condensin I can form. The authors use super-resolution imaging to show that STAG2-cohesin seems to cluster together as it starts to accumulate in the nucleus, and propose that when these complexes encounter each other, they may dimerize.

I much enjoyed reading this manuscript. This is a very thorough study, and the figures are clear and appealing. Though some of the data was already known (Figures 3F and 4C/D come to mind), the manuscript also provides valuable new insights into possible dimerization of cohesin complexes. I have only minor comments. Most of them textual:

Textual comments

- Throughout the results section, the authors are careful not to use the term dimerization, but rather colocalization. There are still instances where dimerization is used, like in the results section heading. I think colocalization is a more accurate term in this case. Fine of course to in the discussion consider that colocalization may reflect dimerization, but probably best to adjust the relevant heading in the results section.

We have adjusted the heading according to the Reviewers suggestion.

- The 2018 paper from the same lab also contains quantifications of condensin complexes on DNA. It would be good to include a direct comparison of the quantifications from 2018 and the new quantifications.

We have explained this in the Methods section and Supplementary Fig 1A and now added a direct comparison between the recent and current study according to the Reviewers suggestion:

Protein numbers on chromosomes during anaphase onset (= at the chromatin binding peak; soluble chromatin fraction subtracted as described in Walther et al. 2018).

- *The nested loop hypothesis of STAG1 and STAG2 was previously proposed by Wutz et al. (2020). Though the authors do extensively reference this paper, it might be good to add that this hypothesis is not entirely novel, and to thus also reference Wutz in the context of this model.*

We have adapted our text to reference the Wutz et al. paper as the original proposer of this model.

In Results section:

“In line with a recent study (Wutz et al., 2020), we therefore propose a double hierarchical loop model for the transition from mitotic to interphase loop extruder driven genome architecture in HeLa cells”

In Discussion section:

“we propose a hierarchical nested loop model for the establishment of the interphase genome organization by the Cohesin loop extruders after mitosis, as has been done previously based on HiC and chromatin binding data of Cohesins during G1 (Wutz et al., 2020).”

- *In their model as well as in the text, the authors claim that there are "no functional interactions between extruders" cohesin and condensin. Though they show that the levels of cohesin are not influenced by condensin depletion and vice versa, I do not think that this proves that there is no functional interaction between the complexes at all. So best to rephrase.*

We rephrased such that it is clear that there is no functional interaction influencing the chromatin binding of each other loop extruder complex.

In results:

“While we cannot exclude a functional interaction of Condensins and Cohesins beyond chromatin binding, this result suggested that mitotic and interphase loop extruders bind to chromatin simultaneously but independently of another during early G1”

And in discussion:

“Although Condensin helps evict chromatin-bound Cohesin during prophase (Hirota et al., 2004; Samejima et al., 2024), we did not find evidence for a functional Condensin-Cohesin interaction that impacts their chromatin binding during telophase and early G1 (Fig. 2D,E).”

In my opinion, some of the data could be moved to the supplemental figures. The reason being that some of the main figures are currently a bit dense. I would suggest moving the following panels to the supplements:

- *Nup153-dependent import of cohesin.*

We would prefer to leave this figure a main figure as we deem it as an important information. Furthermore, we note that we are limited to a total of 5 supplementary figures, which are also rather busy already. On balance we would prefer to leave the figure as is.

- *Effects of cohesin depletion on condensin levels, and vice versa.*

We think that Figure 2 is not overly crowded yet, and would prefer to leave it in its current form.

Finally, I have some questions regarding the presented data:

- *Condensin I levels in the interphase nucleus remain higher than the condensin II levels. This is surprising, as it was previously thought that condensin I is excluded from the nucleus in interphase. I would like to invite the authors to elaborate a bit on this interesting observation.*

We were also surprised by this large amount of Condensin being retained inside the interphase nucleus. However, our quantitative live cell imaging showed that Condensin I became gradually diluted in the growing interphase nucleus and its concentration was not actively restored by newly synthesized protein (Fig 1D, SFig. 1E), indicating no stoichiometric role during interphase. Combined with the fact that Condensin I exhibits a high diffusive mobility in the interphase nucleus (SFig. 1H) further suggests that these complexes are not strongly chromatin bound, as would be expected for progressive loop extrusion.

- For the spot-brightness measurements of a single nanobody, the manuscript states "preliminary data, not shown". In my view, this data should be included.

We now display the data in supplementary figure 5H.

- Ochs et al. (2024) show that cohesin might form dimers, and that this could make up a large proportion of the looping cohesin. I would like to see an estimate of the proportion of dimers that the authors see in their data.

We tried to assess the proportion of monomers and dimers using gaussian mixture models.

Our sample data is near unimodal, making it difficult to fit a bimodal distribution and calculate weights of the respective gaussian distribution.

To nevertheless explore the reviewer's suggestion, we computationally predicted the respective "missing" monomer or dimer peak, to be able to fit a bimodal distribution despite the near-uniform distribution of our sample data. After fitting of the GMM and extraction of weights, we back-calculated the monomer/dimer contribution of the sample.

Based on this approach, we calculated that 8-10% of Cohesin-STAG1 exist as dimers in early G1 and G1; for Cohesin-STAG2 in early G1, we found 6% dimers, and for Cohesin-STAG2 in G1, we found roughly 90% as dimer.

As this analysis required the addition of simulated data for a gaussian mixture model fit in our view it is too preliminary to include into the manuscript.

- It is hypothesized that colocalizing STAG2-cohesin dots do not actively extrude anymore. Does this mean that the complexes are at or near CTCF at this point?

Sub-TAD borders indeed extensively co-localize with CTCF sites in our 3 1 megabase regions that we investigated. It is thus likely that CTCF supports the stalling of Cohesin complexes.

- Do putative 'heterodimers' of cohesin-STAG1 and cohesin-STAG2 also occur, or is it exclusively cohesin-STAG2 that colocalizes with itself?

In our experimental setup, Cohesin-STAG1/2 are detected in the respective endogenous EGFP knock-in cell line via GFP nanobody, to maximize labelling efficiency and allow for a quantitative interpretation of the data. This setup precluded to image STAG1/2 in the same cells and we can thus not address their potential colocalization.

Dear Editor, dear Reviewer,

For the review of our manuscript entitled "**Quantitative imaging of loop extruders rebuilding interphase genome architecture after mitosis**", we were asked to provide

"an orthogonal measure of chromatin occupancy, such as chromatin pulldowns, to corroborate measurements of occupancy by STAG1/2 and CTCF"

Following this suggestion, we have tried extensively to establish a chromatin fractionation and quantitative western blotting in the lab. Unfortunately, we have encountered many technical issues that have prohibited us from providing a reasonable western-blot based result. For a breakdown of our work, see page 2.

The failed western-blot attempts motivated us to carry out the validation experiment with the pre-extraction + immunofluorescence assay that was already presented in Figure 2D&E. We successfully validated the assay's performance with our proteins of interest and could reproduce the expected chromatin association data from our FCS-calibrated imaging and spot-bleach data. For a breakdown of the results, see page 3-6.

Breakdown of western blot attempts:

- Clean synchronization of cells in early G1 is not trivial with current cell synchronization methods such as Nocodazole treatment followed by mitotic shake-off and release into mitotic exit. These methods have worked well for us in combination with microscopy, as we can pick out the right cells and ignore mitotic contaminants. However, this is not possible in a large-scale biochemical fractionation assay.
- We have tried two fractionation protocols, the one used by Abramo et al. 2019 (PMID: 31685986) and a second published one (PMID: 31685986). After some work to establish the routines in the lab, both protocols resulted in good fraction of chromatin, as judged by GAPDH (marker for soluble protein) and H2B (control protein for chromatin). We have decided to use the Abramo et al. 2019 protocol due to its ease of use and since chromatin association during mitotic exit has been tested with it previously.
- We've initially performed western blotting on our traditional western blotting setup, using Luminol+Peroxidase for secondary antibody readout.
 - > we could not identify a set of antibodies providing clean bands for the SMC complexes, especially STAG1
 - > the Peroxidase-based detection was unreliable and prohibited quantitative readout
- We thus decided to move to the "Simple Western" system that we have used in our manuscript for HeLa knock-in validations.
 - > here, STAG1, STAG2 and CTCF antibodies worked well in regular cell lysates, however their performance turned out to be much worse in chromatin fractions, leading to bands with a lot of background
 - > we have tested a range of H2A, H2B and H3 antibodies (8 in total). While all of them worked on the traditional western blot setup, none of them worked in the Simple Western system. Having a decent histone antibody would have been essential to provide a robust normalization of chromatin input.
 - > instead of histones, we tested whether we can normalize the sample input based on total protein content in the chromatin fractions, although this likely will differ between mitotic, eG1 and G1 samples. Unreliable protein quantification by BCA lead to very different amounts of total protein input, which meant we had to do a first run to accurately adjust sample concentrations, and a second one to compare samples.
- In summary, we would need to go through another extensive round of antibody testing and assay optimization until we are in a position to validate the FRAP results by fractionation and western blot. Given our pre-extraction + IF results that we have prepared in the meantime, we think that this additional experimental effort and consumable consumption is not justified.

Pre-extraction + Immunofluorescence

Pre-extraction is a classical assay in cell biology, to remove the soluble fraction of a protein and clear the view for proteins bound to chromatin. It has been used recently in Walther et al. 2018 (PMID: 29632028) to remove soluble Condensin proteins to better visualize chromatin-bound proteins during mitosis.

We have adapted this assay to remove/penetrate cellular and nuclear membranes and liberate the soluble protein pool, in order to retain the chromatin-bound pool. After short pre-extraction, cells are chemically fixed using PFA and stained using immunofluorescence, in order to visualize and quantify the remaining chromatin-bound pool.

We have combined this experiment with cell synchronization and classification in early G1 or G1 to quantify the total chromatin-bound pool in early G1 and G1.

A breakdown of the experiment is provided:

- 1) Cells are seeded in a T175 flask in culture medium one day prior to the experiment
- 2) 1h prior to mitotic shake-off, cells are incubated with Nocodazole to enrich them in mitosis
- 3) Mitotic cells are shaken-off and seeded into microscopy slides, cells are allowed to attach in the presence of Nocodazole
- 4) Cells are released into mitotic exit by washing out Nocodazole. Cells are allowed to undergo mitotic exit either for 45 minutes (= early G1 cells) or 4h (= G1 cells). Note that the Nocodazole-based release is not perfectly synchronous and the early G1 sample will be not pure
- 5) Prior to PFA-based fixation, cells are pre-extracted for 1 minute with 0.25% Tergitol in PBS. This liberates the soluble protein pool and washes it away.
- 6) Cells are fixed with PFA and proteins of interest are stained by IF

To be sure that we are indeed removing/penetrating nuclear membranes, we have tested our pre-extraction assay on G1 cells (see next page).

Validation on G1 cells:

We have fixed G1 cells either without pre-extraction (=non-extracted) or after 1-minute pre-extraction using 0.25% Tergitol in PBS. We have acquired complete high-resolution 3D stacks of >100 cells per condition to integrate the total nuclear fluorescence intensity and compare non-extracted (set to 100%) with the pre-extracted samples. We have compared this data to our FRAP bound-fraction data (Manuscript Table 2) and found that the pre-extraction experiment works very well for STAG1 and STAG2, and ok for CTCF.

Control: pre-extraction in G1 (4h post release)

We have then used this assay to perform pre-extraction of early G1 cells and G1 cells. See next page.

Validation of chromatin binding of STAG1/2 and CTCF

We have performed pre-extraction + IF of early G1 cells and G1 cells. Due to the microscopy approach, we could sort out cells that were undergoing anaphase, as well as cells that are still in mitosis. The mitotic cells measurements were used as additional source of assay validation (see below).

Again, the total protein content per nucleus was integrated as a measure of the total chromatin-bound pool. Integrated intensities per nucleus were normalized to G1 cell measurements.

This allowed for easy comparison to our quantitative imaging data (FCS-calibrated imaging + spot-bleach) that we have shown in our manuscript figure 2L. For earlyG1, we have pooled the data from +6 minutes after AO until 30 minutes after AO (to reflect the heterogeneity in our pre-extracted sample). For G1, we have pooled the measurements from +110-130 minutes after AO.

Clearly, both replicate experiments validate our findings, as

- 1) Cohesin isoforms and CTCF are almost completely evicted from chromatin during mitosis
- 2) Cohesin-STAG1 and STAG2 associate with chromatin during earlyG1
- 3) Relative to G1, more STAG1 and CTCF is already bound in early G1 compared to STAG2.

Exemplary microscopy images are displayed on the next page.

Exemplary microscopy images of pre-extracted earlyG1 and G1 cells probed for the indicated proteins of interest. Dynamic range of fluorescence intensity measurements was set equal for eG1 and G1 measurements of STAG1, STAG2 and CTCF, respectively.

October 9, 2024

RE: JCB Manuscript #202405169R

Dr. Jan Ellenberg
European Molecular Biology Laboratory (EMBL)
Cell Biology and Biophysics Unit
Meyerhofstraße 1
Heidelberg, Baden-Württemberg 69117
Germany

Dear Dr. Ellenberg:

Thank you for submitting your revised manuscript entitled "Quantitative imaging of loop extruders rebuilding interphase genome architecture after mitosis". We would be happy to publish your paper in JCB pending final revisions necessary to meet our formatting guidelines (see details below).

A. MANUSCRIPT ORGANIZATION AND FORMATTING:

Full guidelines are available on our Instructions for Authors page, <http://jcb.rupress.org/submission-guidelines#revised>. Submission of a paper that does not conform to JCB guidelines will delay the acceptance of your manuscript.

1) Text limits: Character count for Articles is < 40,000, not including spaces. Count includes abstract, introduction, results, discussion, and acknowledgments. Count does not include title page, figure legends, materials and methods, references, tables, or supplemental legends.

2) Figures limits: Articles may have up to 10 main figures and 5 supplemental figures/tables.

** Please reformat supplemental figures to occupy a single page. Since this manuscript currently contains 6 main figures, you may make a new main figure(s) from existing supplemental data panels.

3) Figure formatting: Scale bars must be present on all microscopy images, including inset magnifications. Molecular weight or nucleic acid size markers must be included on all gel electrophoresis. Please avoid pairing red and green for images and graphs to ensure legibility for color-blind readers. If red and green are paired for images, please ensure that the particular red and green hues used in micrographs are distinctive with any of the colorblind types. If not, please modify colors accordingly or provide separate images of the individual channels.

4) Statistical analysis: Error bars on graphic representations of numerical data must be clearly described in the figure legend. The number of independent data points (n) represented in a graph must be indicated in the legend. Statistical methods should be explained in full in the materials and methods. For figures presenting pooled data the statistical measure should be defined in the figure legends. Please also be sure to indicate the statistical tests used in each of your experiments (either in the figure legend itself or in a separate methods section) as well as the parameters of the test (for example, if you ran a t-test, please indicate if it was one- or two-sided, etc.). Also, if you used parametric tests, please indicate if the data distribution was tested for normality (and if so, how). If not, you must state something to the effect that "Data distribution was assumed to be normal but this was not formally tested."

** Because JCB does not permit claims without support from primary data, please remove the statement in the caption for Figure 1I or include a panel with these data.

** Please include descriptions of error bars in Figure 2B, Figure S2K, Figure S3D and error bands in Figure S1E-G, S1M-N, and Figure S2C-F.

** Please indicate n for Figure 2D, 2E, Figure S1A, S1I-L, S1U, Figure S2C-F, S2I, and Figure S3D.

** Please indicate statistical tests used in Figure S5I-J.

** Please ensure numerical precision of values matches that of the calculated error shown in Tables 1 and 2 by correcting data in columns "Fraction long term bound" (Table 1), "Chromatin-bound (spot-bleach)" (Table 2), and "Fraction long term bound" (Table 2).

5) Abstract and title: The abstract should be no longer than 160 words and should communicate the significance of the paper for a general audience. The title should be less than 100 characters including spaces. Make the title concise but accessible to a general readership.

6) Materials and methods: Should be comprehensive and not simply reference a previous publication or preprint for details on

how an experiment was performed. Please provide full descriptions in the text for readers who may not have access to referenced manuscripts. We also provide a report from SciScore and an associate score, which we encourage you to use as a means of evaluating and improving the methods section.

** Please include a description of the generation of the primary CTCF antibody from Wutz et al.

** Please briefly describe the CRISPR knock-in validation protocol from Kueblbeck et al.

** Please remove the reference: Beckwich, Brunner et al., in preparation.

7) Please be sure to provide the sequences for all of your primers/oligos, plasmids, and RNAi constructs in the materials and methods. You must also indicate in the methods the source, species, and catalog numbers (where appropriate) for all of your antibodies. Please also indicate the acquisition and quantification methods for immunoblotting/western blots.

8) Microscope image acquisition: The following information must be provided about the acquisition and processing of images:

a. Make and model of microscope

b. Type, magnification, and numerical aperture of the objective lenses

c. Temperature

d. Imaging medium

e. Fluorochromes

f. Camera make and model

g. Acquisition software

h. Any software used for image processing subsequent to data acquisition. Please include details and types of operations involved (e.g., type of deconvolution, 3D reconstitutions, surface or volume rendering, gamma adjustments, etc.).

10) Supplemental materials: There are strict limits on the allowable amount of supplemental data. Articles may have up to 5 supplemental figures. Please also note that tables, like figures, should be provided as individual, editable files. A summary of all supplemental material should appear at the end of the Materials and methods section.

13) ORCID IDs: ORCID IDs are unique identifiers allowing researchers to create a record of their various scholarly contributions in a single place. At resubmission of your final files, please provide an ORCID ID for all authors.

15) A data availability statement is required for all research article submissions. The statement should address all data underlying the research presented in the manuscript. Please visit the JCB instructions for authors for guidelines and examples of statements at (<https://rupress.org/jcb/pages/editorial-policies#data-availability-statement>).

Please note that JCB requires authors to submit Source Data used to generate figures containing gels and Western blots with all revised manuscripts. This Source Data consists of fully uncropped and unprocessed images for each gel/blot displayed in the main and supplemental figures. Since your paper includes cropped gel and/or blot images, please be sure to provide one Source Data file for each figure that contains gels and/or blots along with your revised manuscript files. File names for Source Data figures should be alphanumeric without any spaces or special characters (i.e., SourceDataF#, where F# refers to the associated main figure number or SourceDataFS# for those associated with Supplementary figures). The lanes of the gels/blots should be labeled as they are in the associated figure, the place where cropping was applied should be marked (with a box), and molecular weight/size standards should be labeled wherever possible. Source Data files will be directly linked to specific figures in the published article.

The source code for all custom computational methods published in JCB must be made freely available as supplemental material hosted at www.jcb.org. Please contact the JCB Editorial Office to find out how to submit your custom macros, code for

custom algorithms, etc. Generally, these are provided as raw code in a .txt file or as other file types in a .zip file. Please also include a one-sentence summary of each file in the Online Supplemental Material paragraph of your manuscript.

** The custom scripts referenced from Cai et al., from Cattoglio et al., and from Walther et al., must be included as supplemental material or with a link to an online repository such as GitHub.

Journal of Cell Biology now requires a data availability statement for all research article submissions. These statements will be published in the article directly above the Acknowledgments. The statement should address all data underlying the research presented in the manuscript. Please visit the JCB instructions for authors for guidelines and examples of statements at (<https://rupress.org/jcb/pages/editorial-policies#data-availability-statement>).

B. FINAL FILES:

Thank you for your attention to these final processing requirements. Please revise and format the manuscript and upload materials within 7 days. If you need an extension for whatever reason, please let us know and we can work with you to determine a suitable revision period.

Thank you for this interesting contribution, we look forward to publishing your paper in Journal of Cell Biology.

Sincerely,

Ana Pombo
Monitoring Editor
Journal of Cell Biology

Tim Fessenden
Scientific Editor
Journal of Cell Biology

Reviewer #1 (Comments to the Authors (Required)):

The authors reasonably addressed all my concerns.

I have no further comments.

Reviewer #2 (Comments to the Authors (Required)):

The authors have addressed most of my comments. I am a little disappointed that they did not properly address some of my major and minor comments, most importantly, I do not think some of the bound fraction calculations are fully correct, and their results do seem to disagree with more accurate methods (most notably, properly analyze "fast" single-molecule tracking that accounts for defocalization with minimal tracking errors).

But these are relatively minor and only mildly affect the results, and I do think this paper is pretty important and provides key new insights into the key genome organizers from mitosis to interphase, so I think it may just be worth avoiding further delays and I support the publication of this otherwise excellent paper.

Reviewer #3 (Comments to the Authors (Required)):

The authors have addressed all my comments. Publication of this manuscript is highly recommended. I compliment the authors with this wonderful paper!